# An allosteric switch between the activation loop and a c-terminal palindromic phospho-motif controls c-Src function

Hipólito Nicolás Cuesta-Hernández[1,8], Julia Contreras [1,8], Pablo Soriano-Maldonado[1,6], Jana Sánchez-Wandelmer[2], Wayland Yeung[3], Ana Martín-Hurtado[1], Inés G. Muñoz [4], Natarajan Kannan [3], Marta Llimargas [5], Javier Muñoz[2,7] & Iván Plaza-Menacho [1] ✉

Autophosphorylation controls the transition between discrete functional and conformational states in protein kinases, yet the structural and molecular determinants underlying this fundamental process remain unclear. Here we show that c-terminal Tyr 530 is a de facto c-Src autophosphorylation site with slow time-resolution kinetics and a strong intermolecular component. On the contrary, activation-loop Tyr 419 undergoes faster kinetics and a cis-to-trans phosphorylation switch that controls c-terminal Tyr 530 autophosphorylation, enzyme specificity, and strikingly, c-Src non-catalytic function as a substrate. In line with this, we visualize by X-ray crystallography a snapshot of Tyr 530 intermolecular autophosphorylation. In an asymmetric arrangement of both catalytic domains, a c-terminal palindromic phospho-motif flanking Tyr 530 on the substrate molecule engages the G-loop of the active kinase adopting a position ready for entry into the catalytic cleft. Perturbation of the phospho-motif accounts for c-Src dysfunction as indicated by viral and colorectal cancer (CRC)-associated c-terminal deleted variants. We show that c-terminal residues 531 to 536 are required for c-Src Tyr 530 autophosphorylation, and such a detrimental effect is caused by the substrate molecule inhibiting allosterically the active kinase. Our work reveals a crosstalk between the activation and c-terminal segments that control the allosteric interplay between substrate- and enzyme-acting kinases during autophosphorylation.

In 1980, Hunter and Sefton found that the products of the Rous Sarcoma Virus transforming gene *v-Src* and its cellular homolog *c-Src* phosphorylated tyrosine residues[1,2]. This was the first report of a protein kinase with tyrosine specificity, a seminal work for the cancer research field that led to the discovery of many other viral oncogenes that encoded proteins with tyrosine kinase activity[3]. The further identification of oncogenic mutations in human genes that coded for receptor tyrosine kinases acting as drivers in human cancers such as

[1]Kinases, Protein Phosphorylation and Cancer Group, Structural Biology Programme, Spanish National Cancer Research Center (CNIO), C/Melchor Fernández Almagro num. 3, 28029 Madrid, Spain. [2]Proteomics Unit, Spanish National Cancer Research Center (CNIO), C/Melchor Fernández Almagro num. 3, 28029 Madrid, Spain. [3]Institute of Bioinformatics, Department of Biochemistry & Molecular Biology, University of Georgia, Athens, GA 30602, USA. [4]Protein Crystallography Unit, Spanish National Cancer Research Center (CNIO), C/Melchor Fernández Almagro num. 3, 28029 Madrid, Spain. [5]Institute of Molecular Biology of Barcelona (IMBB) CSIC, 08028 Barcelona, Spain. [6]Present address: Faculty of Experimental Sciences, Universidad Francisco de Vitoria (UFV), 28223 Pozuelo de Alarcón Madrid, Spain. [7]Present address: Ikerbasque, Basque Foundation for Science, IIS Biocruces Bizkaia, Building Biocruces Bizkaia 1, 48903 Cruces, Bizkaia, Spain. [8]These authors contributed equally: Hipólito Nicolás Cuesta-Hernández, Julia Contreras. ✉e-mail: iplaza@cnio.es

lung (*EGFR*), breast (*HER2*), gastric (*c-Kit*), renal and hepatocellular (*MET*) and thyroid (*RET*) cancers among others, illustrated the causative role of this family of proteins in cancer[4,5]. Over the last thirty years, the structural and molecular understanding of proteins with tyrosine kinase activity has allowed the design and development of an important number of new anti-cancer drugs, including small-molecule inhibitors and monoclonal antibodies that revolutionized targeted and personalized therapies in oncology, e.g., Gleevec and Herceptin[6,7].

c-Src belongs to the Src family of kinases (SFKs) composed of 11 structurally similar non-receptor protein tyrosine kinases: Src, Fyn, Lyn, Yes, Blk, Lck, Hck, Fgr, Srm, Brk and Yrk[8], all of which share a conserved arrangement of four different functional domains, named Src homology (SH) domains as well as a regulatory c-terminal sequence. The intrinsically disordered N-terminal region SH4 domain undergoes myristylation on a conserved glycine required for membrane attachment and localization, regulation of kinase activity, and intracellular stability[9,10]. SH3 and SH2 domains allow inter- as well as intramolecular interactions with proline-rich[11–13] and phospho-tyrosine-containing sequences[14,15], respectively, that are critical for c-Src functional regulation. The SH1 catalytic domain displays intrinsic tyrosine kinase activity, which is further regulated by highly dynamic activation and c-terminal regulatory segments, which are also intrinsically disordered regions[16]. c-Src activity is regulated by phosphorylation and protein-protein interactions. In the current paradigm, phosphorylation at Tyr 530 by c-terminal Src kinase (CSK) plays an inhibitory role, while Tyr 419 phosphorylation on the activation loop plays an activating one, although neither of these sites exert full positive or negative regulatory control[17,18]. In a cellular context, dimerization appears to substantially enhance autophosphorylation and the phosphorylation of selected substrates, while interfering with the dimerization precludes catalytic function[19]. Structural studies have revealed that c-Src is held in an autoinhibited monomeric state by intramolecular interactions between the CSK-phosphorylated c-terminal Tyr 530[20] and the SH2 domain. The SH3 domain engages a proline-rich linker between the SH2 and kinase domain N-lobe, where the active site is disrupted by displacement of the αC-helix[16]. These intramolecular connections can be displaced by intermolecular interactions between the SH2 and SH3 domains with higher affinity ligands[21] or by de-phosphorylation of Tyr 530 by a number of phosphatases[22,23]. Autophosphorylation on Tyr 419 stabilizes the activation loop into an active conformation favorable for substrate binding[24,25] and intermolecular phosphorylation[26–28]. However, the precise role of Tyr 419 autophosphorylation in the non-catalytic function and signaling is paradoxically not yet fully understood. In the same line, few studies to date have reported c-terminal Tyr 530 autophosphorylation[29]. These few previous studies lack overall information about the dynamic and temporal regulation as well as the structural determinants driving the c-terminal autophosphorylation. Furthermore, the structure, function and regulation of a hyper-phosphorylated form of c-Src and its implications in cancer as an oncogene are elusive. The dissection of the intra- versus the inter-molecular components for the mechanism of autophosphorylation, as well as the dissection of the catalytic versus the non-catalytic activities of this fundamental process, are key questions in the protein kinase field that are yet to be fully understood.

SFKs expression and activity are upregulated in solid tumors and some hematologic malignancies, often correlating with progressive stages of the diseases[30]. However, c-Src activating mutations or amplifications are rare in human cancers[31,32], with the exception of colorectal cancer[33]. Constitutive c-Src activation in human cancer cells appears to be driven at least partially by secondary alterations in upstream activators. c-Src is activated by a plethora of receptor tyrosine kinases (RTKs), including epidermal growth factor receptor (EGFR), fibroblast growth factor receptor (FGFR) and insulin-like growth factor-1 receptor (IGF1R), among others[34]. Gain of function mutations and/or overexpression of these RTKs cooperate and

synergize with c-Src activation to promote tumorigenesis probably by releasing c-Src from a closed autoinhibited state[35]. On the other hand, CSK-mediated phosphorylation of c-terminal Tyr 530 inactivates c-Src only when the protein is not previously autophosphorylated[36], which indicates the existence of intrinsically coordinated and sequential autophosphorylation events coupled to different conformation and functional states. These data also suggest that perturbation of autophosphorylation could account for the paradoxically high levels of dually-phosphorylated c-Src found in malignant cells[36]. Alternatively, c-Src could function independently of CSK, and the latter might play other roles rather than regulating the former (see "Discussion"). Here we provide functional and structural evidence related to the identification of de facto c-Src autophosphorylation sites and how "canonical" activating and repressive tyrosine residues are actually playing other important roles and functions not previously envisioned. Our data dissect a sequential and coordinated cis-to-trans phosphorylation switch connecting the activation and c-terminal segments that control c-Src enzymatic specificity and non-catalytic functions as a substrate. Evaluation of cancer-related c-terminal variants provides evidence of the existence of a complex allosteric node at the c-terminus of c-Src controlling the crosstalk between substrate- and enzyme-acting kinases, and how perturbation of this allosteric phospho-switch may drive c-Src dysfunction in cancer.

## Results

### A tripartite expression system for heterologous production of unphosphorylated c-Src in bacteria

Biochemical characterization of recombinant protein kinases usually requires low amounts (nano-to-micrograms) of material. On the contrary, biophysical and X-ray crystallography studies demand milligram amounts of homogenous protein of high stability and purity. We have applied a tripartite expression system for the heterologous expression of unphosphorylated c-Src in bacteria. The tripartite system allows the quick production and purification of high yields of active and monodisperse c-Src by co-expression with the YopH tyrosine phosphatase and the GroEL/GroES chaperones. Our procedure is an updated version of a previously established protocol[37] where high-yield production of active and soluble c-Src KD required the co-expression of YopH phosphatase. One of the main caveats of this preliminary bi-partite system was that 90% of the protein was still found in the insoluble fraction. The addition of the GroEL chaperone into the system increased solubility by 5- to 10-fold (Fig. 1a). First, we expressed a three-domain (3D, SH3-SH2-KD) construct (aa 84–536) in BL21 cells (see "Methods") and performed tandem immobilized metal affinity (Fig. 1a), cation exchange (Fig. 1b) and size-exclusion (Fig. 1c) chromatography, in order to separate each individual component to obtain pure and monodisperse c-Src. We applied multi-angle light scattering (MALS) for the determination of absolute molar mass in solution and found that both the unphosphorylated and phosphorylated forms of c-Src are active monomers in solution (Fig. 1d and Supplementary Fig. 1a). In line with this, phosphorylation does not cause any significant effect on the thermal stability of the protein as indicated by differential scanning fluorimetry (DSF) either (Fig. 1e). These data however do not exclude the fact that the intrinsically disordered N-terminal region lacking in the 3D-construct could be implicated in the dimerization process[19]. To test this hypothesis, we expressed and purified a full-length construct of the *Drosophila* c-Src homolog Src42A (see "Methods"). We found again that the full-length protein is active (work in preparation) and monomeric (60 kDa) in solution and that autophosphorylation does not induce direct dimerization (Supplementary Fig. 1a, b). In light of these results, we explored further the potential role of myristoylation on c-Src function and stability by using native and G2-myristoylated peptides in activity and binding (DSF) assays. Enzymatic assays performed with c-Src 3D and KD constructs demonstrated that preincubation of the enzyme with peptide (1:2.5

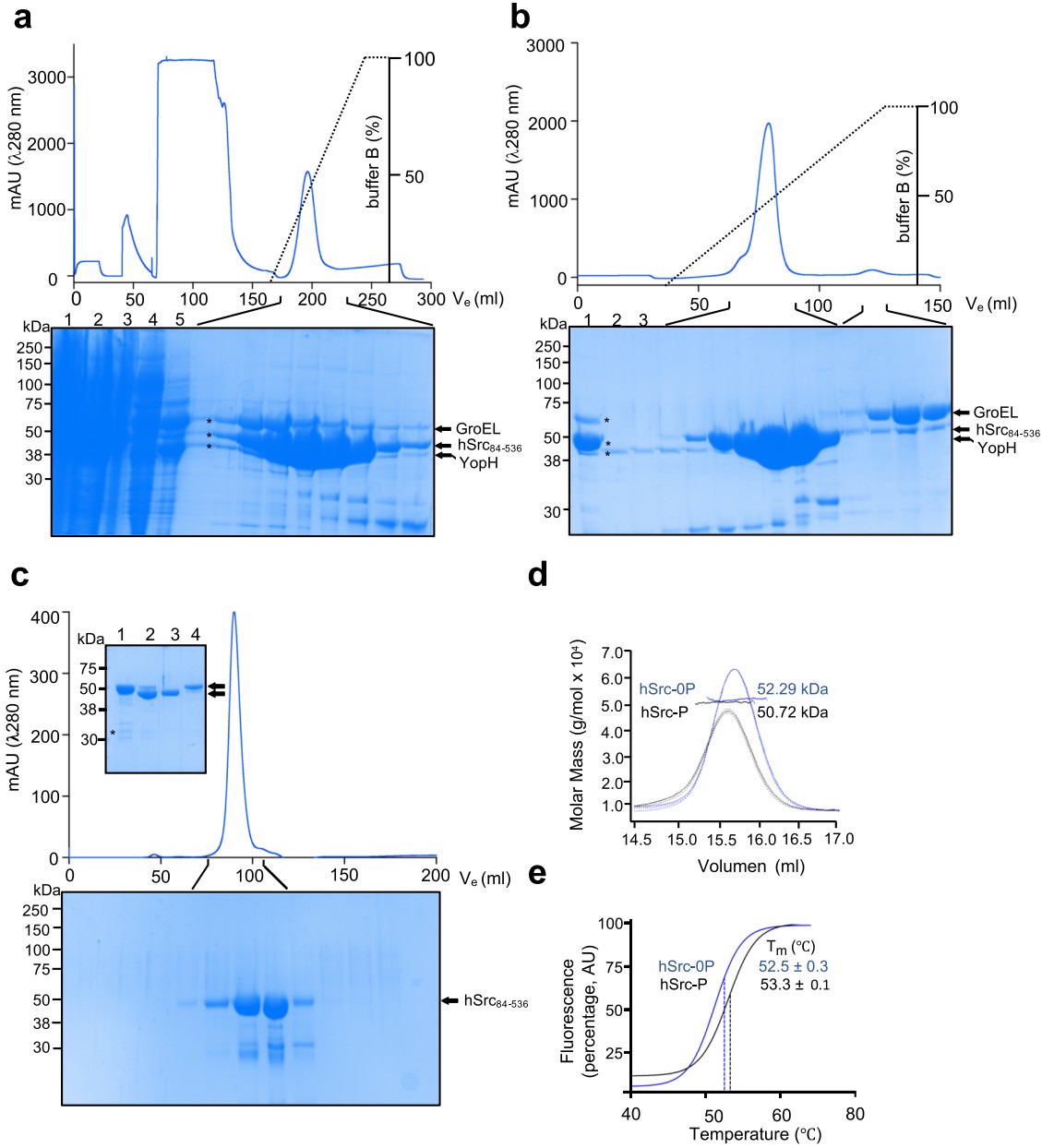

**Fig. 1 | A tripartite expression system for heterologous production of unphosphorylated c-Src in bacteria. a** IMAC chromatogram using a HisTrap column (5 ml). Indicated fractions were run on an SDS-PAGE and stained with Coomassie: 1 crude lysate, 2 insoluble fraction, 3 clear lysate, 4 wash, 5 flow-through. **b** IEC chromatogram using a HiTrap Q HP column (5 ml). Indicated fractions were run on an SDS-PAGE and stained with Coomassie: 1 pulled fractions from (**a**), 2 wash and 3 flow-through. **c** Inset, HisTrap-reverse step. Indicated fractions were run on an SDS-PAGE and stained with Coomassie: (1) input from (**b**), i.e., pull of c-Src positive fractions from IEC, (2) pull of c-Src positive fractions after protease treatment, (3) flow-through and (4) elution with 500 mM imidazole. SEC chromatogram using a Superdex 200 16/60 column. Indicated fractions were run on an SDS-PAGE and stained with Coomassie. Data from (**a**–**c**) are representative of multiple ($n \geq 10$) purification experiments. **d** SEC-MALS chromatogram showing UV ($\lambda$ 280 nm) solid line, light scattering (dashed line), Rayleigh index (dashed soft line) and molar mass (g/mol × 10⁴) dots of unphosphorylated c-Src (0-P, blue) and phosphorylated c-Src (P, 90 min, black). **e** DSF profile showing the melting temperature of unphosphorylated c-Src (0-P) and phosphorylated c-Src (P, 90 min). Inset, data is the mean ± SEM of 3 experiments ($n = 3$). Source data are provided as a Source Data file.

molar ratio, 45 min) does not result in significant changes in the catalytic rates of both constructs toward an exogenous substrate (Supplementary Fig. 1c). In DSF experiments (Supplementary Fig. 1d), none of the peptides caused a significant increment in thermal stability that would have been otherwise indicative of efficient binding. These data together demonstrate that c-Src is an active monomer in solution and that autophosphorylation does not promote direct dimerization. Myristoylation on the intrinsically disordered N-terminal region appears not to affect the dimerization nor direct interaction with the catalytic domain in vitro and lacks any regulatory effects on the catalytic activity.

## Kinetics of c-Src autophosphorylation

In the current paradigm, c-Src is held in an autoinhibited state by intramolecular interactions between the SH2 domain and CSK-phosphorylated Tyr 530 on the c-terminal segment (Fig. 2a). To define the kinetics of c-Src intrinsic autophosphorylation in the absence of CSK we used a c-Src 3D-construct and performed in vitro time-course autophosphorylation assays (0–90 min). Autophosphorylation was monitored by Western blotting (WB) using total and phospho-specific antibodies against Tyr 216, Tyr 419 and Tyr 530 (Fig. 2b). Activation-loop Tyr 419 displays fast autophosphorylation kinetics reaching saturating levels at early time points

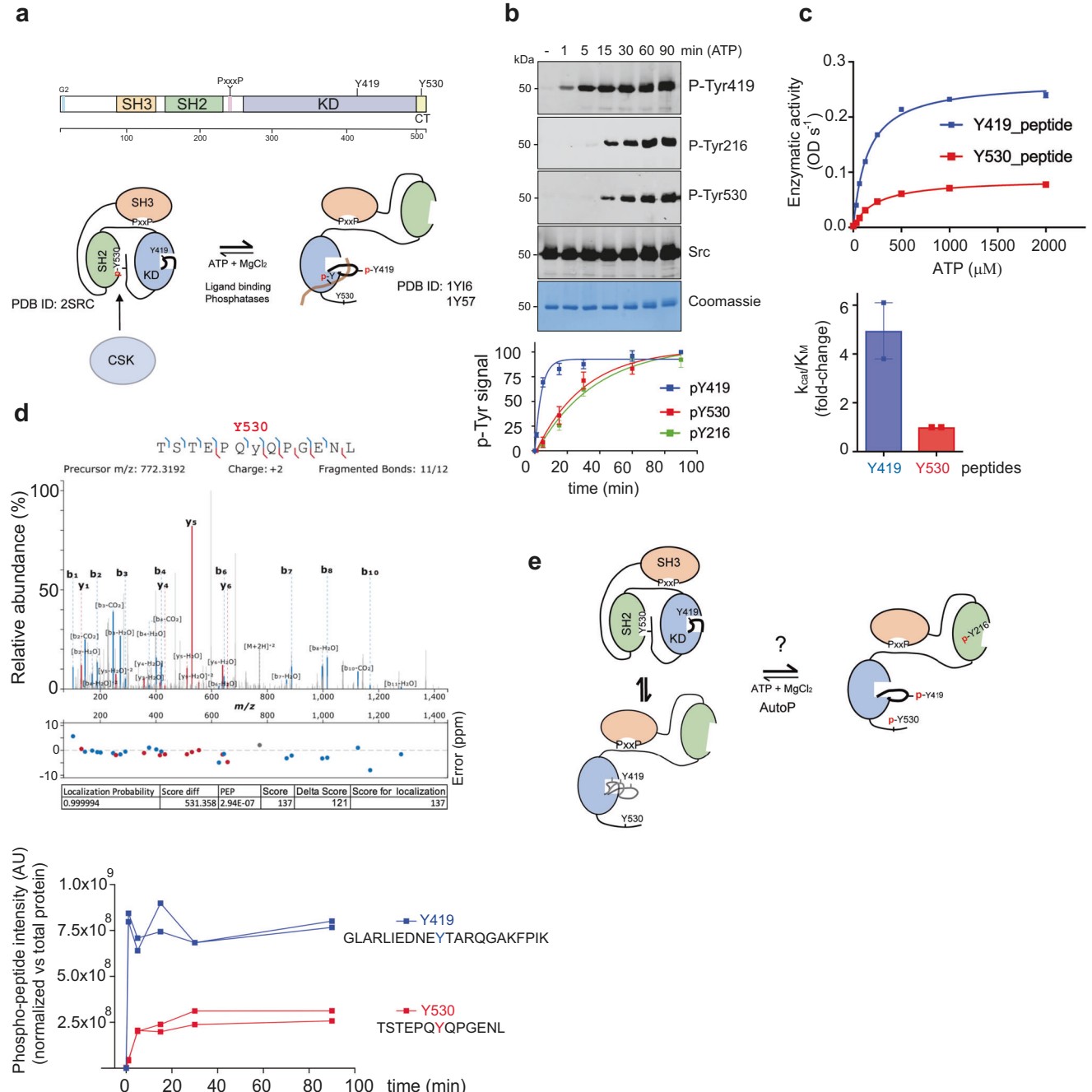

**Fig. 2 | Kinetics of c-Src autophosphorylation. a** Schematic diagram of the functional domains and autophosphorylation sites of c-Src. Representation of c-Src in closed autoinhibited and open (active) states, current paradigm for c-Src activation and regulation by CSK. **b** WB of samples from a time-course autophosphorylation experiment with c-Src WT (3D-construct, 1 μM) in the presence of ATP (1 mM) and $MgCl_2$ (2 mM) for 0–90 min using the indicated antibodies. The total amount of protein was visualized by Coomassie staining. Phospho-tyrosine quantification (total phospho-signal, percentage), data represent mean ± SEM of 3 experiments ($n = 3$). **c** Enzymatic assay performed with c-Src WT (3D-construct, 1 μM) incubated with increasing concentrations of ATP at a fixed concentration (1.5 mg/ml) of c-Src Tyr 419 (IEDNEYTARQG) or Tyr 530 (STEPQYQPGEN) derived peptides. Data represent the mean ± SEM of two experiments ($n = 2$). Enzymatic activity (ODs$^{-1}$ × 10$^{-3}$). Catalytic efficiency constants ($k_{cat}/K_M$, fold difference) are depicted in the panel below. **d** MS2 mass spectrum of the [M+2H] +2 ion (m/z 772.3) reporter for a phosphorylated peptide on Tyr 530 (90 min). Below, phosphorylation kinetics of Tyr 419 and Tyr 530 phosphopeptides measured by mass spectrometry (0–90 min) are depicted. Data shown are the mean of two replicates of a single experiment. Data is representative of four independent experiments. **e** Representation of c-Src hypothetical apo (inactive) states and tri-phosphorylated c-Src resulting in an extended (active) state. Source data are provided in the Source Data file.

(1 to 5 min). In contrast, c-terminal Tyr 530 and SH2 Tyr 216 show slower rates of autophosphorylation, reaching maximum levels at later time points (30 min). Interestingly, autophosphorylated Tyr 216 on the SH2 domain can coordinate with the side chain of R208 (see PDB ID: 2SRC, https://www.rcsb.org/structure/2src) and contribute to the liberation of the c-terminal Tyr 530, in line with a

model where a tri-phosphorylated c-Src molecule would adopt an extended conformation (see Fig. 2e). Enzymatic assays using peptides derived from activation loop and c-terminal phospho-sites (Fig. 2c) corroborated the WB data, with a five-fold increase in the catalytic efficiency constant ($k_{cat}/K_M$) by the Tyr 419-derived peptide over Tyr 530. These data highlight two important observations:

(1) the existence of sequential and coordinated autophosphorylation events coupled to different conformation states, and (2) c-terminal Tyr 530 is a de facto c-Src autophosphorylation site. Considering these two premises, the activation loop of c-Src must be readily accessible to be phosphorylated first, while slower c-terminal Tyr 530 becomes accessible for autophosphorylation at later stages due to restricted access. These data corroborated that in solution, unphosphorylated c-Src 3D is adopting a closed state (Supplementary Fig. 2 and Supplementary Table 1). Next, we used time-resolved high-resolution liquid chromatography-tandem mass spectrometry (LC-MS/MS) to monitor simultaneously multiple autophosphorylation sites. We identified 25 autophosphorylation sites, among them some unexpected serine and threonine residues as well as tyrosine sites (Supplementary Fig. 3). Novel autophosphorylation sites identified and validated in at least two independent experiments were Tyr 95 (βa/βb linker, SH3), Tyr 152 (βA, SH2), Thr 288 (β2), Tyr 385 (αE/β7 linker) and T443 (αF) in the kinase domain[16]. MS data on those best-characterized phospho-sites validate our WB data using phospho-specific antibodies (Fig. 2c).

### c-Src activation-loop Tyr 419 controls enzyme specificity

To evaluate in vitro the activating and inhibitory roles previously described in a cellular context for c-Src activation-loop Tyr 419 and c-terminal Tyr 530, respectively[17,18], we generated tyrosine to phenylalanine mutants in a c-Src 3D-construct and performed in vitro time-course autophosphorylation assays (Fig. 3a). In these experiments we observed a significant degree of phospho-tyrosine activity by a c-Src activation loop mutant Y419F, which was consistently lower compared with the WT. Remarkably, this mutant lacks the capacity to autophosphorylate on Tyr 530 at the c-tail (Fig. 3a). Conversely, a c-Src Y530F mutant displays higher phospho-tyrosine activity, which is also reflected in faster kinetics of Tyr 419 and Tyr 216 autophosphorylation compared with the WT (Fig. 3a). These results were further confirmed by LC-MS/MS (Supplementary Fig. 4). Next, we performed enzymatic assays using a generic peptide derived from c-Abl as an exogenous substrate (Fig. 3b). Under these conditions, the three constructs efficiently phosphorylate the c-Abl peptide, with only a two-fold difference in the $V_{max}$ observed between the WT and Y419F. On the contrary, when a peptide derived from the RET activation loop was tested, there was one order of magnitude difference in the $V_{max}$ between Y419F and the WT (Fig. 3b). In these experiments, WT and c-terminal Y530F mutant displayed similar activities toward the exogenous substrate peptides. These data suggest that a non-phosphorylable c-Src activation loop mutant displays changes in the substrate preferences (i.e., specificity) rather than a catalytic impairment. To test this hypothesis, we used a series of recombinant proteins as intact substrates in an in vitro phosphorylation assay: a fragment of human FAK (aa 1–405) containing Tyr 397, c-Src KD K298M, RET KD K758M and an inactive KIF5B-RET short construct (Fig. 3c–f). All these proteins lack catalytic activity and hence act as substrate surrogates. c-Src 3D constructs WT, Y419F and Y530F were able to efficiently phosphorylate GST-FAK (aa 1–405) and c-Src KD with no significant differences among them (Fig. 3c, d). When a RET KD was used as an intact substrate surrogate, c-Src Y419F (contrary to WT or Y530) was unable to efficiently phosphorylate the activation loop on Tyr 905 (Fig. 3e). These data were further corroborated using a truncated KIF5B-RET fusion construct with a mutation in the HRD motif (Fig. 3f). Consensus phospho-sites sequences scrutiny revealed that out of the three substrates surrogates used, the RET activation loop sequence was the most similar to the optimal c-Src substrate sequence (phosphosite.org), hence from these data we conclude that c-Src activation-loop Tyr 419 controls enzyme specificity toward substrates (Fig. 3g).

### Time-resolved intra- and intermolecular components for the mechanism of c-Src autophosphorylation

In order to dissect the time-resolution of the inter- and intramolecular components for the mechanisms of c-Src Tyr 419 autophosphorylation, we performed time-course reactions with increasing protein concentration (0.25, 0.5, 1 and 2 µM) of c-Src 3D WT, Y419F and Y530F. Equal amounts of samples (see Coomassie) were loaded, and WB was performed using phospho-specific antibodies (Fig. 4a). In the case of c-Src WT, we observed a fast cis-component (protein concentration-independent) for Tyr 419 autophosphorylation at early time points, see at 1 min 0.25, 0.5 and 1 µM concentrations and at 5 min, 0.25 and 0.5 µM concentrations. This was followed by an intermolecular component (protein concentration-dependent) at later time stages see 15 min at 0.5 µM, 5 min at 1 µM and 1 min at 2 µM concentrations (Fig. 4a). Strikingly, activation loop Tyr 419 phosphorylation by a c-terminal Y530F mutant showed a very strong cis-component as indicated by very fast kinetics of Tyr 419 autophosphorylation, which were independent of enzyme concentration (8-fold increase, from 0.25 to 2 µM). Altogether these data showed that in the native state, the mechanism of c-Src activation loop autophosphorylation is defined by a fast intramolecular component that switches over time to an intermolecular process; a cis-to-trans autophosphorylation switch. Furthermore, the crosstalk between the activation loop and c-terminal segment is coordinated and coupled to different conformational and functional states as indicated by: (1) the lack of the intermolecular component by a mutant that cannot be phosphorylated on Tyr 530 and (2) an activation loop mutant that cannot phosphorylate c-terminal Tyr 530. Next, we evaluated the effect of increasing enzyme concentration (from 0.25 up to 2 µM) in the kinetics rates and constants of c-Src WT, Y419F and Y530F for ATP using a c-Abl derived peptide (Supplementary Fig. 5a, b). Src WT and Y419F showed constant catalytic efficiency over the range of enzyme concentrations used, whereas the Y530F mutant showed, contrary to the WT and Y419F, a constant $k_{cat}$ but a decrease in $K_M$ that resulted in a significant 20-fold increase in the catalytic efficiency ($k_{cat}/K_M$) constant (Supplementary Fig. 5a, b). These data demonstrate that c-Src activation loop mutant Y419F is an active enzyme and that a c-terminal Tyr 530 mutant displayed increased catalytic efficiency constant for ATP compared with WT and Y419F. These data support the results from the autophosphorylation assays, where c-Src Y530F displayed higher phospho-tyrosine activity. Note that peptide phosphorylation follows classic Michaelis-Menten kinetics, where the substrate is in large excess over the enzyme, while the cis- and trans-phosphorylation that take place with the intact protein do not represent Michaelis-Menten kinetics and are usually subjected to allosteric control.

Next, in order to discriminate the catalytic (enzyme) from non-catalytic (substrate) properties, we used first a c-Src KD K298M catalytic dead mutant as an intact substrate surrogate in phosphorylation assays (Fig. 4b). In this system, c-Src WT and Y530F efficiently phosphorylated the activation loop of the substrate. Unexpectedly, a c-Src Y419F mutant did also efficiently phosphorylate the substrate that mimics its own activation loop, however, and contrary to the WT and Y530F mutant, it did not phosphorylate c-terminal Tyr 530 in trans. These results further confirm that activation loop Tyr 419 is required for c-terminal Tyr 530 phosphorylation and that none of the two main autophosphorylation sites are required to maintain the enzymatic activity of c-Src.

### Activation-loop Tyr 419 controls c-Src non-catalytic properties

Second, to explore the non-catalytic (substrate) functions of c-Src, we performed the opposite time-course phosphorylation experiments using a c-Src KD construct as an active enzyme and a catalytic dead 3D-K298M construct with and without activation-loop Y419F or c-terminal Y530F mutations as substrate surrogates (Fig. 4c). In the case of a K298M construct, while activation-loop Tyr 419 is efficiently

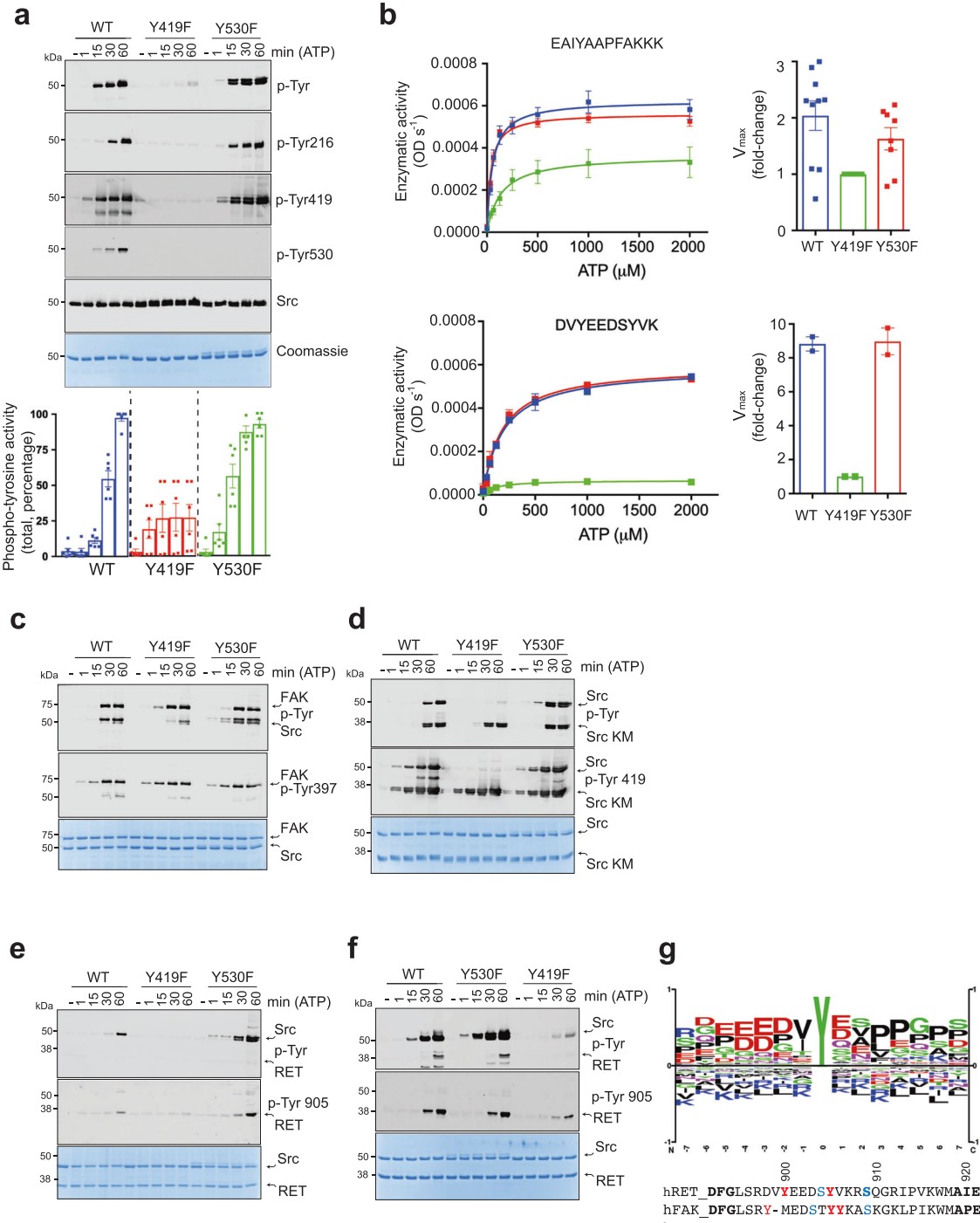

**Fig. 3 | Activation-loop Tyr 419 controls c-Src substrate specificity. a** WB of samples from a time-course autophosphorylation experiment with c-Src WT, Y419F and Y530F (3D-construct, 1 μM) in the presence of ATP (1 mM) and MgCl₂ (2 mM) for 0–60 min using the indicated antibodies. The total amount of protein was visualized by Coomassie staining. Phospho-signal quantification (total phospho-tyrosine), data represent mean ± SEM of 6 experiments (*n* = 6). **b** Enzymatic assay performed with c-Src WT, Y419F and Y530F (3D-construct, 1 μM) incubated with increasing concentrations of ATP at a fixed concentration (4 mg/ml) of peptie: ABL derived peptide (EAIYAAPFAKKK). Data shown are the mean ± SEM of *n* = 12 (WT), *n* = 10 (Y419F) and *n* = 8 (Y530F). Data is representative of 4–6 experiments in duplicate. For the RET activation loop Tyr 905 derived peptide (DVYEEDSFVK), data

shown represent the mean ± SEM, of one experiment in duplicate (*n* = 2). Catalytic efficiency constants ($k_{cat}/K_M$, fold difference) are depicted in the right panel. **c–f** WB of samples from time-course phosphorylation assays with c-Src (3D, 1 μM) WT, Y419F and Y530F and using the following substrates surrogates: FAK (aa 1–405), c-Src KD K298M, RET KD (aa 713–1012) K758M and deltaKIF5B-RET, respectively, using the indicated antibodies. Data are representative of at least 2 independent experiments. **g** Logo consensus sequence for optimal c-Src substrate, from PhosphoSitePlus (CST), and amino acid sequence alignment of human RET, FAK and c-Src activation segments. In color code, phosphorylable residues are depicted, and in bold, already known autophosphorylation sites, numbering from RET. Source data are provided in the Source Data file.

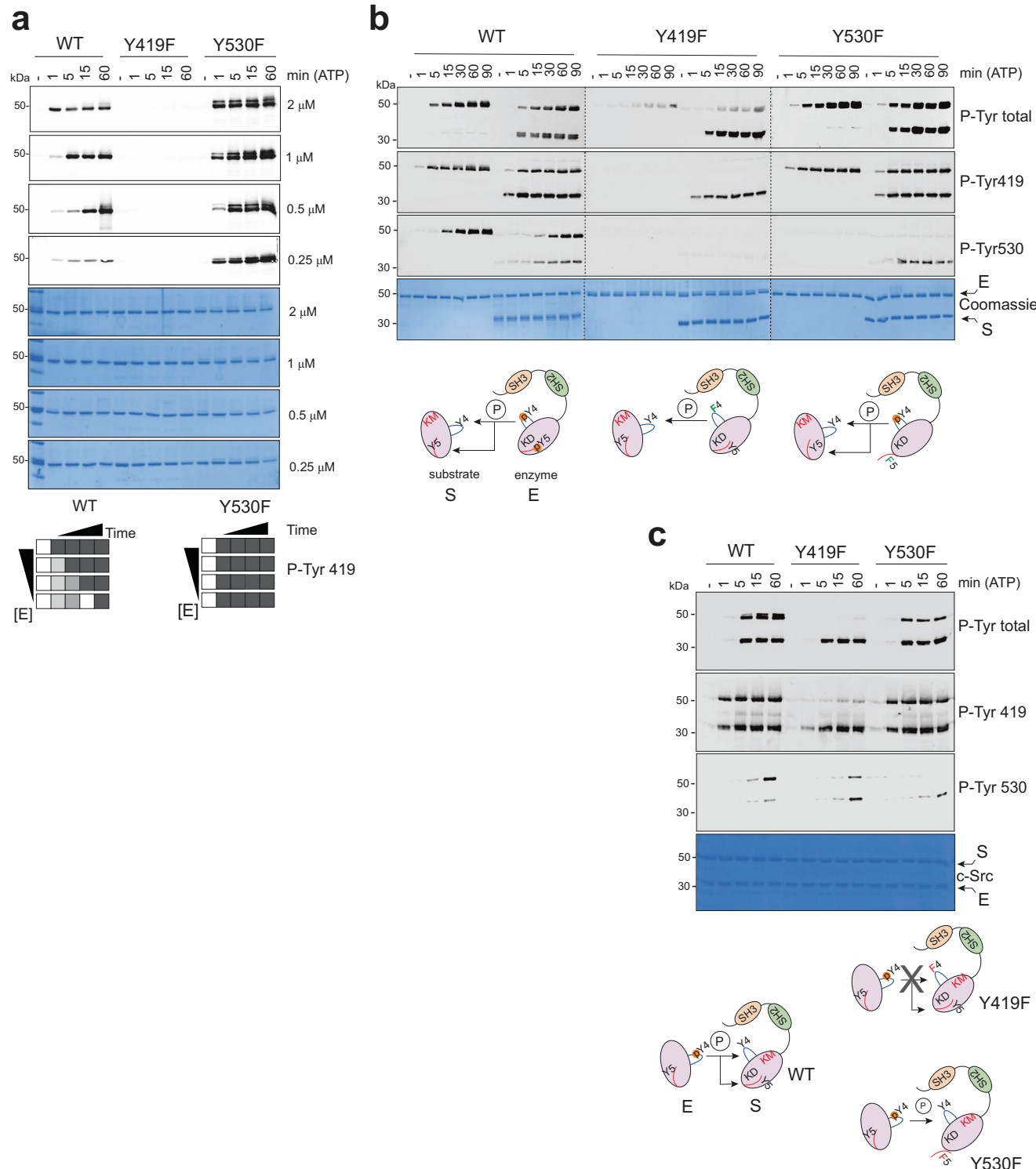

**Fig. 4 | Dissecting cis-versus-trans components for c-Src autophosphorylation.**
**a** WB of samples from a time-course autophosphorylation experiment with WT, Y419F and Y530F c-Src (3D-construct, 0.25–2.5 µM) in the presence of ATP (1 mM) and MgCl₂ (2 mM) for 0–60 min using the indicated c-Src phospho-Tyr 419 antibody. The total amount of protein was visualized by Coomassie staining. **b** WB of samples from a time-course phosphorylation assay (0–90 min) as in (**a**), in the absence and in the presence of a c-Src KD K298M construct as an intact substrate surrogate using the indicated antibodies. The total amount of protein was

visualized by Coomassie staining. Diagram for the enzyme (E) and substrate (S) acting kinases. lower inset. **c** WB of samples from a time-course experiment with c-Src KD WT in the presence of c-Src 3D-K298M constructs with and without Y419F and Y530F mutations as substrates using the indicated antibodies. Total c-Src protein was visualized by Coomassie staining. Diagram for the enzyme (E) and substrate (S) acting kinases. lower inset. Data from (**a**–**c**) are representative of at least 3 independent experiments. Source data are provided in the Source Data file.

phosphorylated by the enzyme-acting kinase, c-terminal Tyr 530 appears to be a less efficient phosphorylable substrate. These data come in line with the preferences observed in the dynamics of autophosphorylation and enzyme kinetics data using phospho-specific antibodies and peptides derived from each consensus phospho-site (Fig. 2b, c). Surprisingly, a substrate surrogate with a Y419F mutation at the activation loop behaves overall as a very poor substrate, showing neglected phosphorylation promoted by the enzyme-acting kinase (Fig. 4c). A substrate with a non-phosphorylable c-terminal Tyr 530 is efficiently phosphorylated by an active kinase molecule, however to a lower level (2-fold) compared with the native substrate (Fig. 4c). These results demonstrate that Tyr 419 controls, in addition to substrate specificity, important substrate-like non-catalytic properties by means of presenting the activation loop to another active molecule for intermolecular phosphorylation.

## In vivo evaluation of an activation loop mutant of c-Src in *Drosophila*

To evaluate the functionality of an activation loop mutant in an in vivo system, we used the fruit fly *Drosophila melanogaster*. We generated transgenic lines expressing the Src42A Y400F mutation (equivalent to Tyr 419 in human c-Src) in different tissues (Fig. 5). First, we analyzed the phenotypic effects produced by the expression of Src42A Y400F in adult tissues (Fig. 5a). The overexpression of a Src42A WT allele in the eye by means of the Gal4/UAS system using the GMR promoter produced a strong phenotype in which flies virtually lack all ommatidia (Fig. 5a), as previously described[38–40]. The overexpression of Src42A Y400F in the eye also produced a clear and penetrant rough eye phenotype (Fig. 5a). In the same line, overexpression of Src42A Y400F in the wing using a SalEPV-Gal4 driver resulted in a penetrant phenotype of rudimentary wings, similar to that of Src42A WT (Fig. 5b). This result indicated that Src42A Y400F is a functional and active protein. We noticed that the defects produced by Src42A Y400F were weaker than those produced by Src42A WT. This may be due to differences in the expression levels, or it may indicate that the Src42A Y400F mutant protein is less active than the WT, lacking some specific functional facet.

Src42A has been shown to play a key role during the embryonic development of the tracheal (respiratory) system. In particular, Src42A is required for the adequate orientation of the membrane growth along the longitudinal axis of the tracheal tubes regulating in this way their size[41,42]. Next, to investigate the phenotypic impact of Src42A Y400F in the developmental trachea, we applied the Gal4/UAS system with the breathless (btl) promoter (Fig. 5c) in an intact genetic background. While the lack of Src42A activity (either using a SrcF80 mutant or a kinase-dead form of Src42A that acts as a dominant negative, Src42AKM) leads to shorter tubes, expression of Src42A WT leads to a moderate elongation of the tracheal tubes in embryos[41,42], see Fig. 5e. In contrast, we found that the expression of Src42A Y400F resulted in a lack of significant elongation (or reduction) of the tube size (Fig. 5c, e). These data suggest that an activation loop Y400F mutant Src42A protein, although functionally competent in other tissues, is less active in the trachea. Alternatively, it may indicate that the mutated protein is not functional in the trachea. To discriminate between these two possibilities, we expressed Src42A Y400F in a loss of function genetic background for Src using *Src^F80* flies[41]. In this experimental setting, we observed that Src42A Y400F is able to fully rescue the tracheal defects seen in *Src^F80* flies, reverting the defects of short tubes (Fig. 5d, e). It was previously shown that Src42A WT[41] fully rescues the tube elongation defects of *Src^F80* while the kinase-dead Src42A KM does not. These results suggest that Src42A Y400F is a functional and active protein capable of rescuing the phenotypical defects promoted by a loss of function Src42A allele in the trachea as a WT protein.

## Crystal structure of c-Src KD in complex with Ponatinib

We solved the crystal structure of a c-Src KD construct (aa 252–536) in complex with Ponatinib at 2.49 Å resolution (Fig. 6a). The crystal belonged to the orthorhombic system, and data was processed using P1 21 1 symmetry (Table 1). The crystal had two molecules of c-Src in the asymmetric unit, and in both chains, the electron density for Ponatinib was perfectly defined with full occupancy. The central part of the activation loop (aa 410–425) was disordered and lacked electron density (Fig. 6a, b). Ponatinib binds to the active site in an extended pose reaching the back and allosteric pockets causing the activation loop to adopt a DFG-out conformation (Fig. 6a–c). This configuration was also seen in the crystal structures of c-Src, c-Abl and c-Kit in complex with Imatinib[43–46]. The imidazopyridazine core of Ponatinib rested on the adenine pocket of the enzyme and formed hydrophobic interactions with residues located within and at the vicinity of the hinge L276 (β1), L396 (β6) and Y343 (hinge), forming one hydrogen bond with main chain nitrogen of M344 (hinge). The ethynyl linkage forms favorable hydrophobic interactions with L396 (β6). The methylphenyl group occupies a hydrophobic pocket formed by T341 (β4), K298 (β3) and I339 (β4), with the oxygen atom from the methyl group forming an additional hydrogen bond with the catalytic K298 (β3). The trifluoromethyl group forms hydrophobic interactions with V405 (A-loop) and also with L325 (αC-β4 loop), while the phenyl group forms hydrophobic contacts with D407 (DFG motif), E313 (αC) and M317 (αC). The methyl piperazine group formed an electrostatic interaction with D407 (A-loop) and one hydrogen bond with R388 (HRD motif), inducing altogether a DFG-out conformation (Fig. 6b, c, g). The DFG-out conformation is not compatible with the proper alignment of the regulatory R-spine[47], which is broken by the flipping of the DFG motif, which is a signature of inactive activation loop conformation (Fig. 6d). Note that the αC-β3 catalytic salt bridge is still formed and is not broken by the inhibitor; however, the glutamate of the DFG motif that is part of the catalytic triad is not engaged due to the breaking of the R-spine by the compound. We evaluated functionally a D407A mutant, which showed, as expected, a diminished phospho-tyrosine activity (Fig. 6e) and a detrimental effect in the binding of Ponatinib as indicated by DSF assays (Fig. 6f).

## A crystallographic snapshot of c-terminal Tyr 530 intermolecular phosphorylation

Another interesting feature of the crystal structure presented in this study is the arrangement of the two molecules of c-Src found in the asymmetric unit, where the back side of the c-lobe of one protomer faced the substrate binding area located at the base of the active site cleft of the other (Fig. 7a, b). The interface is composed of the αI-I'-helix (residues 512–526) of the substrate molecule facing toward the αD (residues 349–354) and αF-αG loop (residues 460–471) of the enzyme-acting molecule with a total surface area of ~1900 Å². A close-up of the interface showed that two acidic residues, D521 and E520 (αI), form a salt bridge with R463 (αF-αG loop). D521 main chain oxygen makes polar contacts with T524 and S525 (αI') main chain nitrogen atoms. T526 and S525 (αI') coordinated with K354 (αD) via weak hydrogen bonding (4 Å). This network of hydrogen bonds and electrostatic interactions was further complemented by the coordination of Q278 and G279 (G-loop) main chain atoms in the enzyme-acting molecule with the side chain of Q529 at the c-terminal on the substrate kinase (Fig. 7b). In this scenario, both molecules adopted a disposition such that we can visualize the c-Src c-terminal Tyr 530 of the substrate molecule positioned for ready entry into the active site of the enzyme to undergo intermolecular phosphorylation. This disposition is similar to some extent to the one observed in the crystal structure of the Csk/c-Src complex, where the interface was formed by the c-terminal αI helix with the preceding αH-αI loop of c-Src and the αD helix of Csk at the entrance of the active site[20]. The main difference, however, between the

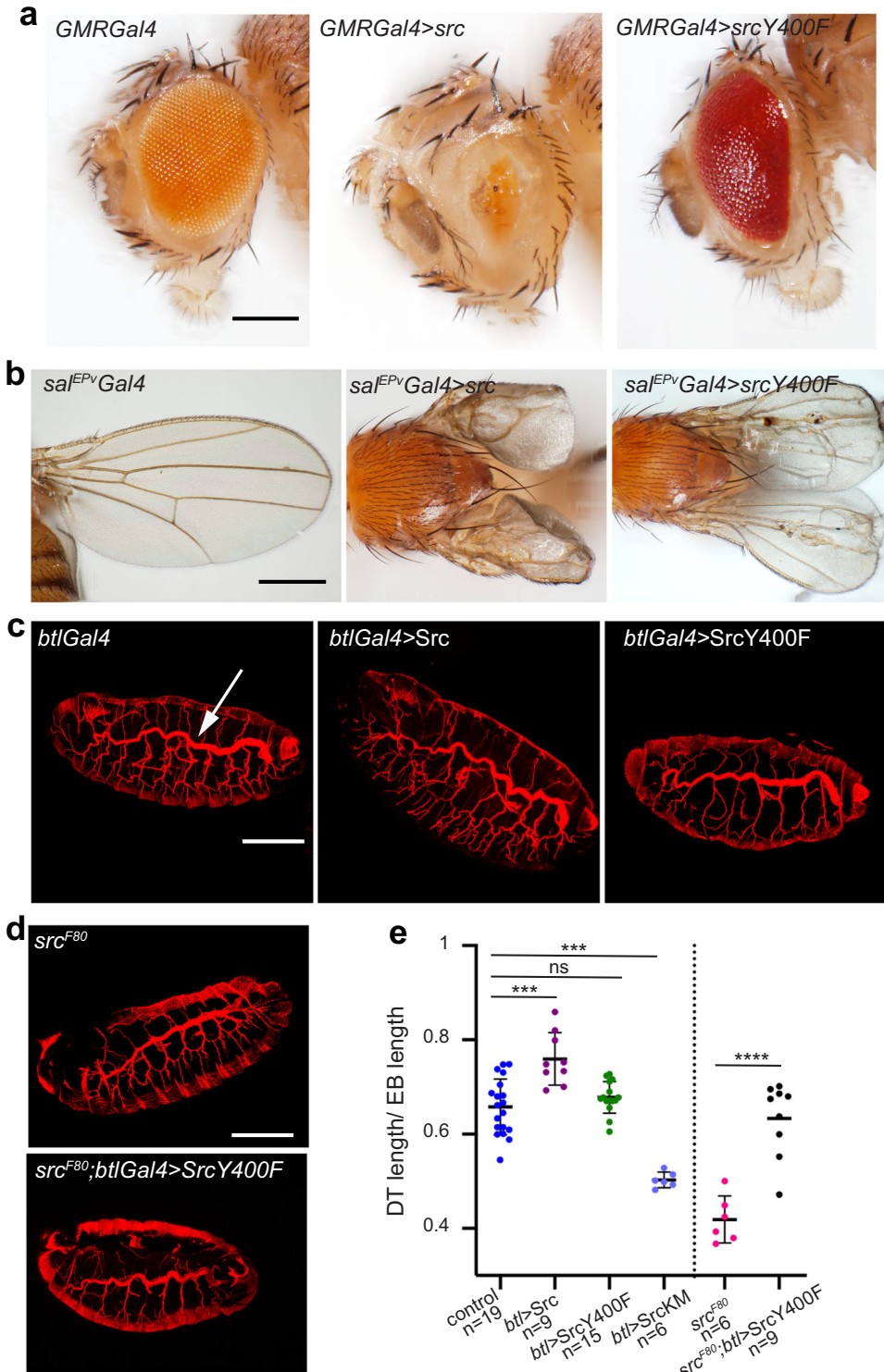

**Fig. 5 | In vivo evaluation of an activation loop mutant of c-Src in *Drosophila*.**
**a** Expression Src42A WT and Y400F mutant in the *Drosophila* eye using a GMR-Gal4 promoter. Representative images of eyes from GMR-Gal4 (control), GMR-Gal4-Src42A WT and GMR-Gal4-Src42A Y400F flies. Scale bar, 500 μm. **b** Expression of Src42A WT and Y400F in the *Drosophila* wing using a salEPv-Gal4 promoter. Representative images of wings from salEPv-Gal4 (control), salEPv-Gal4-Src42A and salEPv-Gal4-Src42A Y400F flies. Scale bar, 500 μm. For (**a**) and (**b**), data shown are representative of ≥10 images. **c** Expression of Src42A WT and Src42A Y400F in the developing *Drosophila* tracheal tube using the btl-Gal4 promoter. Images show projections of confocal sections of lateral views of stage 16 embryos stained for CBP to visualize the tracheal tubes. The length of the DT (arrow in control) is quantified

in (**e**). Scale bar, 100 μm. **d** Rescue experiments to assay Src42A Y400F function. Projections of confocal sections of lateral views of stage 16 embryos stained for CBP to visualize the tracheal tubes. Src mutant embryos (SrcF80 and SrcKM) show reduced DT length. Scale bar, 100 μm. Expression of Src42A Y400F in the trachea of SrcF80 mutants restores DT length (quantifications are shown in (**e**)), control (*n* = 19), *btl*>Src (*n* = 9), *btl*>Src Y400F (*n* = 15), *btl*>SrcKM (*n* = 6), Src*F80* (*n* = 6), Src*F80*; *btl*>Src Y400F (*n* = 9). Unpaired two-tailed Student's *t*-test applying Welch's correction. Differences were considered significant when *p* < 0.05. * *p* < 0.05, ** *p* < 0.01, *** *p* < 0.001, **** *p* < 0.0001 where n.s. means not statistically significant. Source data are provided in the Source Data file.

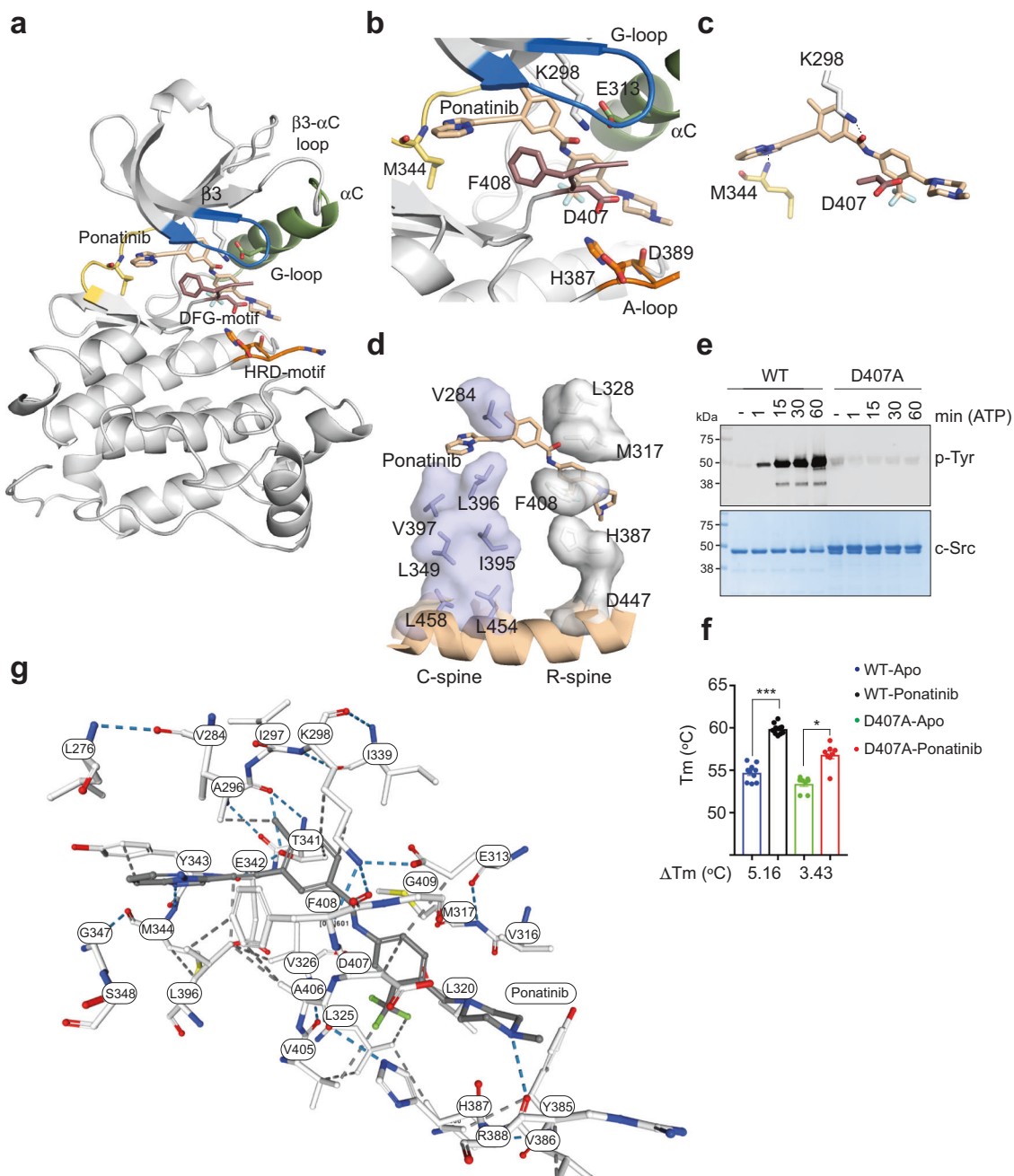

**Fig. 6 | Crystal structure of c-Src KD in complex with Ponatinib. a** Cartoon representation of c-Src KD crystal structure bound to Ponatinib. Selected residues and secondary structure elements are color-coded: G-loop (blue), αC-helix (green), hinge (yellow), HRD motif (orange) and activation segment residues (brown). **b, c** Close-up view of the active site with Ponatinib binding pose showing relevant c-Src interacting residues. **d** Cartoon of c-Src catalytic (C−) and regulatory (R−) spines showing side-chain residues and surface representation and how Ponatinib binding breaks the linear arrangement of the R-spine by flipping F408 from the DFG motif. **e** WB of samples from time-course autophosphorylation experiment with c-Src WT and D407A (3D-construct 2 μM) in the presence of ATP (1 mM) and MgCl₂ (2 mM) for 0−60 min using a total phospho-tyrosine antibody. Total c-Src protein was evaluated by Coomassie staining. Data are representative of 2 independent experiments. **f** DSF data of c-Src 3D-construct (WT and D407A) in apo and Ponatinib-complexed states showing the thermal shift (°C). Data represent the mean of the melting temperature ($T_m$) for each condition ± SEM out of 4 independent experiments in duplicate (*n* = 8). Statistics: *** *p* < 0.001, * *p* = 0.0329, one-way ANOVA multiple comparison Kruskal–Wallis test. **g** Detailed ligand view interactions showing hydrophobic contacts (gray dashed line) and hydrogen bonds (blue dashed line) of all residues from the active site of c-Src bound to ponatinib as from the NGL viewer and RCBS PDB ID: 7OTE. Source data are provided in the Source Data file.

two structures is that in the CSK/c-Src complex, the c-terminal segment of the c-Src substrate molecule is not in contact with the G-loop of CSK (see next "Results" section). In both cases, the structures are compatible with the c-terminal Tyr 530 of the substrate molecule to be positioned at the entrance of the active site to undergo phosphorylation by the active kinase. Superimposition with structures of protein kinases in complex with different peptide substrates: PKA (PDB 1ATP), PKB (PDB 1O6K, 1O6L), PhK (PDB 2PHK) and IRK (PDB 1RK3) revealed that tyrosine and serine/threonine-specific substrates follow different paths along the substrate binding cleft, and that the c-terminal fragment of c-Src including Tyr 530 could readily accommodate onto those paths being compatible with substrate binding and phosphorylation (Fig. 7c).

**Table 1 | Data collection and refinement statistics**

|  | Src KD-ponatinib |
|---|---|
| PDB code | 7OTE |
| Space group | P 1 21 1 |
| Cell dimensions |  |
| A, b, c, Å | 42.15, 124.63, 63.59 |
| A, β, γ | 90.00°, 90.15°, 90.00° |
| Resolution (outer resolution shell), Å | 63.59–2.49 (2.55–2.49) |
| $R_{sym}$ (%) | 19.0 (94.6) |
| $R_{p.i.m}$ (%) | 8.4 (43.0) |
| I/δ | 7.1 (2.6) |
| Completeness (%) | 93.59 (86.67) |
| Redundancy | 4.7 (4.6) |
| No. of unique reflections | 11,062 |
| $R_{work}$ | 0.175 (0.189) |
| $R_{free}$[a] | 0.252 (0.256) |
| Total number of atoms | 4388 |
| Wilson B factor | 36.7 |
| Average isotropic B factors, A$^2$ | 44.0 |
| R.M.S.D. |  |
| Bonds, Å | 0.012 |
| Angles, ° | 1.757 |
| Ramachandran plot (%) (favored/allowed/disallowed) | 92.75/5.88/1.37 |

[a]A total of 5.4% of the data was set aside to compute $R_{free}$.

To confirm that Tyr 530 autophosphorylation was driven by an intermolecular component we performed time-course autophosphorylation assays at increasing enzyme concentrations (from 0.25 to 2 µM) see Fig. 7d. We observed a concentration-dependent effect, as indicated by WB using a phospho-specific antibody. Together our structural and biochemical data demonstrate that the c-terminal Tyr 530 is a de facto c-Src autophosphorylation site driven by a strong intermolecular component.

Furthermore, we probed the crystallographic snapshot of the asymmetric c-Src dimer using unbiased, all-atom molecular dynamics (MD) simulations. In order to simulate a more biologically relevant system, Ponatinib was replaced with ATP and 2 magnesium ions for both monomers. Our simulations capture the c-tail of the substrate molecule moving into the active site cleft of the enzyme molecule where Tyr 530 is stably localized below a distance of 10 Å from the γ-phosphate group of the ATP molecule (Fig. 7e). These data support our model of Tyr 530 intermolecular autophosphorylation. We also note that the asymmetric dimer conformation is not very stable, the orientation of the two molecules is highly dynamic, and it is maintained by a key interaction between R463 on the enzyme molecule which forms a salt bridge with two acidic residues E520 and D521 on the substrate molecule (Fig. 7e). We tested functionally a D521A and 529X mutant constructs and found a detrimental effect on Tyr 530 autophosphorylation by the D521A mutant without having a significant impact on the overall phospho-tyrosine activity (Supplementary Fig. 7a y b). In total, we performed eight replicate MD simulations lasting 1200 ns each. Based on the analysis of these trajectories, we find that the asymmetric dimer conformation, mediated by the R463 salt bridge, is loosely maintained in six replicates, while the conformation is immediately dissociated in two replicates (Supplementary Fig. 6). Out of the six replicates which maintain the asymmetric dimer, half of these capture the substrate C-tail moving into the enzyme active site where Y530 is stably localized below a distance of 10 Å from the γ-phosphate group of ATP molecule for at least 100 ns (Fig. 7e and

Supplementary Fig. 6). We note that the c-terminal tail of both the enzyme and substrate molecule is highly dynamic in nearly all replicate simulations. Overall, our simulations support the formation of a transient asymmetric dimer that precedes Tyr 530 trans-autophosphorylation.

**A c-terminal palindromic phospho-motif controls allosterically c-Src function.** In the crystal structure, we identified a palindromic motif flanking the c-terminal Tyr 530 (PQYQP, aa 528–532) on the substrate molecule making contacts with the active kinase (Fig. 8a). In particular, the side chain of Q529 from the c-tail of the substrate molecule is coordinated via hydrogen bonds with Q278 and G279 main chain oxygens of the G-loop in the kinase molecule (Fig. 8a). These intermolecular contacts were complemented by an intra-molecular network of interacting residues, where the c-terminal N535 (c-tail) main chain nitrogen coordinated with side chain of E489 (αG-αH loop) and the side chain of R362 (αE) formed polar contacts with main chain oxygen of G533 (Fig. 8a). These interactions suggested that a perturbation of the palindromic phospho-motif by c-terminal cancer-associated variants could affect the kinase-substrate interface and hence perturb the c-terminal inter-molecular phosphorylation. To test this premise, we performed a time-course autophosphorylation assay using a v-Src 3D surrogate construct lacking the complete c-terminal segment from residue 528 to 536[1] and a colorectal cancer CRC-truncated mutant at codon 531 missing aa 531–536[33] (see Fig. 8b). Out of the two truncated forms, the c-terminal mutant at codon 531 had a detrimental effect on the activity as indicated by WBs using a total phospho-tyrosine antibody (Fig. 8b). Unexpectedly, the c-Src 531X mutant, despite having a phosphorylable c-terminal Tyr 530, lacked also the capacity to autophosphorylate on this residue, meaning that residues 531 to 536 are required for c-terminal autophosphorylation.

These data however did not recapitulate the catalytic rates for ATP observed by these variants (Fig. 8c), which was indicative of an otherwise intact catalytic activity. Such detrimental effect was not caused either by the perturbation of the consensus phospho-motif; as no difference was found by both c-Src WT and 531X constructs in their capacity to phosphorylate a peptide derived from an intact c-terminal sequence (STEPQYQPGEN) or a c-terminal deleted peptide (STEPQY) (Supplementary Fig. 8). In order to evaluate the catalytic function of the c-terminal variants we used intact substrate surrogates (i.e., KD K298M constructs) in time-course phosphorylation experiments (Fig. 8d). Analysis of the results reveals that both c-terminal variants display comparable catalytic activities to the wild-type as indicated by total-phosphorylation levels of the substrate. We noted that the detrimental effect on autophosphorylation by the c-terminal deleted variants, in particular 531X, was more evident in the presence of the substrate surrogate (Fig. 8d). Next, we explored the non-catalytic properties of de facto substrates (Fig. 8e) in phosphorylation experiments using 3D-K298M constructs with and without c-terminal residues 528–536 (528X) or 531–536 (531X) versus an active c-Src kinase. Surprisingly, both c-terminal variants were also equally and efficiently phosphorylated, as shown by their total phospho-tyrosine levels; however, the 531X mutant could not be phosphorylated on Tyr 530, resulting also in the allosteric inhibition of c-Src Tyr 530 on the active kinase (Fig. 8e). We noticed also a rapid and transient allosteric impediment on Tyr 419 autophosphorylation that recovered over the time-course (Fig. 8e). On the contrary, the inhibitory effect on the c-terminal Tyr 530 phosphorylation of the active kinase was more sustained and significant over time. These data provide evidence of the existence of a complex allosteric node at the c-terminus of c-Src controlling the crosstalk between substrate- and enzyme-acting kinases and how perturbation of this allosteric phospho-switch drives c-Src dysfunction in cancer.

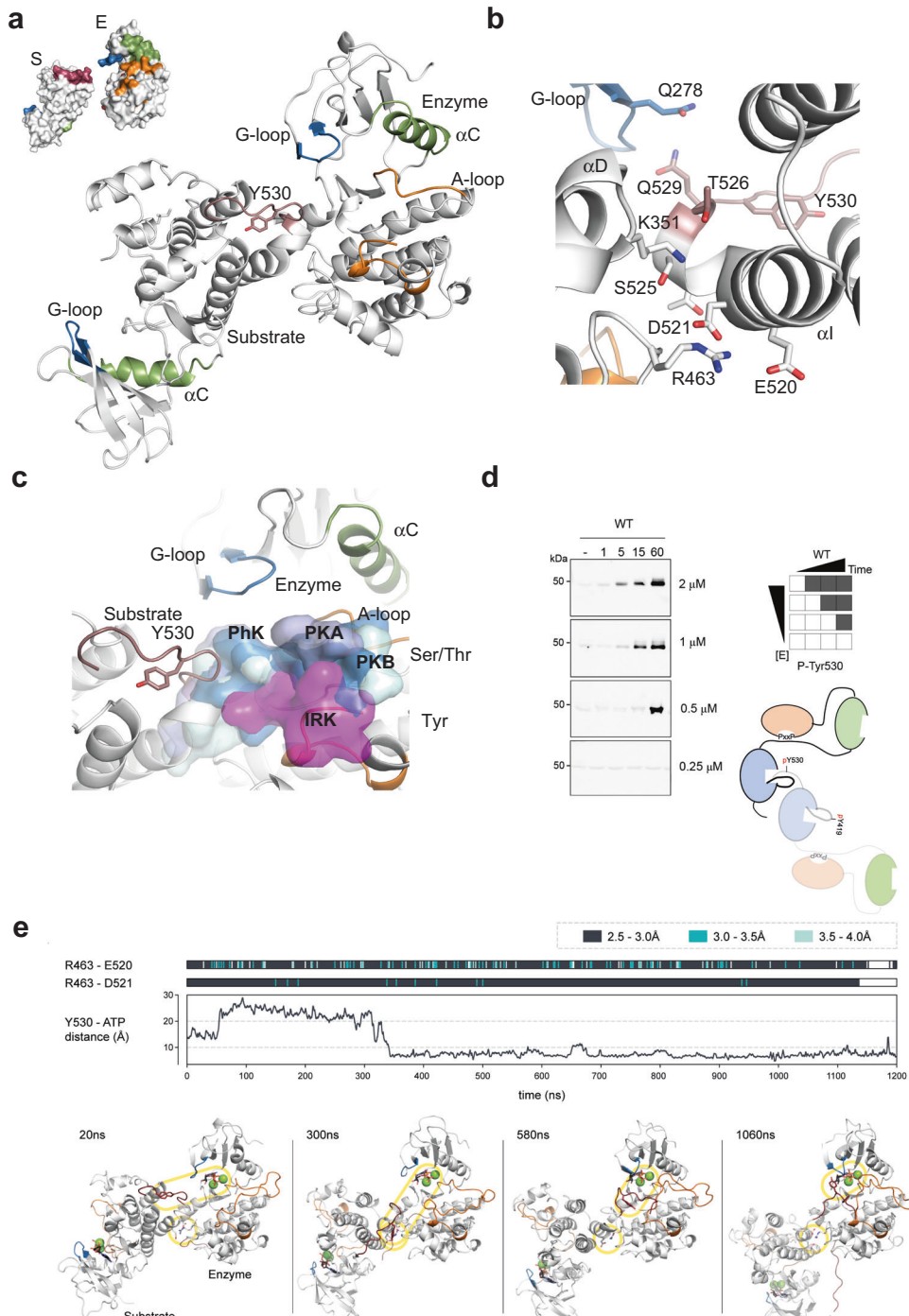

**Fig. 7 | A crystallographic snapshot of c-terminal Tyr 530 intermolecular autophosphorylation. a** Cartoon representation of the two molecules (chain A and B) of the asymmetric unit of the c-Src KD- Ponatinib crystal structure where the back side of the c-lobe of one protomer faced the substrate binding area located at the base of the active site cleft of the other. Selected residues and secondary structure elements are color-coded as in Fig. 5. **b** Close-up view of the interface between enzyme and substrate molecules showing secondary structural elements and interacting residues from (**a**). **c** Superimposition of c-Src KD-Ponatinib crystal structure, with previously described structures of protein kinases in complex with different peptides substrates: PKA (PDB 1ATP, light blue), PKB (PDB 1O6K blue, 1O6L cyan), PhK (PDB 2PHK, soft gray) and IRK (PDB 1RK3, magenta) in surface semitransparent representation. **d** WB analyses of an in vitro time-course autophosphorylation assay with a c-Src WT 3D- increasing concentrations (from 0.25 to 2 μM) of c-Src WT (3D-construct) in the presence of saturating concentrations of ATP

(1 mM) and MgCl₂ (2 mM) for 0–60 min. Total c-Src protein was evaluated by Coomassie staining. Data are representative of 3 independent experiments. **e** Three time-course plots show molecular interactions across a molecular dynamics (MD) simulation. The top two plots show the minimum distance between R463 to E520 (top) and between R463 to D521 (middle). Distance is represented by color, as indicated by the legend in the top-right. The line graph (bottom) shows the minimum distance between Tyr 530 and ATP. Distance is represented across the Y-axis. For all three graphs, time is represented by the shared X-axis at the bottom. Snapshots from four time points in the MD simulation are indicated at the top-left. For each snapshot, the substrate molecule is shown in salmon, while the enzyme molecule is shown in white. The top circle encapsulates Y530 and ATP, while the bottom circle encapsulates R463, E520, and D521, lower panel. Source data are provided in the Source Data file.

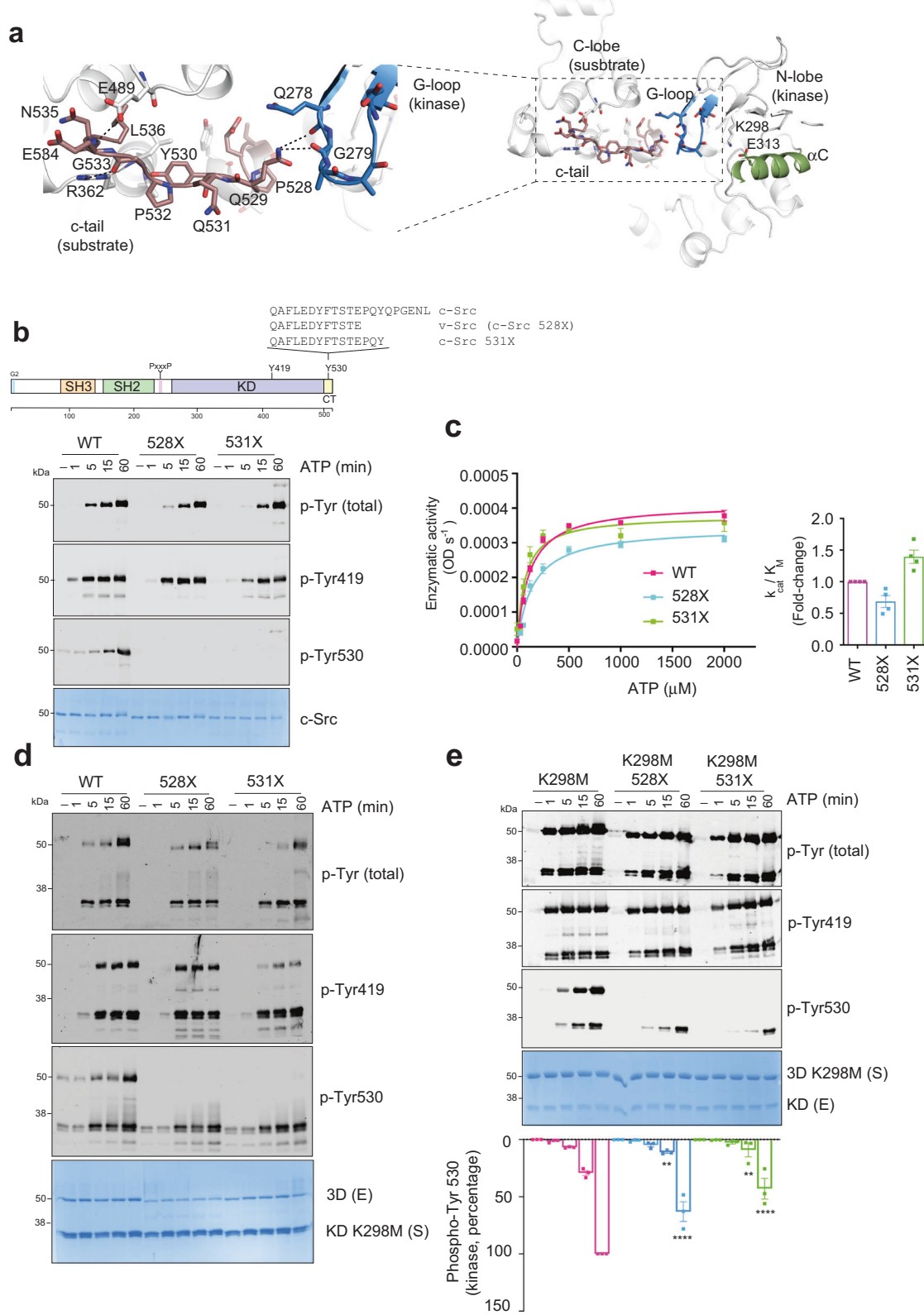

## Discussion

In this study, we uncover a self-autonomous mechanism controlling c-Src function by autophosphorylation. Our work dissects a sequential and coordinated cis-to-trans autophosphorylation switch connecting the activation loop and c-terminal segment that is required for enzyme specificity and non-catalytic functioning as a substrate. Our crystal structure is compatible with the asymmetric arrangement of the substrate and active kinases during intermolecular autophosphorylation. Furthermore, we identify a c-terminal palindromic phospho-motif containing Tyr 530 on the substrate molecule that engages the G-loop of the active kinase prior to intermolecular phosphorylation. Functional evaluation of cancer-related c-terminal variants targeting the palindromic phospho-motif provides evidence of the existence of a complex allosteric node at the c-terminus of c-Src

**Fig. 8 | A c-terminal palindromic phospho-motif controls c-Src function.**
**a** Cartoon representation of the kinase-substrate (chain A and B) asymmetric engagement of two c-Src molecules in the crystal structure. Left panel, close in view of the c-terminal aa sequence of c-Src containing the palindromic PQYQP motif at the interface between the two molecules showing residues coordinated by intra- and intermolecular interactions. **b** Schematic diagram of the functional domains and main autophosphorylation sites of c-Src. Different c-terminal sequence variants are depicted. WB of samples from a time-course autophosphorylation experiment with c-Src WT, v-Src and 531X (3D-construct, 1 μM) in the presence of ATP (1 mM) and MgCl₂ (2 mM) for 0–60 min using the indicated antibodies. The total amount of protein was visualized by Coomassie staining. **c** Enzyme kinetics and catalytic efficiency constant ($k_{cat}/K_M$, fold-change) for ATP using Src WT, v-Src and 531X (3D-construct, 1 μM final concentration) at a fixed

concentration (2–4 mg/ml) of Abl peptide. Data represent the mean ± SEM of 3 experiments in duplicate ($n = 6$). **d** WB of samples from a time-course phosphorylation experiment with c-Src WT, v-Src and 531X (3D-construct, 1 μM) in the presence of a c-Src KD K298M (3 μM) for 0–60 min using the indicated antibodies. The total amount of protein was visualized by Coomassie staining. Data are representative of 3 independent experiments. **e** WB of samples from a time-course phosphorylation experiment with c-Src KD WT (1 μM), in the presence of substrate surrogates c-Src, v-Src and 531X K298M (3D-construct, 3 μM) for 0–60 min using the indicated antibodies. The total amount of protein was visualized by Coomassie staining. Phospho-tyrosine 530 signal quantification on the active kinase molecule, data represent mean ± SEM of 3 experiments ($n = 3$), ** $p = 0.003$, **** $p = 0.0001$, 2-way ANOVA multiple comparison test. Source data are provided in the Source Data file.

modulating the crosstalk between substrate- and enzyme-acting kinases. We applied a tripartite expression system for heterologous production of unphosphorylated c-Src in bacteria (Fig. 1). This system allowed us to increase protein solubility 5- to 10-fold compared with previous bi-partite expression systems where c-Src was co-expressed with YopH or GroEL[27]. By applying a systematic biochemical and mass spectrometry approach, we identify Tyr 530 at the c-terminal segment as a de facto c-Src autophosphorylation site with slow time-resolution kinetics and strong intermolecular component (Figs. 2b–d and 7d). These data are further supported by a crystal structure capturing a snapshot of the c-terminal segment of a c-Src substrate molecule positioned for ready entry into the active site of the kinase molecule (Fig. 7a–c). These results together demonstrate that: (1) the intrinsically disordered N-terminal region of c-Src does not promote direct dimerization in either the apo or the ATP-complexed states (Supplementary Fig. 1a, b); and (2) the lack of binding effect seen on both thermal stability and catalytic activity in experiments using myristoylated G2-derived peptides (Supplementary Fig. 1c, d), made us conclude that contrary to the c-Abl paradigm[48,49], c-Src myristoylation on G2 play a compartmentalization role rather than a functional or catalytic one.

We dissected three sequential and coordinated steps in the mechanism of c-Src autophosphorylation that couples each phosphorylation event to a specific conformational and functional state (Figs. 2 and 3). Initially, a c-Src molecule adopts predominantly a closed inactive configuration with an αC-out and the catalytic salt bridge between K298 and E313 substituted by the E313-R422 tether, based on our SAXS data (Supplementary Fig. 2) and the crystal structure PDB ID: 2SRC. It is likely that a partially extended configuration with a flexible and dynamic activation loop can be adopted as a transition to the active state based on PDB ID: 1Y57 (https://www.rcsb.org/structure/1y57). Autophosphorylation on activation loop Tyr 419 takes place rapidly in cis- as a priming step in the absence of c-terminal phosphorylation, very likely in an open conformation where the SH2 domain is not engaged with c-terminal Tyr 530 (Fig. 9). This step requires the transition of the αC-helix from "out" to "in", so coordination of residues required for catalysis can take place. This priming step switches then to intermolecular phosphorylation at later stages compatible with the release of the activation loop to be exposed so that the active site is accessible to another substrate molecule. This is followed by the trans-phosphorylation of the c-terminal Tyr 530 by another c-Src kinase molecule. The first autophosphorylation step in cis is consistent with a lack of enzyme concentration effect seen at early time points in the phosphorylation reaction of Tyr 419 (Fig. 4a). This is supported by a crystal structure of c-Src (PDB ID: 2SRC) in which Tyr 419 is pointing to the active site and coordinated with the HRD motif (4 Å to R385) at the same time that the γ-phosphate of the ATP analog coordinates with the DFG motif (4.8 Å from D407), and in close range of Tyr 419[16]. The second autophosphorylation step is consistent with an enzyme concentration effect seen at later time points in the Tyr 419 autophosphorylation reaction (Fig. 4a) and consistent with a

crystal structure of a phosphorylated c-Src in active conformation and an extended activation loop (PDB ID: 1YI6, https://www.rcsb.org/structure/1yi6). In this setting, the active and substrate binding sites are accessible to another molecule to undergo intermolecular phosphorylation. The third autophosphorylation step is supported by our structural and biochemical data demonstrating that Tyr 530 is a de facto c-Src autophosphorylation site (Fig. 2b, d) with a strong intermolecular component driven by a protein concentration effect during the autophosphorylation reaction (Fig. 7d). This final step is dependent on the previous phosphorylation of Tyr 419 on the activation loop (see Figs. 3a, b and 4b), so phosphorylation of both activation loop and c-terminal segments are sequential and coordinated events. The perturbation of such a sequential and coordinated mechanism by a c-terminal Y530F mutant bypasses the intermolecular phosphorylation in a way that it only displays a unique cis-component for the mechanism of autophosphorylation (Fig. 4a).

Our data using catalytic dead c-Src K298M variants as intact substrate surrogates provide solid evidence about how "activating" Tyr 419 plays a key non-catalytic role in the presentation of the activation loop of c-Src as a substrate to another kinase molecule (Fig. 4c). On the catalytic facet, rather than having a significant impact on the overall c-Src enzymatic activity (Figs. 3b and 4b), activation loop Tyr 419 dictates enzyme specificity toward substrates (Fig. 3). These data are consistent with the partial phenotypic rescue observed in different tissues by *Drosophila* Src42A (Fig. 5a–d) lacking the activation loop tyrosine (Y400F). Differences in the strength of the phenotypes observed when compared with the expression of the WT allele could be explained by (1) mutant protein being less efficient functionally (i.e., lacking the non-catalytic facet), (2) differences in substrate specificity related to different tissues types, and (3) a worse accumulation of the mutant protein (expression levels were not directly assessed). In the case of tracheal tissue (Fig. 5c, d), it is plausible that in a wild-type background, the lack of substrate presentation properties by the activation loop mutant has no dominant effect and is compensated by the wild-type allele, which is active both as an enzyme and substrate. On the contrary, in a loss of function genetic background using the *Src^F80* allele, whose product lacks enzymatic activity but retains substrate function, the activation loop mutant allele has a dominant effect by rescuing the activity of *Src^F80* in trans via phosphorylation. Furthermore, using K298M substrates surrogates, we uncover an allosteric node at the c-terminal palindromic phospho-motif. This allosteric switch was demonstrated by the fact that contrary to wild-type or a Y530F mutant, a c-terminal oncogenic variant c-Src 531X that lacks residues 531 to 536 and perturbs the palindromic phospho-motif (PQYQP) has a detrimental effect on the phospho-tyrosine activity being unable to autophosphorylate on Tyr 530 (Fig. 8b). In agreement with this, mutating the palindromic phospho-motif to a PAYAP sequence resulted in a total specific loss of Tyr 530 autophosphorylation while keeping intact the catalytic activity (Supplementary Fig. 9). This detrimental impact was caused

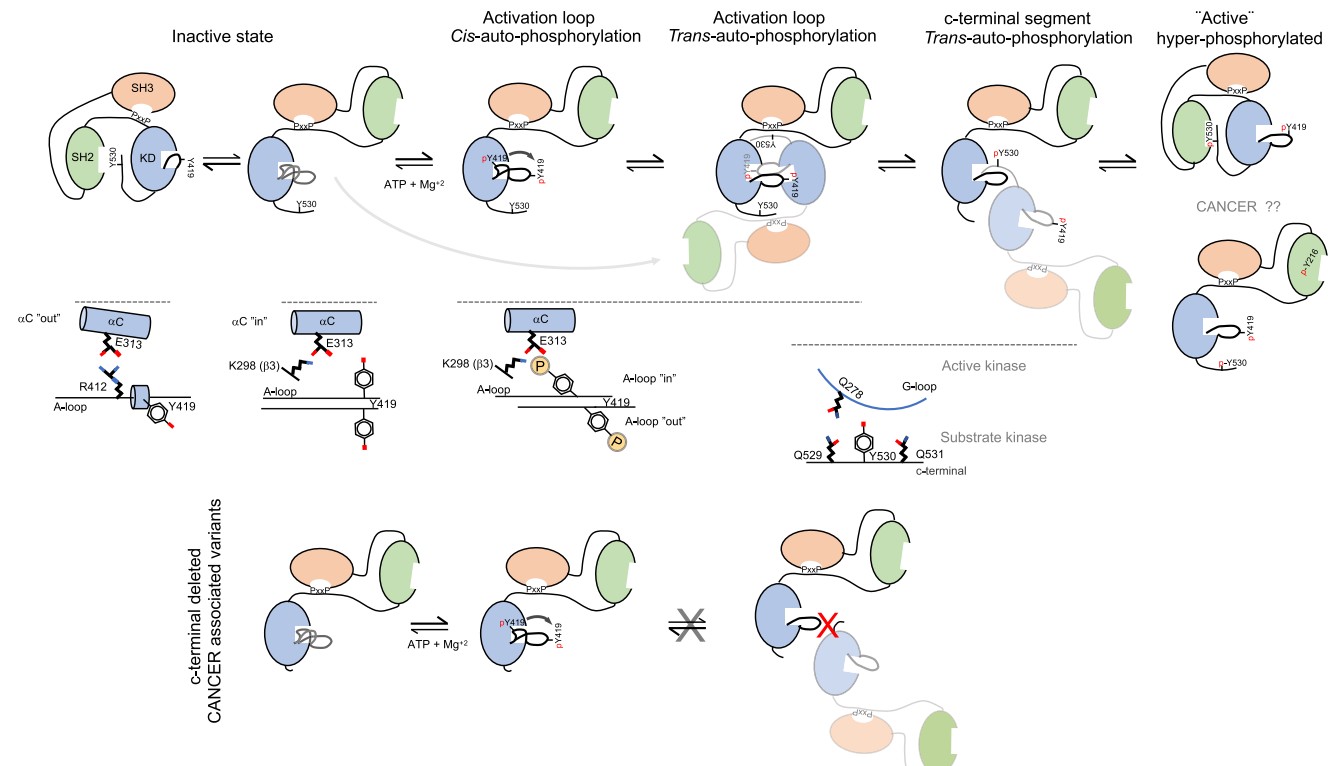

**Fig. 9 | Proposed model for the regulation of c-Src function by self-autonomous autophosphorylation mechanisms.** A c-Src molecule adopts a predominant closed inactive conformation in the absence of c-terminal phosphorylation with an αC-out and the catalytic salt bridge between K298 and E313 substituted by the E313-R422 tether. It is likely that a partially extended configuration with a flexible and dynamic activation loop can be adopted as a transition to the active state. Upon ATP binding, the αC moves to an "in" state, and the catalytic tether (K298-E313) is restored for fast intramolecular phosphorylation on Tyr 419 that switches into an intermolecular mechanism by an activation loop "in-to-out" transition at later stages (intermediate state) that precedes c-terminal Tyr 530 intermolecular phosphorylation of the substrate molecule by the active kinase. Finally, a hyper-phosphorylated c-Src adopts a highly active yet unknown conformation (see also Fig. 2e). This model has important implications in cancer and provides a plausible molecular explanation for the: (1) high levels of hyper-phosphorylated c-Src on both activation and c-terminal segment found in aggressive types of cancer such TNBCs, and (2) perturbation of the allosteric phospho-switch by a c-terminal truncated cancer associated variants. Non-phosphorylable c-terminal variants adopt a predominant extended configuration and undergo intramolecular phosphorylation on the activation loop only. Intermolecular phosphorylation of the c-terminal tyrosine requires contact between a c-terminal palindromic phospho-motif of the substrate molecule and the G-loop of the active kinase.

by an unexpected allosteric effect on the activity of the active kinase by the substrate molecule (Fig. 8e).

Pharmacological inhibition of driver protein kinases by small-molecule inhibitors is one of the main therapeutic strategies to treat cancer patients. We solved the crystal structure of c-Src KD in complex with Ponatinib (Fig. 6a, b), a type-II tyrosine kinase inhibitor used for the treatment of chronic myeloid leukemia (CML) and Philadelphia chromosome–positive (Ph+) acute lymphoblastic leukemia (ALL)[50,51]. This crystal structure resembles to some extent the previously solved c-Src KD-Imatinib crystal structure, and despite sharing the same compound binding pose and DFG-out configuration (Fig. 6b–d), provided structural evidence for an asymmetric dimer arrangement that is in accordance with our biochemical data. Our work provides answers to two important questions for a better understanding of the role of c-Src in human cancers. On one hand, hyper-activation of c-Src in human cancers might result from the perturbation of self-autonomous mechanisms of autophosphorylation and allosteric control. Large-scale genomic sequencing projects indicate that gene amplification and activating mutations in c-Src do not play a significant role in human tumor biology[31,32]. In fact, paradoxically high levels of phosphorylated c-Src in both activation and c-terminal segments are found in aggressive cancer types such TNBC[52]. We have data supporting these findings, on one hand, a previously phosphorylated c-Src protein (90–120 min) is highly active in solution and is able to phosphorylate an intact substrate surrogate with faster kinetics than the non-phosphorylated protein (Supplementary Fig. 10a). On the other, WBs of extracts from tumor cell lines show elevated levels of both phospho-sites (as indicated by phospho-specific antibodies) in LAM1, KELLY, TT and LOXO-292-treated TPC1 cells among other (Supplementary Fig. 10b). It is also plausible that c-Src may function independent of CSK, which may not always play an anti-oncogenic role through the negative regulation of c-Src in carcinogenesis. In this line, elevated expression of CSK in human cancer cell lines appears to correspond to elevated c-Src protein-tyrosine kinase activity, for example, CSK is widely expressed in HT-29 and SW620 cells that contain high levels of active c-Src[53,54]. Furthermore, genetically engineered mouse models showed that in the absence of CSK, c-Src phosphorylation at Tyr 527 in vivo was reduced to about 20–50% of the level in wild-type cells. These results support the notion that autophosphorylation or another kinase is also involved in c-terminal tyrosine phosphorylation[55–57]. We dissect a phosphorylation switch connecting the activation and c-terminal segments that control the catalytic and non-catalytic functions of c-Src. We visualized by X-ray crystallography an asymmetric dimer arrangement of both catalytic domains compatible with the intermolecular phosphorylation mechanism between enzyme and substrate kinases. Further, we identify functionally relevant contacts between a c-terminal palindromic phospho-motif on the substrate and the G-loop of the kinase-acting molecule. Evaluation of cancer-related c-terminal variants perturbing the palindromic phospho-motif provides evidence of the existence of a complex allosteric node at the

c-terminus of c-Src controlling the crosstalk between substrate- and enzyme-acting kinases; and how perturbation of this allosteric phospho-switch drive c-Src dysfunction in cancer. Other examples of asymmetric allostery in protein kinases are the EGFR paradigm, where a crosstalk between the C-lobe of the activating kinase and the N-lobe of the allosterically activated receiver kinase controls activity[58]. In this context, the C-lobe of the 'activator' contacts the N-lobe of the 'receiver' at points of the αC-helix and the β4/β5 loop (PDB ID: 2GS6, https://www.rcsb.org/structure/2gs6). Formation of this asymmetric kinase dimer interface induces allosteric changes in the N-lobe extension of the receiver kinase leading to the conformational changes in the αC-helix and activation loop required to switch on the activated state. This arrangement is analogous to the one previously observed for the CDK2/Cyclin A complex[59] see PDB ID: 1FIN (https://www.rcsb.org/structure/1fin). There are also examples of allosteric communication between the C-lobe of the active kinase and the substrate molecule mediated by the αH and αI helices interface of the kinase, e.g., Tpk1[60,61]. Altogether, the data described here represent de facto "vulnerabilities" that can be therapeutically exploited for the design and development of next-generation c-Src inhibitors targeting non-catalytic and/or allosteric features.

## Methods

### Plasmids

A pET28a-TEV plasmid containing an N-Terminal 6xHis tag followed by a Tobacco etch virus (TEV) protease recognition site was used to express human c-Src (UniProtKB P12931) three-domain (3D, SH3-SH2-KD aa 84–536) or kinase domain (KD, aa 254–536) coding sequences. Both pFCDUET-YopH phosphatase and pGKJE8-GroEl/GroEs chaperones plasmids (Takara chaperone plasmid set cat. # 3340) were used for co-expression experiments. These constructs were a kind gift from Dr. Daniel Lietha (CIB, Madrid). For expression of recombinant *Drosophila* c-Src isoform 42 A (Src42A, UniProtKB Q9V9J3), we used a pET-His10-Src42A plasmid (Addgene #126674) that codes for a full-length *Drosophila* isoform (aa 1–517) with a N-terminal 10x His tag and a Thrombin protease recognition site. For experiments in *Drosophila* a pUAST-attB plasmid encoding *Drosophila* Src homolog isoform A (Src42A) was cloned by amplifying Src42A coding sequence from a pGEX-Src42A donor plasmid (Addgene #126673) using primers with EcoRI (forward 5′- CTGAATAGGGAATTGGGAATTCATGGGTAACTG CCTCACC-3′) and XbaI (reverse 5′- CCTTCACAAAGATCCTCTAG ATCAGTAGGCCTGCGCCTC-3′) restriction sites.

### Site-directed mutagenesis

Site-directed mutagenesis was performed in order to generate all the point mutants and variants described in this study using the Q5-site-directed mutagenesis kit (New England Biolabs) following manufacturer instructions and the indicated primers (see extended "Methods").

### Expression and purification of recombinant proteins

In order to express high yields of soluble, unphosphorylated and monodisperse recombinant human c-Src protein, we followed a modified protocol, in which in addition to YopH phosphatase[37], c-Src was also co-expressed with the chaperone GroEL. For co-expression, pET28a-TEV-hSrc [84–536] and pET28a-TEV-hSrc [254–536] plasmids were co-transformed in *E. coli* BL21 bacteria strand previously transformed with pFCDUET-YopH and pGKJE8-GroEl/GroEs plasmids and grown in 50 ml of LB media containing kanamycin (50 μg/ml), streptomycin (50 μg/ml) and chloramphenicol (34.5 μg/ml) overnight at 37 °C shaking at 200 rpm in a 250 ml Erlenmeyer flask. The next day, the bacterial culture was diluted (1:100) in LB media with the same antibiotic concentrations and grown at 37 °C until the bacterial culture reached an optic density of 0.15 (λ 600 nm), then the culture was cooled to 18 °C and tetracycline was added at a final concentration of 1 ng/ml in order to trigger chaperone expression. When the optical density reached 0.4–0.5 (λ 600 nm), IPTG was added to a final concentration of 500 μM, and the culture was left overnight shaking (200 rpm) at 18 °C. To express recombinant dSrc42A, we followed a similar in which a pET-His10-Src42A plasmid was co-transformed with the pCFDUET-YopH construct only in BL21. The next day, bacterial culture was harvested at $1000 \times g$ and 4 °C. The pellet was resuspended in 50 ml lysis buffer (50 mM Tris pH 8, 500 mM NaCl, 0.1 mM PMSF) and sonicated on ice. Crude lysate was clarified by centrifugation at $48,000 \times g$ at 4 °C for 45 min. Soluble clarified supernatant underwent a further 10-s sonication step and was filtered through a 45 μm syringe filter. Human c-Src was purified by three chromatographic steps (see Fig. 1). First, the lysate was passed through an immobilized metal affinity chromatography (IMAC) column (GE Healthcare HisTrap™ HP) equilibrated with 5 column volumes (CVs) of IMAC buffer A (20 mM Tris pH 8, 150 mM NaCl, 1 mM TCEP, 5% Glycerol) with an GE Healthcare AKTA PURE FPLC at flowrate of 5 ml/min. Next, the column was washed with 90% IMAC buffer A and 10% IMAC buffer B (20 mM Tris pH 8, 150 mM NaCl, 300 mM Imidazole, 1 mM TCEP, 5% Glycerol) until the absorbance signal at a λ 280 nm was stable (usually 10 CVs), after which a 100% gradient with IMAC buffer B was run in 100 ml (20 CVs). Fractions containing the recombinant protein were collected and tested by SDS-PAGE before diluting in IEC (ionic exchange chromatography) buffer A (20 mM Tris pH 8, 1 mM DTT, 5% glycerol) up to 3 times the original volume so NaCl concentration was lowered to 50 mM. Then, the sample was loaded into a GE Healthcare HiTrap Q HP column previously equilibrated with 5 CVs of IEC buffer A, then the column was washed with 5 CVs IEC buffer A, followed by a 100% gradient with IEC buffer B (20 mM Tris pH 8, 500 mM NaCl, 1 mM DTT, 5% glycerol). Fractions containing the recombinant protein were collected and tested by SDS-PAGE, pulled together, and mixed with a His-tagged rTEV protease (20–40 μM) in a 1/20 molar TEV/c-Src stoichiometry. Buffer was supplemented with 2 mM TCEP and digested at 4 °C o/n. Alternatively, digestion was performed for 2 h at room temperature. Next, a His-trap reverse step was undertaken to remove the protease and tag of the recombinant protein. Briefly, His-rTEV protease digested c-Src sample (input) was passed through the HisTrap column previously equilibrated with IMAC buffer A at 2 ml/min flowrate with a peristaltic pump (GE Healthcare Pump P-1) and then washed with 5 CVs of buffer IMAC A supplemented with 50 mM imidazole. Flow through (FT) was taken, checked by SDS-PAGE and UV-absorbance (Thermo Scientific NanoDrop One). FT was then concentrated with a 10–30 kDa cutoff concentrator (Millipore) using an Eppendorf Centrifuge 5810R at $2,178 \times g$ at 4 °C. The sample was then injected in a size-exclusion chromatography (SEC) Superdex 200 16/60 column (GE Healthcare) previously equilibrated with 1.5 CVs of SEC buffer (20 mM Tris pH, 150 mM NaCl, 1 mM DTT, 5% glycerol). Fractions were collected, and protein purity and concentration were tested by SDS-PAGE gel and UV-absorbance.

### Size-exclusion chromatography with multi-angle light scattering (SEC-MALS)

A protein sample (0.2 mg/ml) was injected in a Superdex 200 Increase 10/300 column (Cytiva) previously equilibrated in 20 mM Tris pH, 150 mM NaCl, 1 mM DTT, 5% Glycerol buffer (filtered through a 0.1 μm filter). The chromatographic eluent was monitored by three consecutive detectors in series: (1) a multi-wavelength UV-Vis absorbance detector Monitor UV-900 of the AKTA system (GE Healthcare) with a 10 mm path length flow cell, (2) a light scattering DAWN Heleos 8+ (Wyatt Technology) with detectors at eight different angles (from 32 to 141° from the source) using a linearly polarized GaAs laser operating at 665 nm and (3) an Optilab T-rEX (Wyatt Technology) differential refractive index detector with a laser wavelength of 658 nm. Data collection and analysis were performed using UNICORN 5.10 (GE Healthcare) and ASTRA 6.0.3 (Wyatt Technology) software packages.

## Western blotting and antibodies

SDS-PAGE gels were transferred onto nitrocellulose 0.2 μm membranes (Amersham™). Transferred membranes were immersed in blocking solution (10 mM Tris pH 8, 150 mM NaCl, 5% weight/volume (w/v) skimmed powder milk) for 60 min. After blocking, membranes were washed three times with TBS-T (10 mM Tris pH 8, 150 mM NaCl, Tween-20 0.1% v/v) prior to incubation with primary antibody solution (TBS-T with BSA 5% w/v) o/n at 4 °C shaking. Antibodies used were: phospho-Src Tyr 216 (CSB-PA050132), phospho-Src Tyr 419 (D49G4, CST #6943), Src (36D10, CST #2109) phospho-Src Tyr 530 (Thermo-Fisher 44−662G and CST#2105) and total phospho-Tyr (p-Tyr-100 CST #9411) were diluted at 1:10000−1:5000. After incubation with the first antibody, membranes were washed with 20 ml TBS-T three times and immersed in secondary antibody solution (TBS-T skimmed powder milk 5% m/v). Secondary antibodies anti-rabbit or -mouse IgG DyLight conjugate at 680 or 800 nm (CST #5366, #5151, #5470, #5257) were used at double the dilution factor of the primary during 1 h at room temperature protected from light. After incubation with secondary antibodies, membranes were washed several times with TBS-T. Membranes were scanned in an Odyssey CLx scanner.

## In vitro phosphorylation assays

In vitro phosphorylation assays were performed at room temperature using recombinant proteins at 1 μM final concentration in buffer (20 mM Tris pH, 150 mM NaCl, 1 mM DTT, 5% Glycerol, 2 mM MgCl$_2$) and ATP (1 mM) at the indicated time points (unstimulated, 1, 5, 15, 30, 60 and 90 min). Aliquots from the time course were mixed with 5X Laemmli sample buffer (ThermoFisher) and denaturalized at 95 °C for 1−2 min at 95 °C.

## Enzymatic assays

Phosphorylation rates of peptide substrates by recombinant proteins at 1 μM final concentration were determined by using an NADH-coupled pyruvate kinase assay in the presence of increasing ATP concentrations. The enzyme-substrate solution (20 mM Tris-Cl, 1 mM MgSO$_4$, 400 mM Phosphoenolpyruvate (PEP), 100 mM NADH, 2450U/ml Pyruvate kinase (PK), 2.26 mg/ml Lactate dehydrogenase (LDH)) was prepared at different concentrations of ATP: 0, 0.08, 0.16, 0.32, 0.65, 1.25, 2.5 and 5 mM. The experiments were performed in a 384 well plate (Greiner Bio-one), and the NADH consumption was read at λ 340 nm during 120 cycles of 30 s each by using a Victor multilabel plate reader 1420 Multilabel Counter (Perkin Elmer). In order to obtain catalytic rates and kinetic constants by Michaelis-Menten, experiments were analyzed using Prism software.

## Mass spectrometry

For in-solution digestion, protein samples (2−5 μM) were reduced and alkylated (15 mM TCEP, 30 mM CAA, 30 min at RT in the dark) in the presence of urea 4 M and digested with trypsin in urea 1 M, 50 mM Tris pH 8 overnight at 37 °C (Promega) at an estimated protein:enzyme ratio 1:100. For FASP digestion, proteins were reduced and alkylated (15 mM TCEP, 30 mM CAA, 30 min in the dark, RT) and sequentially digested with chymotrypsin (SigmaAldrich) (protein:enzyme ratio 1:100, o/n at RT). In all cases, digestion was quenched by adding 0.1% TFA, and resulting peptides were desalted using C18 stage-tips. LC-MS/MS was done by coupling an UltiMate 3000 RSLCnano LC system to a Q Exactive Plus mass spectrometer (Thermo Fisher Scientific). Peptides (2−5 μl) were loaded into a trap column (Acclaim™ PepMap™ 100 C18 LC Columns 5 μm, 20 mm length) for 3 min at a flow rate of 10 μl/min in 0.1% formic acid. Then, peptides were transferred to an EASY-Spray PepMap RSLC C18 column (Thermo) (2 μm, 75 μm x 50 cm) operated at 45 °C and separated using a 60 min effective gradient (buffer A: 0.1% FA; buffer B: 100% ACN, 0.1% FA) at a flow rate of 250 nL/min. The gradient used was from 4 to 6% B in 2 min, from 6 to 33% B in 58 min, plus 10 additional

minutes at 98% B. Peptides were sprayed at 1.5 kV into the mass spectrometer via the EASY-Spray source. The capillary temperature was set to 300 °C. The mass spectrometer was operated in a data-dependent mode, with an automatic switch between MS and MS/MS scans using a top 15 method (intensity threshold ≥4.5 × 10$^4$, dynamic exclusion of 5 or 10 s and excluding charges unassigned, +1 and > +6). MS spectra were acquired from 350 to 1500 m/z with a resolution of 70,000 FWHM (200 m/z). Ion peptides were isolated using a 2.0 Th window and fragmented using higher-energy collisional dissociation (HCD) with a normalized collision energy of 27. MS/MS spectra resolution was set to 35,000 (200 m/z). The ion target (IT) values were 3 × 10$^6$ for MS (maximum IT of 25 ms) and 10$^5$ for MS/MS (maximum IT of 110 ms). Raw files were processed with Maxquant (v 1.6 and higher) using the standard settings against the corresponding sequences of the expressed recombinant proteins and an *E. coli* protein database (UniProtKB/Swiss-Prot, 20373 sequences) supplemented with contaminants. Carbamidomethylation of cysteines was set as a fixed modification, whereas oxidation of methionines, protein N-term acetylation and phosphorylation of serines, threonines and tyrosines were set as variable modifications. Minimal peptide length was set to 7 amino acids, and a maximum of two tryptic missed-cleavages were allowed. Results were filtered at 0.01 false discovery rate (FDR), both peptide and protein levels. Raw data were imported into Skyline. Label-free quantification of identified phosphopeptides was performed using the extracted ion chromatogram of the isotopic distribution. Only peaks without interference were used for quantification. Phosphopeptide intensities were normalized by the intensity of non-modified peptides from the target protein.

## Differential scanning fluorimetry (DSF)

To evaluate the thermal stability of recombinant c-Src in the absence of (apo) and in complex with Ponatinib, we applied an indirect SYPRO Orange-based method. For this assay, the total reaction volume was adjusted to 40 μl at 1−2 μM protein, 10 μM inhibitor, and 2 x SYPRO Orange concentrations subjected to a gradient of temperature from 20 to 95 °C. Fluorescence was measured on an Applied Biosystem 7300 Real-Time PCR system.

## *Drosophila* work

*Drosophila melanogaster* strains were maintained and raised at 25 °C under standard conditions. The stocks used, described in in Flybase (http://flybase.org/), are the following ones: GMRGal4 (eye expression, BSDC 9146), salEPvGal4 (wing expression, BDSC 80573), btlGal4 (tracheal expression, gift of S. Hayashi), src$^{F80}$/CyOlacZ; UASSrc/TM6 dfdYFP (gift of S. Luschnig); UASSrcY400F/TM6 dfdYFP (this work). Balancer chromosomes CyO, TM3 or TM6 marked with LacZ, GFP or YFP were used to follow the mutations and constructs of interest in the different chromosomes. Transgenes were generated by injecting the pUAST-attB-Src42a WT and Y400F construct to obtain directed insertions at 68E by the "Transgenesis Service" of the "Centro de Biología Molecular Severo Ochoa" (CBM, Madrid). Transgene expression was achieved using the Gal4/UAS system[62] at 25 °C (GMRGal4) or 29 °C (btlGal4 and salEPvGal4).

Embryos were stained following standard protocols. Primary antibodies used were goat anti-GFP (1:600) from Roche and chicken anti-β-gal (1:200) from Abcam. Alexa Fluor 488, 555, 647 (Invitrogen) secondary antibodies were used at 1:300 in PBT 0.5% BSA. CBP (Chitin Binding Protein, produced by N. Martín in Dr. Casanova's lab, New England Biolabs Protocol) was used as a secondary antibody at 1:300 to detect chitin and visualize the tracheal branches. Images from fixed embryos were taken using Leica TCS-SPE with the 20x and 63x immersion oil (1.40−0.60; Immersol 518F−Zeiss oil) objectives and additional zoom. Images from adults were obtained with an Olympus MVX10 macroscope using EFI (extended focus imaging) at the ADM facility of IRB-PCB. For morphometric analyses, confocal projections

were used to analyze the length of the embryonic dorsal trunk (DT) stained with CBP in stage 16 embryos. We traced the path using the freehand line selection tool of Fiji (ImageJ) software between the junction DT/Transverse Connective (TC) from metamere 2 to 9 following the DT curvature. DT length was expressed as the ratio between the DT path and the length of the embryo. Data from quantifications was imported and treated in the Excel software and in GraphPad Prism 9.0.0, where graphics were finally generated. Graphics shown are scatter dot plots, where bars indicate the mean and the standard deviation. Statistical analyses comparing the mean of two groups of quantitative continuous data were performed in GraphPad Prism 9.0.0 using unpaired two-tailed Student's $t$-test applying Welch's correction. Differences were considered significant when $p < 0.05$. * $p < 0.05$, ** $p < 0.01$, *** $p < 0.001$, **** $p < 0.0001$ where n.s. means not statistically significant.

### Crystallization, diffraction, data collection, and processing

Crystals of the c-Src KD (aa 252–536, human) in complex with ponatinib were obtained by mixing recombinant protein and ponatinib in a 1:2 molar ratio. The sample was concentrated using an Amicon Ultra 10K molecular weight cutoff centrifugal filter device (Merck Millipore, Billerica, Massachusetts, USA) up to 8 mg/ml. The final protein concentration was determined by UV spectroscopy (Nanodrop one, Thermo Scientific). Crystallization for c-Src KD in complex with ponatinib was achieved by sitting-drop vapor diffusion at 20 °C in MRC-2 crystallization plates (Molecular Dimensions, Newmarket, Suffolk, England) by mixing 0.5 µl of protein-ligand solution at 8 mg/ml with 0.5 µl of reservoir solution. After several days some crystals were obtained in a drop containing 50 mM sodium acetate pH 4.6, 100 mM sodium chloride, 20% w/v PEG 4000 and 10% v/v 2-Propanol. A cryo-protectant solution consisting of the reservoir solution, including 20% (v/v) glycerol, was used to freeze the crystals and mounted in Litho-Loops (Molecular Dimensions, Newmarket, England) prior to vitrification in liquid nitrogen for data collection. X-ray diffraction data were collected at a wavelength of 0.98 Å on beamline XALOC-BL13 of the ALBA Synchrotron Light Facility (Barcelona, Spain) using a Pilatus 6 M pixel detector (Dectris Ltd, Baden, Switzerland). Crystals were kept at 100 K during data collection. Reflections were integrated with the program iMOSFLM and reduced using POINTLESS, AIMLESS and TRUNCATE, all integrated in the Collaborative Computational Project Number 4 (CCP4). Molecular replacement was carried out using the atomic coordinates of c-Src KD in complex with imatinib (PDB ID: 3EL8, https://www.rcsb.org/structure/3el8) using PHASER[63]. Adjustment of the model was performed with COOT[64], and refinement was carried out with Refmac5[65] applying twin refinement amplitude-based settings. Model validation was carried out with MOLPROBITY, and structure figures were made using the graphics program PYMOL. For data statistics, see Table 1.

### Small-angle X-ray scattering (SAXS)

SAXS experiments were conducted at the beamline B21 of the Diamond Light Source (Didcot, UK)[66]. A sample of 40 µl of Src WT (3D-construct) at a concentration of 3 mg/ml was delivered at 20 °C via an in-line Agilent 1200 HPLC system in a Superdex 200 Increase 3.2/300 column (Cytiva), using a running buffer composed of 20 mM Tris pH 8.0, 150 mM NaCl, 1% glycerol and 1 mM DTT. The continuously eluting samples were exposed for 20 s, and a total number of 599 frames were recorded using an X-ray wavelength of 1 Å and a sample-to-detector (Eiger 4M) distance of 3.6 m. The frames recorded immediately before the elution of the sample were subtracted from the protein scattering profiles. The Scåtter software package (www.bioisis.net) was used to analyze data, buffer-subtraction, scaling, merging and checking possible radiation damage of the samples. The $R_g$ value was calculated with the Guinier approximation, assuming that at very small angles $q < 1.3/R_g$. The

particle distance distribution, $D_{max}$, was calculated from the scattering pattern with GNOM, and shape estimation was carried out with DAMMIF/DAMMIN, all these programs were included in the ATSAS package[67]. The protein molecular mass was estimated with GNOM. A generated PDB-based homology model was made using the program COOT by manually adjusting the X-ray structures (PDB ID: 2SRC, https://www.rcsb.org/structure/2src), into the envelope given by SAXS until a good correlation between the real-space scattering profile calculated for the homology model matched the experimental scattering data. This was computed with the program FoXS[68].

### Molecular dynamics simulations

The crystal structure of the active Src dimer presented in this study (PDB ID: 7OTE, https://www.rcsb.org/structure/7ote) was used as a starting conformation for molecular dynamics simulations. The disordered residues in the activation loop (410–426) were modeled using a crystal structure of Src in the active conformation (PDB ID: 1YI6). The dimer model was minimized using RosettaRelax[69] with all-atom constraints to the native coordinates. On the activation loop, Tyr 419 was modeled as a phospho-tyrosine using PyTMs[70]. In the kinase active site, we replaced the Ponatinib with an ATP and 2 magnesium ions. These modifications were all applied to both monomers. Unbiased all-atom molecular dynamics simulations were performed using GROMACS 2021.4[71]. Structures were parameterized using the CHARMM36[72] force field and solvated with the TIP3P water model. Random solvent molecules were replaced with sodium or chloride ions to neutralize the charge of the system and bring the concentration to 0.1 mol/l. The system was contained in a dodecahedron at least 1 nm larger than the protein from all sides with periodic boundary conditions. Long-range interactions were calculated with particle mesh Ewald. Neighbor lists were maintained using the Verlet cutoff scheme. The system underwent the steepest descent minimization until the maximum force was <100 kJ/mol. Canonical ensemble[73] was used to heat the system from 0 to 310 K in 100 ps. Isothermal–isobaric ensemble[74] (1 bar, 310 K) was applied for 100 ps. Positional restraints were applied during equilibration. Production runs used 2 fs time steps.

## Data availability

The crystallographic coordinates and structure factors for the crystal structure of human c-Src KD in complex with Ponatinib reported in this paper were deposited in the Protein Data Bank (PDB) with the PDB ID code 7OTE. The mass spectrometry proteomics data have been deposited to the ProteomeXchange Consortium via the PRIDE partner repository with the dataset identifier PXD037287. The SAXS data was deposited in the small-angle scattering biological data bank (SASBDB) with ID: SASDSV8. Data related to this article are provided in the Supplementary Information/Source Data file, further enquiries to iplaza@cnio.es. Source data are provided with this paper.

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

## Acknowledgements

We are grateful to Clara Santiveri and Ramón Campos from the Spectroscopy and Nuclear Magnetic Resonance Unit (CNIO) for excellent technical support, the Genomics Unit (CNIO), and to members from the Kinases, Protein Phosphorylation and Cancer Group for their technical assistance, in particular, Alicia Marín, Marina Rodríguez and Julio Martínez-Torres for helping with recombinant protein expression and purification, plasmid generation and cloning. To Eduardo Zarzuela from the Proteomics Unit (CNIO) for assistance with Mass Spectrometry studies. We thank Nabil Djouder and Javier Klett for their helpful comments and advice on the manuscript. The authors would like to thank Diamond Light Source for beamtime (beamline B21, proposal mx30297) and the ALBA Synchrotron Light Facility (XALOC-BL13 beamtime, proposal 2020094499) and their staff for assistance during data collection. We thank the Centro Nacional de Investigaciones Oncológicas (CNIO), which is supported by the Instituto de Salud Carlos III and recognized as a "Severo Ochoa" Centre of Excellence (ref. CEX2019-000891-S, awarded by MCIN/AEI/ 10.13039/5 01100011033) for core funding and supporting this study. This work was further supported by projects: grant PGC2018-098449-B-I00 (M.L.) by Spanish Ministerio de Ciencia e Innovación (MCIN), BFU2017-86710-R funded by MCIN/AEI /10.13039/ 501100011033 and ERDF "A way of making Europe", PID2020-117580RB-I00 funded by MCIN/ AEI /10.13039/501100011033, RYC-2016-1938 funded by MCIN/AEI /10.13039/501100011033 and ESF "Investing in your future", and a Marie Curie WHRI-ACADEMY International grant (number 608765) to I.P.-M. FP7-PEOPLE-2013-COFUND - Marie-Curie Action: "Co-funding of regional, National and International Programmes" International grant (number 608765) to I.P.-M. and BES-2017 (ref: EV-2015-0510-17-3) grant from Agencia Estatal de Investigación to I.P.-M. and H.N.C.-H.

## Author contributions

Conception of the study and experimental design (I.P.-M.), manuscript writing, data analysis and figures preparation (I.P.-M.), data analysis *Drosophila* work, and figure preparation (M.L.), mass spectrometry including data analysis (J.S.-W. and J.M.), SAXS studies including data analysis and figure preparation (I.M.), MD simulations analyses and figure preparation (W.Y. and K.N.), experimental work (J.C., H.N.C.-H., P.S.-M. and A.M.-H.), X-ray crystallography (H.N.C.-H., P.S.-M) and funding acquisition (I.P.-M.).

## Competing interests

The authors declare no competing interests.

## Additional information

Iván Plaza-Menacho.

**Peer review information** *Nature Communications* thanks Susan Taylor,
QinXi Li, Usha Nagarajan and the anonymous reviewer(s) for their con-
tribution to the peer review of this work. A peer review file is available.

