## [Peer Review File · Nature Communications]

REVIEWER COMMENTS

Reviewer #1 (Remarks to the Author):

Cuesta, et al REVIEW

This is a comprehensive analysis of the complex mechanisms that are associated with Src auto-phosphorylation. The authors elucidate some novel pathways and convincingly distinguish between cis-auto-phosphorylation of the Activation Loop (Y419) vs. trans-auto-phosphorylation of the C-terminal tail (Y530). The latter provides insights into how Src may trans-phosphorylate heterologous substrates although the authors focus here primarily on different mechanisms for auto-phosphorylation. The authors first describe an improved method for purification of de-phosphorylated Src in *E. coli* and then go on to do a rigorous kinetic analysis. They find that there is a rapid auto-phosphorylation of Y416 and then show that this is a cis-mediated process. A second slower rate is associated with phosphorylation of Y530 on the C-terminal tail. Although the authors are working primarily with an *in vitro* system and purified protein, they nevertheless demonstrate this cross talk between the C-tail and the Activation Loop (AL), and it is likely that similar cross talk does occur in cells since they are using a three domain construct, not just the kinase domain. This is surprising as the C-terminal site was thought to be phosphorylated in cells by Csk. They creatively use a kinase dead version of Src as a surrogate substrate and also test their model with several other surrogate protein substrates as well as with peptides. The data shows convincingly that Src is capable of auto-phosphorylating its C-tail and the mechanistic insights are valid. The physiological relevance of this phosphorylation in cells is still an open question but the fact that dually phosphorylated Src is found in cells suggests that this question of Y530 phosphorylation in cells is still not clear. And cis-phosphorylation of Y419 is a process that is shared by almost all protein kinases but has not been explored in such depth previously. In addition to showing convincingly that auto-phosphorylation of the C-tail Y530 occurs in trans, in contrast to the Y416, which occurs by cis- auto-phosphorylation, the authors also provide a crystal structure to validate how the trans-auto-phosphorylation might work. Although the crystal structure in the presence of a type II inhibitor clearly represents a snapshot of one state in an ensemble of many likely intermediate conformations where the A-Loop is disordered, it does provide a plausible model showing how the C-tail could be presented to the active site of the adjacent kinase. The authors spend a great deal of time talking about the allosteric cross talk that takes place between the two sites - Y416 and Y530 where the A-Loop is toggling between an "In" (inactive) state and an "Out" (active) state. While much attention has focused on DFG In/Out and α C In/Out as part of the activation mechanism, less attention is focused on the A-Loop In/Out and, in particular, how the initial cis-autophosphorylation of Y416 on the A-Loop triggers the reorganization of the A-Loop to its "Out" state. This is the transition that the authors have captured here. While it is difficult to follow all of the data and it could be written more clearly, I think that this is an important contribution not only to Src activation but also to this transition process that is relevant for all kinases.

The quality of this data is excellent, and these are challenging experiments to do. The work also introduces some important new concepts that will be important for the field to evaluate, as the results will undoubtedly be true for other tyrosine kinases and perhaps for all protein kinases. The logic is hard to follow in some places and the writing could be clearer. There are some serious concerns as cited below, but overall this work should be considered for publication after some significant revisions.

General Recommendation: The authors should definitely distinguish between heterologous protein substrates vs. auto-phosphorylation when the target site is tethered in close proximity to the active site. My own hypothesis is that trans-phosphorylation of a heterologous substrate will require the ordered A-Loop "Out" conformation with the A-Loop phosphorylated on Y419 whereas trans-auto-phosphorylation of the C-Tail will be able to utilize an intermediate state as has been captured in this study. It would simplify the reading of the paper if the authors focused on A-Loop "In" vs. A-Loop "Out" and on cis-auto-phosphorylation and trans-auto-phosphorylation vs. trans-phosphorylation of heterologous proteins that are co-localized with Src at the plasma membrane. Cis-auto-

phosphorylation is easy because the site (Y419) is right there at the active site and simply requires the binding of the adenine ring of ATP. Phosphorylation of this site, however, pY419, then initiates the transition of the A-Loop from its "In" and inactive state to its "Out" and fully active state. In this study they have not captured that "Out" and fully active conformation. Instead they are exploring the dynamic transition of the A-Loop. This is challenging precisely because the process is so dynamic, but it is extremely important. I thus think that this paper provides important insights for how other kinases navigate the positioning of the Activation Loop. The fact that many mutations in many kinases are localized to this A-Loop that begins with the DFG motif and ends with the P-site emphasizes the critical importance of this dynamic region. The paper, however, is difficult to read.

Specific Concerns:

- 1. Clarification of steps.** The overlying message of the manuscript is that there are three steps to the activation process for Src once the inhibitory C-terminal phosphate is removed. The first, as demonstrated by the results described here, is phosphorylation of Y416 on the activation loop, which is fast and mediated by a cis-auto-phosphorylation. The second is the dynamic reorganization of the Activation Loop, which the authors seem to have trapped in an intermediate state in their crystal structure. The third is the trans-auto-phosphorylation of the C-terminal tail. The authors need to clarify this as their description of the second step in the discussion is not very clear.
- 2. Crosstalk.** The authors capture cross talk between the C-terminal tail and the AL which has not been described before. Src is more active, for example, when Y509 in the C-tail is mutated to Phe. They point out the discrepancies with Csk being the sole mechanism for phosphorylation of Tyr530, which is the accepted dogma. Is auto-phosphorylation of Y530 valid in the presence of Csk? And what about the phosphatase for pY530? Is this known and is it in close proximity? These processes will all be competing in cells. The fact that dual phosphorylated Src is found in cells argues that crosstalk between the two sites could be physiologically important.
- 3. Tyr/Phe mutants.** In addition, a Tyr-to-Phe mutation is not entirely convincing because the OH may also play a positive or a negative role. In particular, the Phe would be more hydrophobic and might better fill a hydrophobic pocket and lock it into place. Does the OH on the Tyr416 make hydrogen bonds to anything when Src is in its inhibited state?
- 4. Surrogate substrates.** Why do the authors not use a real substrate instead of dead version of FGFR and the Ret Receptor, which are thought to be activators of Src? These receptors are actually activators of Src in cells, not substrates; however their Activation Loops serve as surrogate substrates. It would be interesting to characterize a physiological substrate, but this is probably beyond the scope of this paper. The Activation Loop would likely need to be in a fully "Out" conformation to facilitate transfer to a heterologous substrate. However, physiological substrates of Src such as focal adhesion proteins are all localized together with Src at the plasma membrane, and this co-localization could help to stabilize the "Out" conformation of the A Loop.
- 5. Crystal structure.** This is a structure of only the kinase domain in the presence of a type II inhibitor so it is in an open conformation, not a closed conformation. It is not clear that the two Src molecules trapped in the crystal lattice are biologically relevant given that the two flanking domains, SH2 and SH3 are missing; however, it is likely that the authors have trapped one of many "intermediate states". They do show how the hydrophobic cap that usually surrounds the adenine ring of ATP is captured by the inhibitor. They could show this better in a figure, but it is described well in the text.
- 6.** The first step in the activation process is the cis-auto-phosphorylation of Y416, and this will cause the AL to flip into an "Out" conformation. The conformation trapped in their structure must be phosphorylated on the AL. Do the authors know that this is phosphorylated even though the AL is disordered in the structure? They know that the E310 is contacting K298 which means the bond that locks the AL into an "In" and inactive conformation (E310-422) is broken but the final assembly of the Activation Loop is still not complete as the R-spine is broken.
- 7.** What does pTyr216 do? When they look at the Y419F mutant in Figure 3A, there is a significant amount of overall pTyr but this does not seem to be accounted for by pY509. Is this due to Tyr216?

8. The interactions of the C-Tail with the Glycine-rich Loop (G-Loop) are intriguing. Although they point out that a palindromic sequence flanks Ty530, they do not explain why this is necessary. They also do not mutate the single sites or each PQ to see if this diminishes phosphorylation of the C-Tail. Would this not validate the importance of the palindromic sequence on both sides of Y530?

9. MD Simulations. Although this is well beyond the scope of this paper, it is likely that MD simulations might capture some of the interactions shown here. The authors should show the temperature factors for the two molecules in the crystal structures. The region around the nucleotide is probably very stable whereas the C-tail of the adjacent molecule docked into the active site of the kinase is likely to be more flexible. A segment of the A-Loop is totally disordered and the temperature factors would likely show that the AL in general is not very stable.

10. Inhibitor. Why did the authors use Ponatinib, which is a type II inhibitor? This is probably why they have trapped this intermediate conformation. Did they try other inhibitors? Type II inhibitors tend to stabilize an open conformation where the N and C-lobes are apart. Did they try a type I inhibitor which may have trapped a more fully close conformation where the activation loop is ordered? Actually there is only one structure in the PDB showing Src that is phosphorylated on its activation loop, and this is not reported in any publication. This is curious and suggests that the pY416 may be hard to trap.

11. Activation Loop. The authors say that the activation loop, which in this structure must be phosphorylated on Y416, is disordered. They should show this very carefully and indicate exactly which regions are disordered as well as the temperature factors. The Activation Loop begins with the DFG aspartic acid, and this Asp seems to be ordered. Is this correct? Immediately after the p-site (Y419) is the P+1 loop, which is thought to be important for recognizing the P+1 residue in the substrate although this may be different for tyrosine and serine kinases. This region is important as R422 is anchored to E310 in inactive Src, which causes the A-Loop to be in an inhibitory "In" conformation instead of in an "Out" conformation, which is associated with the active state. In this structure the Activation Loop and at least part of the subsequent Activation Segment is in a dynamic intermediate state. Which residues exactly are disordered? In the open and active conformation the HRD arginine typically interacts with the p-Site residue on the Activation Loop. Since the pY416 is not visible, what about the HRD arginine? If the HRD Arg is not anchored to the phosphate on the Activation Loop then the AL is clearly in some kind of transition state.

Reviewer #2 (Remarks to the Author):

In this manuscript, the authors bring forward evidence that c-Src-Y530 is a c-Src auto-phosphorylation site and c-Src-Y419 phosphorylation is required for c-Src-Y530 intermolecular phosphorylation. The authors also show that Tyr419 residing in the activation-loop undergoes fast kinetics and a cis-to-trans phosphorylation-switch in controlling Tyr530 auto-phosphorylation. Interestingly, Tyr419 also controls enzyme specificity and no-catalytic function as a substrate of c-Src. Impressively, c-Src c-terminal residues Y531 to 536 are required for c-Src Y530 and global auto-phosphorylation, and a mutant deleted of these residues seems to show dominant negative effect to the WT c-Src in auto-phosphorylation. These findings provide some new insights on the activation of c-Src from the view of structure. However, there still remain some concerns to be clarified:

Major concerns:

1. Except experiments relating to drosophila, the most of this work was done with purified protein in vitro. The c-Src auto-phosphorylation paradigm was, of course, supposed to occur in a scenario without upstream signaling regulation. I want to know how such auto-phosphorylation paradigm is regulated intracellularly by upstream signals to meet specific biological functions, in other words, the author should find a biological or pathological context under which such c-Src auto-phosphorylation paradigm play a role.

2. Given Src Tyr419 auto-phosphorylation and Tyr530 phosphorylation has already been well known to regulate c-Src enzymatic activity reversely, the authors should provide evidence that both Tyr419 phosphorylation and Tyr530 phosphorylation occurs simultaneously at the comparable level in mammalian cell or tissue to confirm their in vitro result that Tyr419 and Tyr530 are sequentially phosphorylated. For example, immunohistochemical staining of successive tissue sections with antibodies against phosphorylated Src-Y419 and Src-Y530, individually, or immunoprecipitation of c-Src followed by WB with antibodies against phosphorylated Src-Y419 and Src-Y530, individually.

3. According to the authors' results, c-Src-Y419 auto-phosphorylation is required for c-Src-Y530 phosphorylation. This should be confirmed by replacement of endogenous c-Src with c-Src-Y419F and further detection of c-Src-Y530 phosphorylation signal in mammalian cells.

4. The authors show that Src42A-Y400F almost completely rescued the drosophila phenotype caused by overexpression of wildtype Src. This is predictable, because such phenomenon has been previously observed by other groups (Pedraza et al., 2004; Putz, 2019; Forster and Luschnig, 2012; Nelson et al., 2012) and it is well known that Y419F mutation severely disrupts Src enzymatic activity. However, it is unknown whether phosphorylation of Y400 (corresponding to Y419F) is required for the phosphorylation of drosophila tyrosine corresponding to Y530F, so the phosphorylation levels of Y400 and the drosophila tyrosine corresponding to Y530F should be detected in drosophila with overexpression of wildtype Src42A, Src42A-Y400F or Y to F mutant of drosophila tyrosine corresponding to Y530F, together with the observation of corresponding phenotypes.

5. According to in vitro data, the authors concluded that cancer associated c-terminal deleted variants inhibit allosterically wildtype c-Src activity by a dominant negative effect. This proposal should be verified by overexpression of these c-terminal deleted variants in drosophila or mammalian cells to further observe the relating phenotypes.

6. It is very interesting that c-Src Y419 phosphorylation plays roles in its specificity toward substrate. What is the mechanism? Is Src-Y530 phosphorylation involved in such mechanism. This should be at least discussed.

Minor concerns:

1. There are many spelling mistakes in this manuscript. For example, 1) page 1, address section: "Tel.: +34 +34"; 2) a full stop is needed for the end of Summary; 3) page 8, line 15: "20.000 rpms"; 4) page11, line 4, "2450U/ml" should be "2450 U/mL"; 5) page19, title of paragraph2, "-inter-molecular", and so on.

2. Some scale bars are missed (Fig. 5A and 5B).

Reviewer #4 (Remarks to the Author):

The manuscript of Cuesta et al. addresses an important problem and delivers a significant amount of meticulous work and important results. However, I can't recommend it for publication in Nature Communications in its current form without significant revision. The main problem is that the current context of the problem is not presented properly and the results are not interpreted in a satisfactory way. The following are the particular points I would like to address.

p. 5. "In the current paradigm, phosphorylation at Tyr 530 by c-terminal Src kinase (CSK) is inhibitory, while phosphorylation on Tyr 419 in the activation loop is activating, although neither of these phosphorylation sites by themselves exerts full positive or negative regulatory control"

Here and further in the text, the authors describe the activation of Src, the role of dimerization, and SH2, and SH3 domains, however, the main point of the manuscript is the phosphorylation of Y419 and Y530. They refer the reader to two Roskoski reviews to understand what are the current problems of the current paradigm and mention that "the precise role of Y419 autophosphorylation... not yet fully understood". In my opinion, the double phosphorylation of Src has to be explained in detail as this is the main problem. The

role of CSK has to be clearly outlined and put in the context of the Src activation (see for example Fig.8 in Sun et al. 1998 PMID: 9794236). The distinction between cis and trans autophosphorylation should be made in the Introduction to put the results in the general context.

The structural aspect of the CSK phosphorylation of Y530 should be also introduced as the authors present their structure of the suggested autophosphorylation complex of Src. They mention Levinson's structure of the CSK/Src complex in the results but it has to be presented in the Introduction as the general picture of the problem looks rather vague and doesn't provide a comprehensive description of the current state.

p.17 "c-terminal Tyr 530 and SH2 Tyr 216 show slower rates of autophosphorylation reaching maximum levels at later time points (30-60 min)."

How these times are related to the CSK rates? Can authors comment on the biological relevance of slow (1-hour) rates of autophosphorylation?

p.22. Labels on Fig.S6D should be larger. There are panels F, G, and H in Figure S6 but they are not discussed in the text.

p.24. "This disposition is analogous to the one observed in the crystal structure of the Csk-c-Src complex where the interface was formed by the c-terminal aI helix with the preceding aH-aI loop of c-Src and the aD helix of Csk at the entrance of the active site (Levinson et al., 2008)"

As I mentioned earlier the CSK/Src complex should be mentioned in the Introduction as the reader needs to know that the structural context of Y530 phosphorylation is already known. The word "analogous" is very elusive and doesn't provide an important comparison between the two complexes. My understanding is that the CSK/Src complex is different as the C-terminal part of Src and the palindromic motif are not in contact with the G-loop of CSK. Levinson et al. argue that it is still possible that Y530 can be positioned at the active site of the enzyme. The authors do not discuss the difference and later in the Discussion say that their "crystal structure captures the arrangement of the substrate and active kinases during inter-molecular autophosphorylation" (p. 27). This is not necessarily true. The structure is compatible with the arrangement.

p.25. Throughout Fig. 7 proteins are labeled as WT, 528X, and 531X. It would be helpful for the reader to have the same labeling at the A panel for the C-terminal sequences.

"Unexpectedly, the c-Src 531X mutant, despite having a phosphorylatable c-terminal Tyr 530, lacked also the capacity to auto-phosphorylate on this residue, meaning that residues 531 to 536 are required for c-terminal auto-phosphorylation"

It is not really unexpected as usually, a peptide requires P+1 residue to be docked properly at the active site. In general, the text is full of subjective terms such as "unexpectedly" or "strikingly" (7 times). If the authors consider a particular result surprising they have to explain why or just avoid unnecessary emotional adjectives. What is surprising for one person may be not surprising for another.

p. 26 "A quick inspection of the results reveals that both c-terminal variants display comparable catalytic activities to the wild-type as indicated by total phosphorylation levels of the substrate (Fig. 7E)."

There are obvious differences between the activities. The authors should specify what they consider to be "comparable" and why this level of similarity is enough for their conclusions.

p.27 . "we identify a c-terminal palindromic phospho-motif containing Tyr 530 on the substrate molecule engaging the P-loop of the active kinase"

The fact that there is a palindromic sequence at the tail is indeed intriguing. Apparently, the authors consider it to be so important that they use it in the title of the manuscript.

However, the importance of the sequences being palindromic is not discussed. What is the author's view on this observation? How conserved is the palindrome? Can it be just a random fact?

"... provides evidence of the existence of a complex allosteric node at the c-terminus of c-Src modulating the crosstalk between substrate- and enzyme-acting kinases"

This is indeed a very interesting finding that is similar to other examples of allosteric communication between the "bottom" of the C-lobe and kinase activity. For example a set of Herman's papers on Tpk1 (PMIDs: 16751660,19364808). Presenting these results in a broader context would be beneficial for the reader.

p.28. "The third step is supported by our structural and biochemical data demonstrating that Tyr 530 is a de facto c-Src autophosphorylation site"

The fact that Y530 is an autophosphorylation site is well-known and should be clearly explained in the Introduction.

p. 29. There is a reference to the Graphical abstract but I couldn't find it in the Submission.

Reviewer #1 (Remarks to the Author):

This is a comprehensive analysis of the complex mechanisms that are associated with Src auto-phosphorylation. The authors elucidate some novel pathways and convincingly distinguish between cis-auto-phosphorylation of the Activation Loop (Y419) vs. trans-auto-phosphorylation of the C-terminal tail (Y530). The latter provides insights into how Src may trans-phosphorylate heterologous substrates although the authors focus here primarily on different mechanisms for auto-phosphorylation. The authors first describe an improved method for purification of de-phosphorylated Src in *E. coli* and then go on to do a rigorous kinetic analysis. They find that there is a rapid auto-phosphorylation of Y416 and then show that this is a cis-mediated process. A second slower rate is associated with phosphorylation of Y530 on the C-terminal tail. Although the authors are working primarily with an *in vitro* system and purified protein, they nevertheless demonstrate this cross talk between the C-tail and the Activation Loop (AL), and it is likely that similar cross talk does occur in cells since they are using a three-domain (3D-) construct, not just the kinase domain. This is surprising as the C-terminal site was thought to be phosphorylated in cells by Csk. They creatively use a kinase dead version of Src as a surrogate substrate and also test their model with several other surrogate protein substrates as well as with peptides. The data shows convincingly that Src is capable of auto-phosphorylating its C-tail and the mechanistic insights are valid. The physiological relevance of this phosphorylation in cells is still an open question but the fact that dually phosphorylated Src is found in cells suggests that this question of Y530 phosphorylation in cells is still not clear. And cis-phosphorylation of Y419 is a process that is shared by almost all protein kinases but has not been explored in such depth previously.

In addition to showing convincingly that auto-phosphorylation of the C-tail Y530 occurs in trans, in contrast to the Y416, which occurs by cis- auto-phosphorylation, the authors also provide a crystal structure to validate how the trans-auto-phosphorylation might work. Although the crystal structure in the presence of a type II inhibitor clearly represents a snapshot of one state in an ensemble of many likely intermediate conformations where the A-Loop is disordered, it does provide a plausible model showing how the C-tail could be presented to the active site of the adjacent kinase. The authors spend a great deal of time talking about the allosteric cross talk that takes place between the two sites - Y416 and Y530 where the A-Loop is toggling between an "In" (inactive) state and an "Out" (active) state. While much attention has focused on DFG In/Out and α C In/Out as part of the activation mechanism, less attention is focused on the A-Loop In/Out and, in particular, how the initial cis-auto-phosphorylation of Y416 on the A-Loop triggers the reorganization of the A-Loop to its "Out" state. This is the transition that the authors have captured here. While it is difficult to follow all of the data and it could be written

more clearly, I think that this is an important contribution not only to Src activation but also to this transition process that is relevant for all kinases.

The quality of this data is excellent, and these are challenging experiments to do. The work also introduces some important new concepts that will be important for the field to evaluate, as the results will undoubtedly be true for other tyrosine kinases and perhaps for all protein kinases. The logic is hard to follow in some places and the writing could be clearer. There are some serious concerns as cited below, but overall this work should be considered for publication after some significant revisions.

General Recommendation: The authors should definitely distinguish between heterologous protein substrates vs. auto-phosphorylation when the target site is tethered in close proximity to the active site. My own hypothesis is that trans-phosphorylation of a heterologous substrate will require the ordered A-Loop "Out" conformation with the A-Loop phosphorylated on Y419 whereas trans-auto-phosphorylation of the C-Tail will be able to utilize an intermediate state as has been captured in this study. It would simplify the reading of the paper if the authors focused on A-Loop "In" vs. A-Loop "Out" and on cis-auto-phosphorylation and trans-auto-phosphorylation vs. trans-phosphorylation of heterologous proteins that are co-localized with Src at the plasma membrane. Cis-auto-phosphorylation is easy because the site (Y419) is right there at the active site and simply requires the binding of the adenine ring of ATP. Phosphorylation of this site, however, pY419, then initiates the transition of the A-Loop from its "In" and inactive state to its "Out" and fully active state. In this study they have not captured that "Out" and fully active conformation. Instead they are exploring the dynamic transition of the A-Loop. This is challenging precisely because the process is so dynamic, but it is extremely important. I thus think that this paper provides important insights for how other kinases navigate the positioning of the Activation Loop. The fact that many mutations in many kinases are localized to this A-Loop that begins with the DFG motif and ends with the P-site emphasizes the critical importance of this dynamic region. The paper, however, is difficult to read.

Response: We are very grateful for the time taken to review carefully our work and the positive and constructive evaluation. We are very happy reviewer 1 acknowledges the quality of our data and the complexity and difficulty of the biochemical experiments here presented. We agree our work introduces some important new concepts that will be relevant to the protein kinase field. We agree with most of the comments and observations made by the referee, and following his/her indications we have made a significant effort to distinguish between heterologous protein substrates and auto-phosphorylation. I think the hypothesis of the reviewer is correct i.e. trans-phosphorylation of a heterologous substrate will require of an ordered, phosphorylated and extended activation loop "out" whereas trans-auto-phosphorylation of the c-tail could be able to utilize an intermediate state as we capture in our structure. We agree with the reviewer that focusing on activation loop in-versus-out would simplify the message and will help with the reading of the paper,

which we acknowledge it is complex at some parts of the manuscript (see modified graphical abstract summary on page 40)

Specific Concerns:

1. Clarification of steps. The overlying message of the manuscript is that there are three steps to the activation process for Src once the inhibitory C-terminal phosphate is removed. The first, as demonstrated by the results described here, is phosphorylation of Y416 on the activation loop, which is fast and mediated by a cis-auto-phosphorylation. The second is the dynamic reorganization of the Activation Loop, which the authors seem to have trapped in an intermediate state in their crystal structure. The third is the trans-auto-phosphorylation of the C-terminal tail. The authors need to clarify this as their description of the second step in the discussion is not very clear.

Response 1: As the reviewer clearly indicates, we postulate that there are three steps in the c-Src auto-phosphorylation process (in the absence of any CSK input). First, we observe a fast cis-auto-phosphorylation on the activation loop (Fig. 4A) which is compatible with the arrangement of the activation loop "in" configuration seen in a previously solved crystal structure (PDB ID: 2SRC). Second, the activation-loop auto-phosphorylation switches on time into an intermolecular (concentration dependent) component (Fig.4A) compatible with an activation loop "in-to-out" switch (intermediate state) that precedes the intermolecular auto-phosphorylation of the c-terminal segment on Tyr530 as shown on Fig. 6. This under our view is plausible as the activation loop must adopt an "out" configuration in order to allow a substrate e.g. the c-terminal segment or any other heterologous protein substrate into the active site for intermolecular phosphorylation. It is possible also that for the c-terminal intermolecular auto-phosphorylation an intermediate state is sufficient as indicated in our crystal structure (and very well pointed by the reviewer). We have clarified this in the discussion and also in the graphical abstract, where we have tried, following the reviewer indications, to simplify the message to keep it more focused (see revised manuscript, graphical abstract section on page 40).

2. Crosstalk. The authors capture cross talk between the c-terminal tail and the AL which has not been described before. Src is more active, for example, when Y530 in the C-tail is mutated to Phe. They point out the discrepancies with Csk being the sole mechanism for phosphorylation of Tyr530, which is the accepted dogma. Is auto-phosphorylation of Y530 valid in the presence of Csk? And what about the phosphatase for pY530? Is this known and is it in close proximity? These processes will all be competing in cells. The fact that dual phosphorylated Src is found in cells argues that crosstalk between the two sites could be physiologically important.

Response 2: We captured a sequential and coordinated crosstalk between the activation loop and the c-terminal segment, where Tyr419 phosphorylation comes first and is required prior Tyr 530 intermolecular auto-phosphorylation can take place

(see Fig 2, 3 and 4). The reviewer ask the question if auto-phosphorylation of Tyr 530 is valid in the presence of Csk? This is a very good question. Previous studies demonstrate that CSK mediated phosphorylation of c-terminal Tyr 530 inactivates c-Src only when the protein is not previously auto-phosphorylated (Sun et al., 1998). These previous findings highlighted that auto-phosphorylation may have a self-autonomous role and compete with CSK and other c-terminal targeting kinases.

Regarding phosphatases that could target Y530 for de-phosphorylation, as we indicate in the introduction, both protein-tyrosine phosphatase 1B (Bjorge et al., 2000) and SHP-1 (Somani et al., 1997) appears to be mayor players. As the referee points out, the fact that high levels of dual phosphorylated c-Src is found cancer cell lines (Sun et al., 1998) and in tumor samples from patients with aggressive cancers e.g. TNBC (Nelson et al., 2020) indicates that the crosstalk between the activation and c-terminal segment can be physiological relevant.

3. Tyr/Phe mutants. In addition, a Tyr-to-Phe mutation is not entirely convincing because the OH may also play a positive or a negative role. In particular, the Phe would be more hydrophobic and might better fill a hydrophobic pocket and lock it into place. Does the OH on the Tyr416 make hydrogen bonds to anything when Src is in its inhibited state?

Response 3: We apply systematically site directed mutagenesis on tyrosine phospho-sites by replacing them for a non-phosphorylable Phe. In this way, we keep the side chain integrity without the OH in order to minimize undesirable folding, conformational or steric effects on the protein by the replacement of the side chain. When we look at the crystal structure of the kinase domain of c-Src in an inhibited state (PDB ID: 2SRC) activation loop Tyr 419 is pointing towards the active site in close proximity to the HRD motif (4Å to R388, 4.8Å to D389 and 6.9Å to the gamma phosphate of the ATP. This is in contrast to the extended conformation of the activation loop with a phosphorylated Tyr419, which is solvent accessible (PDB ID1YI6)

4. Surrogate substrates. Why do the authors not use a real substrate instead of dead version of FGFR and the Ret Receptor, which are thought to be activators of Src? These receptors are actually activators of Src in cells, not substrates; however, their Activation Loops serve as surrogate substrates. It would be interesting to characterize a physiological substrate, but this is probably beyond the scope of this paper. The Activation Loop would likely need to be in a fully "Out" conformation to facilitate transfer to a heterologous substrate. However, physiological substrates of Src such as focal adhesion proteins are all localized together with Src at the plasma membrane, and this co-localization could help to stabilize the "Out" conformation of the A Loop.

Response 4: Here we used FAK as a *bona fide* c-Src susbtrate surrogate as well as catalytic impaired kinase domain of RET and c-Src (see Fig. 3C-F and Fig. 4B and C).

It is true that RET is able to phosphorylate and activate c-Src on the activation loop, but at the same time we have found that c-Src is able to phosphorylate RET on the activation loop too, to undergo a reciprocal and bidirectional phosphorylation (work in progress). In fact, RET activation loop Tyr 905 consensus aa sequence (EEDSYVKRS) is very similar to the optimal c-Src consensus substrate sequence (EEDVYESPP), see Fig. 3G and PhosphoSitePlus <https://www.phosphosite.org>). That is the rationale behind the use of such c-Src substrate surrogates. We agree with the reviewer that the activation loop would need to be in an extended fully “out” conformation to facilitate transfer to a heterologous substrate. As the reviewer points out, physiological substrates such as focal adhesion proteins are all localized together with c-Src at the plasma membrane, and this co-localization could help to stabilize the “out” conformation of the activation loop.

5. Crystal structure. This is a structure of only the kinase domain in the presence of a type II inhibitor so it is in an open conformation, not a closed conformation. It is not clear that the two Src molecules trapped in the crystal lattice are biologically relevant given that the two flanking domains, SH2 and SH3 are missing; however, it is likely that the authors have trapped one of many “intermediate states”. They do show how the hydrophobic cap that usually surrounds the adenine ring of ATP is captured by the inhibitor. They could show this better in a figure, but it is described well in the text.

Response 5: We have shown graphically on Fig. S6G the hydrophobic cap of the catalytic spine that surrounds the inhibitor in the active site in full details.

6. The first step in the activation process is the cis-auto-phosphorylation of Y416, and this will cause the AL to flip into an “Out” conformation. The conformation trapped in their structure must be phosphorylated on the AL. Do the authors know that this is phosphorylated even though the AL is disordered in the structure? They know that the E310 is contacting K298 which means the bond that locks the AL into an “In” and inactive conformation (E310-422) is broken but the final assembly of the Activation Loop is still not complete as the R-spine is broken.

Response 6: We agree with the reviewer that the first step in the activation process is the cis-auto-phosphorylation of Tyr 419, and this will cause the activation loop to flip into an “out” conformation. However, we do not have evidence the activation loop in the crystal structure is phosphorylated. What we can see is the catalytic salt bridge formed in our crystal structure, which means the bond that locks the activation loop into an “in” and inactive conformation (E310-422) is broken but the final assembly of the activation Loop is still not complete as the R-spine is broken and the DFG is in an out configuration

7. What does pTyr216 do? When they look at the Y419F mutant in Figure 3A, there

is a significant amount of overall pTyr but this does not seem to be accounted for by pY509. Is this due to Tyr216?

Response 7: This is an interesting observation, as both c-terminal Tyr530 and Tyr 216 (SH2) are located in close proximity (13 Å) in the inactive c-Src crystal structure (PDB ID: 2SRC, see image below). We show that Y216 is a bona fide auto-phosphorylation site and that it appears to be up-regulated in the case of the Y530F mutants, possibly as a compensatory feedback (see Fig.S3 and S4). MS data confirms that the Y530F is hyperphosphorylated compared with c-Src wild-type (see Fig. 3, 4 and Fig. S4 A-F)

Figure. Cartoon representation of c-Src (PDB ID: 2SRC) showing the spatial location of c-terminal Y530 and Y216 (SH2) and their distance.

8. The interactions of the C-Tail with the Glycine-rich Loop (G-Loop) are intriguing. Although they point out that a palindromic sequence flanks Ty530, they do not explain why this is necessary. They also do not mutate the single sites or each PQ to see if this diminishes phosphorylation of the C-Tail. Would this not validate the importance of the palindromic sequence on both sides of Y530?

Response 8: We agree these are unexpected and very interesting findings. What we postulate based on our crystal structure and the biochemical data presented on Fig. 7 is that there are contacts between residues of this palindrome sequence at the c-terminus of the substrate molecule and the G-loop of the active molecule that are required for the correct positioning of the c-terminal segment in order to undergo intermolecular phosphorylation. If we perturb the palindrome sequence e.g. 531X mutant (PQYstop) the protein lacks the capacity to auto-phosphorylate on the c-terminal Tyr530 and have a slower phospho-tyrosine activity in auto-phosphorylation experiments (Fig. 7B). Interestingly this deleterious effect is not caused by a direct effect on the net catalytic activity, as both wild-type and the 531X constructs have similar activities in phosphorylating an exogenous substrate surrogate and peptide (see Fig. 7C and E). It is not caused either by a perturbation of the consensus phospho-motif sequence as the enzymatic activity towards a wild-type c-terminal

peptide containing Y530 (STEPQYQPGEN) and a peptide derived from the 531X construct (STEPQY) is exactly the same (see this new data on Fig. S9). We have made three more mutants targeting the palindromic and c-terminal segment residues. First, we generated a 529X (PAstop) mutant lacking Tyr 530. This mutant appears functionally normal in terms of phospho-tyrosine activity both auto- and phosphorylation towards a peptide substrate, and behaves overall as the 528X construct (see new Fig. S8). Based on MD data simulations (see also answer to next point 9), the c-terminal residue E521 makes important contacts with R463 that also contributes for the proper positioning of the c-terminal segment prior intermolecular phosphorylation. A c-Src E521A mutant display slower auto-phosphorylation kinetics specifically on Tyr530 auto-phosphorylation (see new Fig. S8). Furthermore, we also generated a PAYPA mutant. This is a critical mutant lacking Q529 and 531 side-chains while keeping intact Y530. This mutant specifically lacks the capacity to auto-phosphorylate on c-terminal Y530 (total loss) despite being functional in terms of auto-phosphorylation and catalytic activity (see new Fig. S10). These results indicate once more that the perturbation of the contacts between the c-terminal palindromic motif of the substrate molecule and the G-loop of the active kinase are required for the auto-phosphorylation in trans of the c-terminal Tyr 530.

9. MD Simulations. Although this is well beyond the scope of this paper, it is likely that MD simulations might capture some of the interactions shown here. The authors should show the temperature factors for the two molecules in the crystal structures. The region around the nucleotide is probably very stable whereas the C-tail of the adjacent molecule docked into the active site of the kinase is likely to be more flexible. A segment of the A-Loop is totally disordered and the temperature factors would likely show that the AL in general is not very stable.

Response 9: We agree with the reviewer that MD simulations can capture some of the interactions describe in our study. In fact, we have made long range (1.2 μ s) MD simulations based on our crystal structure to capture how the c-terminal segment of the substrate molecule can accommodate on the active site of the active kinase molecule to undergo intermolecular phosphorylation (see new data Fig. 6E) in several replicates (see also Fig. S7). In particular we measured the distances between the γ -phosphate group of the ATP in the active site of the active kinase and the side-chain of Tyr 530 on the substrate molecule and found stable interactions during the 1.2 μ s simulation run. These data confirm the crystal structure and biochemical results presented in our study supporting further our interpretation and conclusion.

10. Inhibitor. Why did the authors use Ponatinib, which is a type II inhibitor? This is probably why they have trapped this intermediate conformation. Did they try other inhibitors? Type II inhibitors tend to stabilize an open conformation where the N and C-lobes are apart. Did they try a type I inhibitor which may have trapped a more fully close conformation where the activation loop is ordered? Actually, there is only one

structure in the PDB showing Src that is phosphorylated on its activation loop, and this is not reported in any publication. This is curious and suggests that the pY416 may be hard to trap.

Response 10: We used Ponatinib because the crystal structure of c-Src in complex with this compound was not known by the time we set up the experiments. We have tried other type-I inhibitors and nucleotides with full-length constructs of WT and Y/F mutants of both the activation and c-terminal segment. Unfortunately, in the case of the full-length constructs the crystals we obtained diffracted poorly. We wanted initially to capture the conformational landscape of the active site, but it must be so dynamic and versatile that it is very difficult to trap. We agree with the reviewer by looking at the ratio between c-Src crystal structures solved so far with a phosphorylated and un-phosphorylated activation loop Tyr.

11. Activation Loop. The authors say that the activation loop, which in this structure must be phosphorylated on Y416, is disordered. They should show this very carefully and indicate exactly which regions are disordered as well as the temperature factors. The Activation Loop begins with the DFG aspartic acid, and this Asp seems to be ordered. Is this correct? Immediately after the p-site (Y419) is the P+1 loop, which is thought to be important for recognizing the P+1 residue in the substrate although this may be different for tyrosine and serine kinases. This region is important as R422 is anchored to E310 in inactive Src, which causes the A-Loop to be in an inhibitory "In" conformation instead of in an "Out" conformation, which is associated with the active state. In this structure the Activation Loop and at least part of the subsequent Activation Segment is in a dynamic intermediate state. Which residues exactly are disordered? In the open and active conformation, the HRD arginine typically interacts with the p-Site residue on the Activation Loop. Since the pY416 is not visible, what about the HRD arginine? If the HRD Arg is not anchored to the phosphate on the Activation Loop then the AL is clearly in some kind of transition state.

Response 11: As the reviewer correctly indicates, the activation loop in our crystal structure is disordered. The activation-loop missing aa sequence goes from residues 410, just after the DFG-motif (aa 407-9) which is ordered and out, to residue 426, so R422 is not captured in the structure and we cannot provide evidence about the activation loop being in an out configuration (active state). However, as we mentioned on **response 6** we do not have evidence the activation loop in the crystal structure is phosphorylated.

Reviewer #2 (Remarks to the Author):

In this manuscript, the authors bring forward evidence that c-Src-Y530 is a c-Src auto-phosphorylation site and c-Src-Y419 phosphorylation is required for c-Src-Y530 intermolecular phosphorylation. The authors also show that Tyr419 residing in the activation-loop undergoes fast kinetics and a cis-to-trans phosphorylation-switch in controlling Tyr530 auto-phosphorylation. Interestingly, Tyr419 also controls enzyme specificity and no-catalytic function as a substrate of c-Src. Impressively, c-Src c-terminal residues Y531 to 536 are required for c-Src Y530 and global auto-phosphorylation, and a mutant deleted of these residues seems to show dominant negative effect to the WT c-Src in auto-phosphorylation. These findings provide some new insights on the activation of c-Src from the view of structure. However, there still remain some concerns to be clarified:

Major concerns:

1. Except experiments relating to drosophila, the most of this work was done with purified protein in vitro. The c-Src auto-phosphorylation paradigm was, of course, supposed to occur in a scenario without upstream signaling regulation. I want to know how such auto-phosphorylation paradigm is regulated intracellularly by upstream signals to meet specific biological functions, in other words, the author should find a biological or pathological context under which such c-Src auto-phosphorylation paradigm play a role.

Response 1: In our manuscript we provide evidence of a self-autonomous auto-phosphorylation mechanisms that controls the functional and conformational landscape of c-Src, and how cancer associated c-terminal deleted variants perturb this mechanism. Further, we found that a double phosphorylated c-Src on both the activation loop and c-terminal segment is still an active protein in vitro (Fig. S11A). The fact that dual phosphorylated Src members are found in cancer cells (Hardwick and Sefton, 1997; Nika et al., 2010; Sun et al., 1998) and that in samples from TNBC patients elevated levels of both Y419 and Y530 phosphorylation are found (Nelson et al., 2020) argues that the crosstalk between the two sites could be physiologically important. These previous finding together with our data provides a plausible scenario for an alternative paradigm for the mechanism of c-Src activation driven by an auto-phosphorylation mechanism.

Previous studies demonstrate that CSK mediated phosphorylation of c-terminal Tyr 530 inactivates c-Src only when the protein is not previously auto-phosphorylated (Sun et al., 1998). These previous findings highlighted that auto-phosphorylation may compete with CSK and other c-terminal targeting kinases. Our work dissects the auto-phosphorylation mechanisms in an isolated system with no CSK input. As already indicated, the fact that high levels of dual phosphorylated c-Src is found in

cancer cells patient samples of some types of aggressive cancers e.g. TNBC (Nelson et al., 2020) indicates this can be physiologically relevant.

As we state in the discussion: Our work provides answers to two important questions for a better understanding of the role of c-Src in human cancers. On one hand, hyperactivation of c-Src in human cancers might result from the perturbation of auto-phosphorylation and allosteric control. Large-scale genomic sequencing projects indicate that gene amplification and activating mutations in c-Src do not play a significant role in human tumor biology (Bailey et al., 2018; Curtis et al., 2012). In fact, paradoxical high levels of phosphorylated c-Src in both activation and c-terminal segments are found in aggressive cancer types such TNBC (Nelson et al., 2020) and NSCLC (unpublished). We have data supporting these findings where a previously phosphorylated c-Src protein (90-120 min) is active and able to phosphorylate an intact substrate surrogate with faster kinetics than the non-phosphorylated protein (Fig. S11A). On the other hand, c-Src may function independent of CSK, which may not always play an anti-oncogenic role through the negative regulation of c-Src in carcinogenesis. In this line, elevated expression of CSK in human cancer cell lines appears to correspond to elevated c-Src protein-tyrosine kinase activity (Watanabe et al., 1995) e.g. CSK is widely expressed in HT-29 and SW620 cells that contain high-levels of active c-Src (Li et al., 1996). Furthermore, genetically engineered mouse models showed that in the absence of CSK, c-Src phosphorylation at Tyr 527 in vivo was reduced to about 20-50% of the level in wild-type cells. These results additionally supported the notion that auto-phosphorylation and/or other kinase could be driving c-terminal tyrosine phosphorylation (Imamoto and Soriano, 1993; MacAuley et al., 1993)

In addition, we find three critical scientific evidences that supports the notion that c-Src intrinsic activity independent of direct upstream inputs can play an important role in cancer cells:

i) c-Src can phosphorylate and activate directly RTKs such as EGFR and RET. For example, it is well established that EGFR is phosphorylated directly on Tyr 845 by c-Src and that this phosphorylation event is important for the biology of cancer cells, in particular DNA synthesis, and malignant cell proliferation (for a comprehensive review see, (Sato, 2013). In the same line, it has been shown that inactive RET mutations that prevent cellular proliferation could be rescued by c-Src and v-Src promoted intermolecular phosphorylation (Kato et al., 2002). We have extensive data showing that c-Src can directly phosphorylate RET on the activation loop both in vitro and in cell-based assays (work in progress)

ii) Implications in the mechanism of resistance to anti-cancer therapies. c-Src have been shown to play an important role in the response to anti-EGFR therapies, and co-treatment with c-Src inhibitors re-sensitizes resistant cancer cells (for a comprehensive review see (Belli et al., 2020).

iii) In the same line, the paradoxical activation of c-Src and its downstream phosphorylation cascade induced by Src-targeted and RTK-targeted kinase inhibitors have huge implications also in the mechanism of resistance to many TKIs and anticancer-therapies (Higuchi et al., 2021). In the paper by Higushi and co-workers, they highlight a self-autonomous mechanism induced by the binding of an ATP-competitor inhibitor that relieves c-Src auto-inhibition and favors an open state compatible with substrate binding (e.g. FAK) and phosphorylation. Whether this model applies also to auto-phosphorylation needs to be further explored.

2. Given Src Tyr419 auto-phosphorylation and Tyr530 phosphorylation has already been well known to regulate c-Src enzymatic activity reversely, the authors should provide evidence that both Tyr419 phosphorylation and Tyr530 phosphorylation occurs simultaneously at the comparable level in mammalian cell or tissue to confirm their in vitro result that Tyr419 and Tyr530 are sequentially phosphorylated. For example, immunohistochemical staining of successive tissue sections with antibodies against phosphorylated Src-Y419 and Src-Y530, individually, or immunoprecipitation of c-Src followed by WB with antibodies against phosphorylated Src-Y419 and Src-Y530, individually.

Response 2: We have tested a set of different cell lines: TPC1, MZ-CRC-1, KELLY, LAN1, TT, SK-N-AS, MCF7, MCF7CTED, BT459, MDA-MB-231), some of them also treated with LOXO-292 (1 mM, 90 min) in WBs analyses using total c-Src and Tyr419 and Tyr530 phospho-specific antibodies and found that some cell lines display simultaneous high levels of Tyr530 and Tyr 419 phosphorylation such KELLY, LAN1, TT and TPC1-LOXO-292-treated cells (see Fig. S11B). These observations were further supported by HEK293T cells ectopically expressing increasing amounts of c-Src display a dose-dependent increase in the phosphorylation levels of both c-terminal Tyr530 and activation loop Tyr419 (see also response 5)

3. According to the authors' results, c-Src-Y419 auto-phosphorylation is required for c-Src-Y530 phosphorylation. This should be confirmed by replacement of endogenous c-Src with c-Src-Y419F and further detection of c-Src-Y530 phosphorylation signal in mammalian cells.

Response 3: Our findings are based on in vitro experiments using recombinant protein and a highly controlled experimental system. What we conclude is that Y419 is required for Y530 auto-phosphorylation. In a cellular system, we cannot assure that another kinase could be targeting c-terminal tyrosine in the absence of activation loop Y419. To monitor auto-phosphorylation in a cell-based context is extremely difficult, as it is not possible to control the input of other kinases that can interact and phosphorylate c-Src in trans nor the fact that ATP is already present in the system. From our previous experience with other protein kinases e.g. RET in previous published work (see Plaza-Menacho et al Mol Cell 2014, and Cell Reports 2016); the same question was also raised by reviewers. We found that in order to track temporal

kinetics of the auto-phosphorylation reaction precisely and accurately, the only possible way was in vitro using highly pure, monodisperse and un-phosphorylated protein in a fully controlled test tube reaction in the presence of exogenous MgCl₂ and ATP in a time-course experiment.

4. The authors show that Src42A-Y400F almost completely rescued the drosophila phenotype caused by overexpression of wildtype Src. This is predictable, because such phenomenon has been previously observed by other groups (Pedraza et al., 2004; Putz, 2019; Forster and Luschnig, 2012; Nelson et al., 2012) and it is well known that Y419F mutation severely disrupts Src enzymatic activity. However, it is unknown whether phosphorylation of Y400 (corresponding to Y419F) is required for the phosphorylation of drosophila tyrosine corresponding to Y530F, so the phosphorylation levels of Y400 and the drosophila tyrosine corresponding to Y530F should be detected in drosophila with overexpression of wildtype Src42A, Src42A-Y400F or Y to F mutant of drosophila tyrosine corresponding to Y530F, together with the observation of corresponding phenotypes.

Response 4: As far as we are aware, in none of the scientific works cited by the referee the Src42A activation loop Y400F mutant allele was used nor evaluated. They used in all of them mainly a constitutively active (CA) allele with a c-terminal Y/F mutation (Y511F), a kinase dead allele (KM) with a mutation in the catalytic lysine (K276M), and a loss of function F80 allele, but not an activation loop mutant allele (see below). Furthermore, we do not present any phenotypic rescue related to Src overexpression (extra-long tubes) with the Y400F mutant allele. In fact, we rescue the loss of function of Src (F80 allele) by the Y400F mutant (see Fig. 5D and E).

In any case the second part of the comment is an interesting question raised by the reviewer. We have tried to monitor dSrc42A auto-phosphorylation using commercially available c-Src phospho-specific antibodies (phospho-Src Tyr419 (D49G4) CST #6943 and phospho-Src Tyr 530 (ThermoFisher 44- 662G and CST#2105), but they appears not to recognize activation-loop nor c-terminal segment tyrosines in dSrc. We thought initially, this was due to their consensus sequence differing from their human/chicken/mouse homolog sequence from which the antibody was raised against. However, we just found that the phospho-Src Tyr 419 Thermo # 44-660G can recognize dSrc phosphorylated on the activation loop. In any case, as we indicated in the manuscript, we are currently working on the structural and molecular characterization of a full length dSrc42A construct, and the dissection of the molecular mechanisms of auto-activation will be submitted and published as a follow up story due to the time frame given for the resubmission.

A detailed list of the constructs used in the works cited by the referee:

Pedraza et al., 2004: Wild-type src64, src42 and csk cDNA were cloned into the pUAST vector for P-element transformation (Brand and Perrimon, 1993). The Src64YF and

Src42YF constitutively active mutants were generated by PCR, confirmed by sequencing and cloned into the pUAST vector. UAS-p21 and GMRGal4 were generous gifts from I Hariharan and M Freeman respectively (Pedraza et al., 2004)

Nelson et al., 2012: The specific alleles used for images are as follows: Src4226-1, Src64KO, nrv223B, dDaamEx68, vermKG07819, and Atp1R2. The UAS-Src42WT and UAS-Src64 lines are described in 22. The UAS-Src42KM transgene contains a K276M mutation that abolishes catalytic activity as described in (Shindo et al., 2008) (Nelson et al., 2012)

Forster and Luschnig, 2012: Src42A26-1, UAS-Src42A.CA (Y511F), UAS-Src42A.KM (K295M) and aa variants UAS-Src42A, UAS-Src42A, UAS-Src42A (Forster and Luschnig, 2012)

Putz et al., 2019: To investigate whether elevated, constitutively-active or kinase-deficient SRC disrupts the N-Cad AJ between R3 and R4, UAS-transgenic lines for Src42, constitutively active Src42CA, kinase deficient Src42KD and Src64 were used (Putz, 2019)

5. According to in vitro data, the authors concluded that cancer associated c-terminal deleted variants inhibit allosterically wildtype c-Src activity by a dominant negative effect. This proposal should be verified by overexpression of these c-terminal deleted variants in drosophila or mammalian cells to further observe the relating phenotypes.

Response 5: We agree with the referee this is an important point. For this purpose, we have conducted experiments in HEK293T cells in which we ectopically expressed increasing amounts of c-Src full-length WT and a c-terminal deleted variant 531X to evaluate the phosphorylation levels of c-Src (see figure below panel A). In addition, we performed combinatorial experiments in which a fixed amount of c-Src WT plasmid was co-expressed with increasing amounts of the 531X to evaluate the dominant negative effect by the mutant allele (see figure below panel B).

First, we overexpress in HEK293T cells increasing amounts of c-Src WT and we observed a dose dependent increase in the levels of Tyr 419, Tyr 530 and total phospho-tyrosine (see panel A, below). This contrasted with the lower levels of total and Tyr419 phosphorylation levels showed by cells expressing the same amount of the c-Src 531X mutant. These data indicate a potential detrimental effect by the 531X mutant. In the case of Tyr 530 phosphorylation, as expected from our in vitro experiments with recombinant protein, there was no detectable Tyr 530 phosphorylation signal by the c-terminal palindromic deleted variant (see panel A, below). As a control, we used antibodies that detected total c-Src protein, one of them was directed against a C-terminal epitope (present only in the WT c-Src). The second antibody directed to an N-terminal epitope resulted in the identification of both WT and the c-terminal deleted 531X variant (see panel A, below)

Next, we co-expressed c-Src WT and 531X plasmids in HEK293T cells to capture the dominant negative effect of the c-terminal deleted variant on the wild-type allele. In particular, we co-expressed a fixed amount of c-Src WT (0.5 μg) with increasing amounts of c-Src 531X (0, 0.5, 1, 2 and 4 μg) and we could not observe any transactivation effect by the WT construct. In fact, we found a detrimental impact not only on the levels of Tyr530 phosphorylation (as we expected) but also on the phosphorylation levels of the activation loop Tyr419 when compared with the 531X construct alone. These data suggest a potential dominant negative effect of the 531X allele over the wild-type (see panel B, below). However, we acknowledge that to monitor auto-phosphorylation and its allosteric regulation in a cell-based system is very complicated and makes us to take cautiously the message about this point in the manuscript. We have opted to remove the conclusive statement in the highlights (see page 3), and see modified text also on the introduction on page 6.

Figure. HEK293T cells were transfected with increasing amounts of the indicated plasmids using Fugene following manufacturer's protocol. Whole cell lysates were subjected to WBs using the indicated antibodies.

6. It is very interesting that c-Src Y419 phosphorylation plays roles in its specificity toward substrate. What is the mechanism? Is Src-Y530 phosphorylation involved in such mechanism. This should be at least discussed.

Response 6:

We agree with the reviewer this is an interesting and unexpected finding. The data we present in this manuscript related to the function of the activation loop is dual. On one hand we show that Y419 is required for the substrate molecule to be presented into the active site of an active kinase molecule for intermolecular phosphorylation (non-catalytic role, see Fig. 4C). On the other hand, we show that a non-phosphorylatable activation loop Y419F mutant is catalytically competent but is

able to phosphorylate preferentially a set of substrates e.g. FAK and Src versus others e.g. RET (see Fig. 3B and C-F). Interestingly, when we look at the consensus sequence of an optimal c-Src substrate, it is very similar to the aa sequence of the activation loop of RET GLSDVYEEDEYVKRSQ with a EED motif at position -4, -3 and -2 from Y905 (see Fig. 4G). How can the activation loop control directly the selection of the substrate molecule? We speculate that the dynamic nature of the activation loop will allow to reach the P+1 substrate binding site and sterically impede or modulate substrate access into the active site. Alternatively, the activation loop may prime protein-protein contacts with the substrate molecule for adequate positioning into the substrate binding site prior catalysis.

Minor concerns:

1. There are many spelling mistakes in this manuscript. For example, 1) page 1, address section: "Tel.: +34 +34"; 2) a full stop is needed for the end of Summary; 3) page 8, line 15: "20.000 rpms"; 4) page11, line 4, "2450U/ml" should be "2450 U/mL"; 5) page19, title of paragraph2, "-inter-molecular", and so on.
2. Some scale bars are missed (Fig. 5A and 5B).

Response:

We have corrected the spelling mistakes throughout the text, we thank the reviewer for checking this.

Reviewer #4 (Remarks to the Author):

The manuscript of Cuesta et al. addresses an important problem and delivers a significant amount of meticulous work and important results. However, I can't recommend it for publication in Nature Communications in its current form without

significant revision. The main problem is that the current context of the problem is not presented properly and the results are not interpreted in a satisfactory way. The following are the particular points I would like to address.

p. 5. "In the current paradigm, phosphorylation at Tyr 530 by c-terminal Src kinase (CSK) is inhibitory, while phosphorylation on Tyr 419 in the activation loop is activating, although neither of these phosphorylation sites by themselves exerts full positive or negative regulatory control"

Here and further in the text, the authors describe the activation of Src, the role of dimerization, and SH2, and SH3 domains, however, the main point of the manuscript is the phosphorylation of Y419 and Y530. They refer the reader to two Roskoski reviews to understand what are the current problems of the current paradigm and mention that "the precise role of Y419 autophosphorylation... not yet fully understood". In my opinion, the double phosphorylation of Src has to be explained in detail as this is the main problem. The role of CSK has to be clearly outlined and put in the context of the Src activation (see for example Fig.8 in Sun et al. 1998 PMID: 9794236). The distinction between cis and trans autophosphorylation should be made in the Introduction to put the results in the general context. The structural aspect of the CSK phosphorylation of Y530 should be also introduced as the authors present their structure of the suggested autophosphorylation complex of Src. They mention Levinson's structure of the CSK/Src complex in the results but it has to be presented in the Introduction as the general picture of the problem looks rather vague and doesn't provide a comprehensive description of the current state.

Response:

We agree with the reviewer comments and suggestions and we have now included in the revised version of the manuscript more details and information about the highlighted points as listed below:

i) We have made more emphasis on the double phosphorylated c-Src paradigm by providing new data on the phosphorylation (see revised introduction page 5, and discussion on pages 33 and new supplemental figure S11B)

ii) We have clearly outlined in the introduction the role of CSK on c-Src Tyr530 phosphorylation including and mentioning Levinson's Csk-c-Src crystal structure. Further, on new Fig. 2A, we have included a graphical depiction on the role of CSK in the current model for c-Src regulation (see introduction page 5 and new Fig. 1A)

iii) We have made a distinction between cis and trans autophosphorylation at the introduction, and we have introduced as well as the distinction on catalytic versus non-catalytic functions in protein kinases (see introduction page 5) to put the results in the general context.

p.17 "c-terminal Tyr 530 and SH2 Tyr 216 show slower rates of autophosphorylation reaching maximum levels at later time points (30-60 min)."

How these times are related to the CSK rates? Can authors comment on the biological relevance of slow (1-hour) rates of autophosphorylation?

Response p.17: In the manuscript by Advani et al. (Advani et al., 2017) the authors look at the phosphorylation of c-Src Tyr530 in vitro by recombinant Csk and Chk, and found that Csk was a better catalyst of the reaction compared with Chk. To reach this conclusion, in these experiments the authors did not performed time-course phosphorylation assays but instead they measured the activity at a final time point of 20 min. Although, these data apparently indicate higher preference of Csk for c-terminal Tyr 530 phosphorylation, without observing the saturation kinetics of the reaction it is difficult to compare the rates of phosphorylation by Csk and those related to c-Src auto-phosphorylation when the assays have been conducted under different conditions.

The slower timing in the process of c-terminal Tyr 530 phosphorylation compared with the activation loop correspond to a sequential and coordinated series of phosphorylation and conformational events, where fast activation loop auto-phosphorylation in cis switches into a trans-phosphorylation component that precedes at later stages the c-terminal intermolecular phosphorylation. In our case, despite we refer to the time required for the auto-phosphorylation reaction to reach saturation to 30-60 min, we are able to detect already significant phosphorylation levels of Tyr 530 from min 15 or earlier by means of WBs using phospho-specific antibodies and tandem LC/MS-MS. We speculate that the slower kinetics of c-terminal auto-phosphorylation can account for a different functional facet where at early time points activation loop phosphorylation is required for the protein to adopt a catalytically competent open conformation and select for the substrate to be phosphorylated i.e. substrate specificity. At later stages, a fully auto-phosphorylated c-Src molecule on both the activation loop and c-terminal segment may adopt a close conformation but presenting the activation loop in an "out" configuration (non-catalytic function) to favor protein-protein interaction with substrates and or other kinases that regulate c-Src function in trans for complex assembly prior phosphorylation (see Fig.S11A)

p.22. Labels on Fig.S6D should be larger. There are panels F, G, and H in Figure S6 but they are not discussed in the text.

Response p.12: The labels are now larger from size 17 to size 20, and the complete set of panels are discussed in the text (see supplemental information, Fig. S6). Please note that in order to focus our work and to avoid redundancy we have removed some structural information related to Fig. S6.

p.24. "This disposition is analogous to the one observed in the crystal structure of the Csk-c-Src complex where the interface was formed by the c-terminal α I helix with the preceding α H- α I loop of c-Src and the α D helix of Csk at the entrance of the active site (Levinson et al., 2008)".

As I mentioned earlier the CSK/Src complex should be mentioned in the Introduction as the reader needs to know that the structural context of Y530 phosphorylation is already known. The word "analogous" is very elusive and doesn't provide an important comparison between the two complexes. My understanding is that the CSK/Src complex is different as the C-terminal part of Src and the palindromic motif are not in contact with the G-loop of CSK. Levinson et al. argue that it is still possible that Y530 can be positioned at the active site of the enzyme. The authors do not discuss the difference and later in the Discussion say that their "crystal structure captures the arrangement of the substrate and active kinases during inter-molecular autophosphorylation" (p. 27). This is not necessarily true. The structure is compatible with the arrangement.

Response p.24: We agree with the reviewer comments and suggestions. In the revised version, the crystal structure of the CSK and c-Src complex is now mentioned in the introduction (see previous response on remarks to the author, and page 5 revised introduction). The reviewer is correct and we should have specified the differences between the crystal structure of the CSK-c-Src complex (Levinson et al. 2008) and our crystal structure. We have now included in the results section the following modified text (see page 25).

"This disposition is similar to some extent to the one observed in the crystal structure of the Csk/c-Src complex where the interface was formed by the c-terminal α I helix with the preceding α H- α I loop of c-Src and the α D helix of Csk at the entrance of the active site (Levinson et al., 2008)". The main difference however between the two crystal structures is that in the CSK/c-Src complex the c-terminal segment of the c-Src substrate molecule is not in contact with the G-loop of CSK. In any case, both structures are compatible with the c-terminal Y530 to be positioned for ready entry into the active site of the enzyme".

In the discussion we have modified the text according to the reviewer indications as follow (see page 27)

"Our crystal structure is compatible with the arrangement of the substrate and active kinases during inter-molecular autophosphorylation"

p.25. Throughout Fig. 7 proteins are labeled as WT, 528X, and 531X. It would be

helpful for the reader to have the same labeling at the A panel for the C-terminal sequences.

Response p.25: We have now made the nomenclature consistent among the different constructs for the c-terminal sequences on Figure 7

"Unexpectedly, the c-Src 531X mutant, despite having a phosphorylatable c-terminal Tyr 530, lacked also the capacity to auto-phosphorylate on this residue, meaning that residues 531 to 536 are required for c-terminal auto-phosphorylation"

It is not really unexpected as usually; a peptide requires P+1 residue to be docked properly at the active site. In general, the text is full of subjective terms such as "unexpectedly" or "strikingly" (7 times). If the authors consider a particular result surprising they have to explain why or just avoid unnecessary emotional adjectives. What is surprising for one person may be not surprising for another.

Response: We have done enzymatic assays using peptides derived from an intact c-terminal Tyr 530 (STEPQYQPGEN) sequence and also the 531X sequence lacking the right part of the palindrome (STEPQY). We found that both constructs WT and 531X efficiently phosphorylate the 531X sequence-derived STEPQY peptide with no significant differences in their K_M nor V_{max} values. These data conclusively demonstrate that the lack of auto-phosphorylation on Tyr 530 seen by the 531X mutant is not driven by a perturbation of the consensus phospho-motif sequence surrounding Tyr 530 nor differences in their intact catalytic activities. These data have been included in the revised manuscript supplemental Fig. S9.

We get the point of the reviewer, we have tried to avoid the use of unnecessary emotional adjectives in the revised manuscript, thanks for the advice. We have tried to avoid the overuse of e.g. strikingly see page 20 and 22.

p. 26 "A quick inspection of the results reveals that both c-terminal variants display comparable catalytic activities to the wild-type as indicated by total phosphorylation levels of the substrate (Fig. 7E)."

There are obvious differences between the activities. The authors should specify what they consider to be "comparable" and why this level of similarity is enough for their conclusions.

Response p.26: In Fig. 7E we measured and referred to the capacity of active c-Src 3D constructs (WT and c-terminal variants) to phosphorylate a catalytically impaired Src KD K298M mutant as a substrate surrogate. In this assay we aim to measure the intact enzymatic capacity toward the substrate. We found no differences between the 3 constructs. It is true that in this experiment we can still capture the capacity of c-Src 3D to auto-phosphorylate (see upper 50 kDa band) in the presence of the substrate. In this case we can see indeed differences in the levels of auto-

phosphorylation (comparable and consistent with data from Fig. 7B), but in this particular experiment what we are measuring is the activity towards the substrate (KD K298M, see lower 32 kDa band in the gel), and this activity does not change among the different constructs tested.

p.27. "we identify a c-terminal palindromic phospho-motif containing Tyr 530 on the substrate molecule engaging the P-loop of the active kinase"

The fact that there is a palindromic sequence at the tail is indeed intriguing. Apparently, the authors consider it to be so important that they use it in the title of the manuscript. However, the importance of the sequences being palindromic is not discussed. What is the author's view on this observation? How conserved is the palindrome? Can it be just a random fact?

Response p.27: We agree with the reviewer that the findings of the palindromic phospho-motif are intriguing and interesting. This motif is highly conserved in all the vertebrate c-Src orthologs from human, chicken, rat to xenopus and zebrafish; However, it is not conserved in invertebrates (*Drosophila*, *C. elegans*, *Hydra*, and bacteria *E. fluviatilis*, and *M. ovata*), see Segawa et al 2007. We do not believe this is a random fact due to the functional, structural and genetic evidence provided that altogether indicate this is an important functional motif. In addition to the 531X (PQYstop) mutant we have generated a palindrome PAYAP mutant. This is a critical mutant lacking Q529 and Q531 side-chains while keeping intact Y530. This mutant lacks the capacity to auto-phosphorylate specifically on c-terminal Y530 despite being functional in terms of auto-phosphorylation and catalytic activity towards peptide substrates (see new Fig. SX10). In total we have made three new mutants targeting the palindromic and c-terminal segment residues, namely a 529X and D521A. See also response 8 to reviewer 1. These results indicate once that the perturbation of the contacts between the c-terminal palindromic motif of the substrate molecule and the G-loop of the active kinase are required for the intermolecular auto-phosphorylation of c-terminal Tyr 530.

"... provides evidence of the existence of a complex allosteric node at the c-terminus of c-Src modulating the crosstalk between substrate- and enzyme-acting kinases"

This is indeed a very interesting finding that is similar to other examples of allosteric communication between the "bottom" of the C-lobe and kinase activity. For example, a set of Herman's papers on Tpk1 (PMIDs: 16751660,19364808). Presenting these results in a broader context would be beneficial for the reader.

Response: As suggested by the reviewer, in order to broaden the context of our findings we have included in the discussion information about other examples of asymmetric allosteric communication in protein kinases. In particular, we refer to the

EGFR paradigm of asymmetric crosstalk between the C-lobe of the activating kinase and the N-lobe of the allosterically activated i.e. receiver kinase (Zhang et al., 2006). In this context the C-lobe of the 'activator' contacts the N-lobe of the 'receiver' at points of the α C-helix, the β 4/ β 5 loop and an N-terminal extension of the N-lobe (PDB ID: 2GS6). Formation of this asymmetric kinase dimer induces allosteric changes in the N-lobe extension of the receiver kinase leading to the conformational changes in its α C-helix and A-loop required to switch on the activated state (Zhang et al., 2006). This arrangement is analogous to the one observed for the CDK2/CyclinA complex (Jeffrey et al., 1995), see also PDB ID:1FIN. There are also examples of allosteric communication between the C-lobe of the active kinase and the substrate molecule mediated by the α H and α I helices interface of the kinase e.g. Tpk1 (Deminoff et al., 2006; Deminoff et al., 2009), see discussion page 34.

References

- Advani, G., Lim, Y.C., Catimel, B., Lio, D.S.S., Ng, N.L.Y., Chueh, A.C., Tran, M., Anasir, M.I., Verkade, H., Zhu, H.J., et al. (2017). Csk-homologous kinase (Chk) is an efficient inhibitor of Src-family kinases but a poor catalyst of phosphorylation of their C-terminal regulatory tyrosine. *Cell Commun Signal* 15, 29.
- Bailey, M.H., Tokheim, C., Porta-Pardo, E., Sengupta, S., Bertrand, D., Weerasinghe, A., Colaprico, A., Wendl, M.C., Kim, J., Reardon, B., et al. (2018). Comprehensive Characterization of Cancer Driver Genes and Mutations. *Cell* 174, 1034-1035.
- Belli, S., Esposito, D., Servetto, A., Pesapane, A., Formisano, L., and Bianco, R. (2020). c-Src and EGFR Inhibition in Molecular Cancer Therapy: What Else Can We Improve? *Cancers (Basel)* 12.
- Bjorge, J.D., Pang, A., and Fujita, D.J. (2000). Identification of protein-tyrosine phosphatase 1B as the major tyrosine phosphatase activity capable of dephosphorylating and activating c-Src in several human breast cancer cell lines. *J Biol Chem* 275, 41439-41446.
- Brand, A.H., and Perrimon, N. (1993). Targeted gene expression as a means of altering cell fates and generating dominant phenotypes. *Development* 118, 401-415.
- Curtis, C., Shah, S.P., Chin, S.F., Turashvili, G., Rueda, O.M., Dunning, M.J., Speed, D., Lynch, A.G., Samarajiwa, S., Yuan, Y., et al. (2012). The genomic and transcriptomic architecture of 2,000 breast tumours reveals novel subgroups. *Nature* 486, 346-352.
- Deminoff, S.J., Howard, S.C., Hester, A., Warner, S., and Herman, P.K. (2006). Using substrate-binding variants of the cAMP-dependent protein kinase to identify novel targets and a kinase domain important for substrate interactions in *Saccharomyces cerevisiae*. *Genetics* 173, 1909-1917.
- Deminoff, S.J., Ramachandran, V., and Herman, P.K. (2009). Distal recognition sites in substrates are required for efficient phosphorylation by the cAMP-dependent protein kinase. *Genetics* 182, 529-539.
- Forster, D., and Luschnig, S. (2012). Src42A-dependent polarized cell shape changes mediate epithelial tube elongation in *Drosophila*. *Nat Cell Biol* 14, 526-534.
- Hardwick, J.S., and Sefton, B.M. (1997). The activated form of the Lck tyrosine protein kinase in cells exposed to hydrogen peroxide is phosphorylated at both Tyr-394 and Tyr-505. *J Biol Chem* 272, 25429-25432.
- Higuchi, M., Ishiyama, K., Maruoka, M., Kanamori, R., Takaori-Kondo, A., and Watanabe, N. (2021). Paradoxical activation of c-Src as a drug-resistant mechanism. *Cell Rep* 34, 108876.

Imamoto, A., and Soriano, P. (1993). Disruption of the *csk* gene, encoding a negative regulator of Src family tyrosine kinases, leads to neural tube defects and embryonic lethality in mice. *Cell* 73, 1117-1124.

Jeffrey, P.D., Russo, A.A., Polyak, K., Gibbs, E., Hurwitz, J., Massague, J., and Pavletich, N.P. (1995). Mechanism of CDK activation revealed by the structure of a cyclinA-CDK2 complex. *Nature* 376, 313-320.

Kato, M., Takeda, K., Kawamoto, Y., Iwashita, T., Akhand, A.A., Senga, T., Yamamoto, M., Sobue, G., Hamaguchi, M., Takahashi, M., et al. (2002). Repair by Src kinase of function-impaired RET with multiple endocrine neoplasia type 2A mutation with substitutions of tyrosines in the COOH-terminal kinase domain for phenylalanine. *Cancer Res* 62, 2414-2422.

Levinson, N.M., Seeliger, M.A., Cole, P.A., and Kuriyan, J. (2008). Structural basis for the recognition of c-Src by its inactivator Csk. *Cell* 134, 124-134.

Li, S., Ke, S., and Budde, R.J. (1996). The C-terminal Src kinase (Csk) is widely expressed, active in HT-29 cells that contain activated Src, and its expression is downregulated in butyrate-treated SW620 cells. *Cell Biol Int* 20, 723-729.

MacAuley, A., Okada, M., Nada, S., Nakagawa, H., and Cooper, J.A. (1993). Phosphorylation of Src mutants at Tyr 527 in fibroblasts does not correlate with in vitro phosphorylation by CSK. *Oncogene* 8, 117-124.

Nelson, K.S., Khan, Z., Molnar, I., Mihaly, J., Kaschube, M., and Beitel, G.J. (2012). *Drosophila* Src regulates anisotropic apical surface growth to control epithelial tube size. *Nat Cell Biol* 14, 518-525.

Nelson, L.J., Wright, H.J., Dinh, N.B., Nguyen, K.D., Razorenova, O.V., and Heinemann, F.S. (2020). Src Kinase Is Biphosphorylated at Y416/Y527 and Activates the CUB-Domain Containing Protein 1/Protein Kinase C delta Pathway in a Subset of Triple-Negative Breast Cancers. *Am J Pathol* 190, 484-502.

Nika, K., Soldani, C., Salek, M., Paster, W., Gray, A., Etzensperger, R., Fugger, L., Polzella, P., Cerundolo, V., Dushek, O., et al. (2010). Constitutively active Lck kinase in T cells drives antigen receptor signal transduction. *Immunity* 32, 766-777.

Pedraza, L.G., Stewart, R.A., Li, D.M., and Xu, T. (2004). *Drosophila* Src-family kinases function with Csk to regulate cell proliferation and apoptosis. *Oncogene* 23, 4754-4762.

Putz, S.M. (2019). Mbt/PAK4 together with SRC modulates N-Cadherin adherens junctions in the developing *Drosophila* eye. *Biol Open* 8.

Sato, K. (2013). Cellular functions regulated by phosphorylation of EGFR on Tyr845. *Int J Mol Sci* 14, 10761-10790.

Shindo, M., Wada, H., Kaido, M., Tateno, M., Aigaki, T., Tsuda, L., and Hayashi, S. (2008). Dual function of Src in the maintenance of adherens junctions during tracheal epithelial morphogenesis. *Development* 135, 1355-1364.

Somani, A.K., Bignon, J.S., Mills, G.B., Siminovitch, K.A., and Branch, D.R. (1997). Src kinase activity is regulated by the SHP-1 protein-tyrosine phosphatase. *J Biol Chem* 272, 21113-21119.

Sun, G., Sharma, A.K., and Budde, R.J. (1998). Autophosphorylation of Src and Yes blocks their inactivation by Csk phosphorylation. *Oncogene* 17, 1587-1595.

Watanabe, N., Matsuda, S., Kuramochi, S., Tsuzuku, J., Yamamoto, T., and Endo, K. (1995). Expression of C-terminal src kinase in human colorectal cancer cell lines. *Jpn J Clin Oncol* 25, 5-9.

Zhang, X., Gureasko, J., Shen, K., Cole, P.A., and Kuriyan, J. (2006). An allosteric mechanism for activation of the kinase domain of epidermal growth factor receptor. *Cell* 125, 1137-1149.

Reviewer #3 comments

In this article, authors Cuesta et al., show how auto-phosphorylation of c-Src undergoes conformational changes to bring about two different roles.

Two major stumbles of this manuscript is

1. Line numbers are added to indicate the comments
2. The flow of the manuscript is not uniform. It is a mixture of too many writing styles. It is extremely distracting and hard to understand. So authors should make the style of writing in the manuscript to be same and uniform.

Minor Comments

In Summary:

The lines "In line with this, we visualize by X-ray crystallography a snapshot of Tyr 530 ready entry prior catalysis" needs to be restructured.

Similarly the lines " We show that c-terminal residue ... the active kinase" has to be broken into two lines.

In Introduction

pg.4 – line 7 – the word "codified" should be replaced as "codes for"

pg.5- line 20 – the word "corrupted" does not sound scientific and needs to be replaced.

In Materials and Methods

Plasmids

word "codifying" that is occurring in several places needs to be replaced.

pg.7- line 1- word "codifying" needs to be changed to "containing";

pg.7- line 3 and 11- word "codifying" needs to be changed to "coding"

pg.7- line 8- word "codifying" needs to be changed to "codes for"

pg. 7-lines 7, 8,10 in the word "drosophila" D has to be changed to capital letter

pg. 8 line 15- "20.000" rpm should be 20,000 rpm

pg.10 line 14- " factor o" must be "factor of"

pg.12 line 8 – " ion target values were 3e6 for MS (maximum IT of 25 ms) and 1e5 " – does IT mean ion target, if so IT should be written in brackets as ion target (IT) values 6 in 3e6 should be made as superscript; 5 in 1e5 should be made as superscript

pg.12 line 18- E. coli should be italicized

pg.12 line 21-please expand FDR

Results

pg.19 line 9- "catalytically" should be changed as "catalytic"

pg.21 line 9-"catalytically impaired" should be changed as "catalytic activity impaired"

pg.22 line 2 Drosophila melanogaster should be italicized not just Drosophila

pg.25 line 12- " we use in time course auto-phosphorylation assays "must be "we performed a time course of auto-phosphorylation assay using a"

Discussion

pg.27 line 4- "According with this" should be either made as "Accordingly" or replace with some other word

pg.27 line 20 - words "with: i) results demonstrating that should be rephrased as "with result demonstrates that: (i)"

pg.27 line 25 - statement "play a compartmentalization rather than a functional or catalytic role" should be changed as play a compartmentalization role rather than a functional or catalytic one.

pg.28 line 6 – Line "Auto-phosphorylation...c-terminal Tyr-530" needs rephrasing!

pg.28 line 9- it is mentioned as "See graphic abstract"! Where is it??

pg.28 line 14- it is mentioned as "Step-2". Which one/Where is Step-1?

pg.28 line 12-Line "The first step...Try 419" is too big and complex to understand ! It has to be made into smaller statements and understandable.

Pg.29 line 5 – It begins with Third, where is second?? Only First of all is mentioned in pg.27 line 11.

But second point is missing!

Major Comments

The concept and functional role of c-Src has already been shown using several techniques (although authors have missed to refer to those work here), in this study using crystal structures the authors have captured only the events of changes in configuration from closed to open structures, their status of phosphorylation associated on Tyr 416 and Tyr 530 residues. They have shown the importance of the allosteric phospho-switch and their possible implications. This might shed some light on investigation of cancerous cells in future.

pg.22-Section 6:

The work on Drosophila is not done in the right way to address the role of c-Src mutant in A-loop. Src products (be Src64B which is not used in this study or Src42A used in this study) in Drosophila are implicated more in cytoskeletal regulation in addition to its role as oncogenes. The authors have performed the overexpression studies using the mutant construct in non-specific tissues which may not reflect the normal or appropriate physiological conditions. The observed phenotypic effects may be due to the role in regulation of cytoskeletal activities and nothing to do with its oncogenic role. Instead authors must have used Drosophila model of cancer with any combination of activated mutations (Example Ras Protein or Notch components! Refer Flybase for work on the Drosophila models) to show the functional activity of the mutant construct. Such studies are already shown using Src64B as well as for Src42A. Authors should refer to Flybase for all the work using flies to design experiments. Overexpression studies using UAS/GAL4 system is completely misleading. Authors must have generated clones in activated Ras85D and/or activated Src42A to understand the functional role. They can also use RAS-CSK system to understand the SRC activity. Otherwise this simple overexpression and rescue experiments may not reflect the appropriate function. Because titration levels of CSK activity on CSK targets cannot be assessed or ruled out by changes in phenotypes. They must be due to non-specific activity. Since this entire work shown in Section 6 needs more work to be performed to prove the functional role of Src42A mutant. This section 6 may be either removed or atleast moved to supplementary data.

This work is not yet ready for publication in Nature publication in this stage.

Reviewer #1

Remarks to the Author:

This is a comprehensive analysis of the complex mechanisms that are associated with Src auto-phosphorylation. The authors elucidate some novel pathways and convincingly distinguish between cis-auto-phosphorylation of the Activation Loop (Y419) vs. trans-auto-phosphorylation of the C-terminal tail (Y530). The latter provides insights into how Src may trans-phosphorylate heterologous substrates although the authors focus here primarily on different mechanisms for auto-phosphorylation. The authors first describe an improved method for purification of de-phosphorylated Src in *E. coli* and then go on to do a rigorous kinetic analysis. They find that there is a rapid auto-phosphorylation of Y416 and then show that this is a cis-mediated process. A second slower rate is associated with phosphorylation of Y530 on the C-terminal tail. Although the authors are working primarily with an in vitro system and purified protein, they nevertheless demonstrate this cross talk between the C-tail and the Activation Loop (AL), and it is likely that similar cross talk does occur in cells since they are using a three-domain (3D-) construct, not just the kinase domain. This is surprising as the C-terminal site was thought to be phosphorylated in cells by Csk. They creatively use a kinase dead version of Src as a surrogate substrate and also test their model with several other surrogate protein substrates as well as with peptides. The data shows convincingly that Src is capable of auto-phosphorylating its C-tail and the mechanistic insights are valid. The physiological relevance of this phosphorylation in cells is still an open question but the fact that dually phosphorylated Src is found in cells suggests that this question of Y530 phosphorylation in cells is still not clear. And cis-phosphorylation of Y419 is a process that is shared by almost all protein kinases but has not been explored in such depth previously.

In addition to showing convincingly that auto-phosphorylation of the C-tail Y530 occurs in trans, in contrast to the Y416, which occurs by cis auto-phosphorylation, the authors also provide a crystal structure to validate how the trans-auto-phosphorylation might work. Although the crystal structure in the presence of a type II inhibitor clearly represents a snapshot of one state in an ensemble of many likely intermediate conformations where the A-Loop is disordered, it does provide a plausible model showing how the C-tail could be presented to the active site of the adjacent kinase. The authors spend a great deal of time talking about the allosteric cross talk that takes place between the two sites - Y416 and Y530 where the A-Loop is toggling between an "In" (inactive) state and an "Out" (active) state. While much attention has focused on DFG In/Out and α C In/Out as part of the activation mechanism, less attention is focused on the A-Loop In/Out and, in particular, how the initial cis-auto-phosphorylation of Y416 on the A-Loop triggers the reorganization of the A-Loop to its "Out" state. This is the transition that the authors have captured here. While it is difficult to follow all of the data and it could be written more clearly, I think that this is an important contribution not only to Src activation but also to this transition process that is relevant for all kinases.

The quality of this data is excellent, and these are challenging experiments to do. The work also introduces some important new concepts that will be important for the

field to evaluate, as the results will undoubtedly be true for other tyrosine kinases and perhaps for all protein kinases. The logic is hard to follow in some places and the writing could be clearer. There are some serious concerns as cited below, but overall this work should be considered for publication after some significant revisions.

General Recommendation: The authors should definitely distinguish between heterologous protein substrates vs. auto-phosphorylation when the target site is tethered in close proximity to the active site. My own hypothesis is that trans-phosphorylation of a heterologous substrate will require the ordered A-Loop “Out” conformation with the A-Loop phosphorylated on Y419 whereas trans-auto-phosphorylation of the C-Tail will be able to utilize an intermediate state as has been captured in this study. It would simplify the reading of the paper if the authors focused on A-Loop “In” vs. A-Loop “Out” and on cis-auto-phosphorylation and trans-auto-phosphorylation vs. trans-phosphorylation of heterologous proteins that are co-localized with Src at the plasma membrane. Cis-auto-phosphorylation is easy because the site (Y419) is right there at the active site and simply requires the binding of the adenine ring of ATP. Phosphorylation of this site, however, pY419, then initiates the transition of the A-Loop from its “In” and inactive state to its “Out” and fully active state. In this study they have not captured that “Out” and fully active conformation. Instead they are exploring the dynamic transition of the A-Loop. This is challenging precisely because the process is so dynamic, but it is extremely important. I thus think that this paper provides important insights for how other kinases navigate the positioning of the Activation Loop. The fact that many mutations in many kinases are localized to this A-Loop that begins with the DFG motif and ends with the P-site emphasizes the critical importance of this dynamic region. The paper, however, is difficult to read.

Response: We are very grateful for the time taken to review carefully our work and the positive and constructive evaluation. We are very happy reviewer 1 acknowledges the quality of our data and the complexity and difficulty of the biochemical experiments here presented. We agree our work introduces some important new concepts that will be relevant to the protein kinase field. We agree with most of the comments and observations made by the referee, and following his/her indications we have made a significant effort to distinguish between heterologous protein substrates and auto-phosphorylation. I think the hypothesis of the reviewer is correct i.e. trans-phosphorylation of a heterologous substrate will require of an ordered, phosphorylated and extended activation loop “out” whereas trans-auto-phosphorylation of the c-tail could be able to utilize an intermediate state as we capture in our structure. We agree with the reviewer that focusing on activation loop in-versus-out would simplify the message and will help with the reading of the paper, which we acknowledge it is complex at some parts of the manuscript (see modified graphical abstract summary on page 40)

Specific Concerns:

1. Clarification of steps. The overlying message of the manuscript is that there are three steps to the activation process for Src once the inhibitory C-terminal phosphate is removed. The first, as demonstrated by the results described here, is phosphorylation of Y416 on the activation loop, which is fast and mediated by a cis-auto-phosphorylation. The second is the dynamic reorganization of the Activation Loop, which the authors seem to have trapped in an intermediate state in their crystal structure. The third is the trans-auto-phosphorylation of the C-terminal tail. The

authors need to clarify this as their description of the second step in the discussion is not very clear.

Response 1: As the reviewer clearly indicates, we postulate that there are three steps in the c-Src auto-phosphorylation process (in the absence of any CSK input). First, we observe a fast cis-auto-phosphorylation on the activation loop (Fig. 4A) which is compatible with the arrangement of the activation loop “in” configuration seen in a previously solved crystal structure (PDB ID: 2SRC). Second, the activation-loop auto-phosphorylation switches on time into an intermolecular (concentration dependent) component (Fig.4A) compatible with an activation loop “in-to-out” switch (intermediate state) that precedes the intermolecular auto-phosphorylation of the c-terminal segment on Tyr530 as shown on Fig. 6. This under our view is plausible as the activation loop must adopt an “out” configuration in order to allow a substrate e.g. the c-terminal segment or any other heterologous protein substrate into the active site for intermolecular phosphorylation. It is possible also that for the c-terminal intermolecular auto-phosphorylation an intermediate state is sufficient as indicated in our crystal structure (and very well pointed by the reviewer). We have clarified this in the discussion and also in the graphical abstract, where we have tried, following the reviewer indications, to simplify the message to keep it more focused (see revised manuscript, graphical abstract section on page 40).

2. Crosstalk. The authors capture cross talk between the c-terminal tail and the AL which has not been described before. Src is more active, for example, when Y530 in the C-tail is mutated to Phe. They point out the discrepancies with Csk being the sole mechanism for phosphorylation of Tyr530, which is the accepted dogma. Is auto-phosphorylation of Y530 valid in the presence of Csk? And what about the phosphatase for pY530? Is this known and is it in close proximity? These processes will all be competing in cells. The fact that dual phosphorylated Src is found in cells argues that crosstalk between the two sites could be physiologically important.

Response 2: We captured a sequential and coordinated crosstalk between the activation loop and the c-terminal segment, where Tyr419 phosphorylation comes first and is required prior Tyr 530 intermolecular auto-phosphorylation can take place (see Fig 2, 3 and 4). The reviewer ask the question if auto-phosphorylation of Tyr 530 is valid in the presence of Csk? This is a very good question. Previous studies demonstrate that CSK mediated phosphorylation of c-terminal Tyr 530 inactivates c-Src only when the protein is not previously auto-phosphorylated (Sun et al., 1998). These previous findings highlighted that auto-phosphorylation may have a self-autonomous role and compete with CSK and other c-terminal targeting kinases.

Regarding phosphatases that could target Y530 for de-phosphorylation, as we indicate in the introduction, both protein-tyrosine phosphatase 1B (Bjorge et al., 2000) and SHP-1 (Somani et al., 1997) appears to be mayor players. As the referee points out, the fact that high levels of dual phosphorylated c-Src is found cancer cell lines (Sun et al., 1998) and in tumor samples from patients with aggressive cancers e.g. TNBC (Nelson et al., 2020) indicates that the crosstalk between the activation and c-terminal segment can be physiological relevant.

3. Tyr/Phe mutants. In addition, a Tyr-to-Phe mutation is not entirely convincing because the OH may also play a positive or a negative role. In particular, the Phe would be more hydrophobic and might better fill a hydrophobic pocket and lock it into

place. Does the OH on the Tyr416 make hydrogen bonds to anything when Src is in its inhibited state?

Response 3: We apply systematically site directed mutagenesis on tyrosine phospho-sites by replacing them for a non-phosphorylatable Phe. In this way, we keep the side chain integrity without the OH in order to minimize undesirable folding, conformational or steric effects on the protein by the replacement of the side chain. When we look at the crystal structure of the kinase domain of c-Src in an inhibited state (PDB ID: 2SRC) activation loop Tyr 419 is pointing towards the active site in close proximity to the HRD motif (4Å to R388, 4.8Å to D389 and 6.9Å to the gamma phosphate of the ATP. This is in contrast to the extended conformation of the activation loop with a phosphorylated Tyr419, which is solvent accessible (PDB ID1YI6)

4. Surrogate substrates. Why do the authors not use a real substrate instead of dead version of FGFR and the Ret Receptor, which are thought to be activators of Src? These receptors are actually activators of Src in cells, not substrates; however, their Activation Loops serve as surrogate substrates. It would be interesting to characterize a physiological substrate, but this is probably beyond the scope of this paper. The Activation Loop would likely need to be in a fully “Out” conformation to facilitate transfer to a heterologous substrate. However, physiological substrates of Src such as focal adhesion proteins are all localized together with Src at the plasma membrane, and this co-localization could help to stabilize the “Out” conformation of the A Loop.

Response 4: Here we used FAK as a *bona fide* c-Src substrate surrogate as well as catalytic impaired kinase domain of RET and c-Src (see Fig. 3C-F and Fig. 4B and C). It is true that RET is able to phosphorylate and activate c-Src on the activation loop, but at the same time we have found that c-Src is able to phosphorylate RET on the activation loop too, to undergo a reciprocal and bidirectional phosphorylation (work in progress). In fact, RET activation loop Tyr 905 consensus aa sequence (EEDSYVKRS) is very similar to the optimal c-Src consensus substrate sequence (EEDVYESPP), see Fig. 3G and PhosphoSitePlus <https://www.phosphosite.org>). That is the rationale behind the use of such c-Src substrate surrogates. We agree with the reviewer that the activation loop would need to be in an extended fully “out” conformation to facilitate transfer to a heterologous substrate. As the reviewer points out, physiological substrates such as focal adhesion proteins are all localized together with c-Src at the plasma membrane, and this co-localization could help to stabilize the “out” conformation of the activation loop.

5. Crystal structure. This is a structure of only the kinase domain in the presence of a type II inhibitor so it is in an open conformation, not a closed conformation. It is not clear that the two Src molecules trapped in the crystal lattice are biologically relevant given that the two flanking domains, SH2 and SH3 are missing; however, it is likely that the authors have trapped one of many “intermediate states”. They do show how the hydrophobic cap that usually surrounds the adenine ring of ATP is captured by the inhibitor. They could show this better in a figure, but it is described well in the text.

Response 5: We have shown graphically on Fig. S6G the hydrophobic cap of the catalytic spine that surrounds the inhibitor in the active site in full details.

6. The first step in the activation process is the cis-auto-phosphorylation of Y416, and this will cause the AL to flip into an “Out” conformation. The conformation trapped in their structure must be phosphorylated on the AL. Do the authors know that this is phosphorylated even though the AL is disordered in the structure? They know that the E310 is contacting K298 which means the bond that locks the AL into an “In” and inactive conformation (E310-422) is broken but the final assembly of the Activation Loop is still not complete as the R-spine is broken.

Response 6: We agree with the reviewer that the first step in the activation process is the cis-auto-phosphorylation of Tyr 419, and this will cause the activation loop to flip into an “out” conformation. However, we do not have evidence the activation loop in the crystal structure is phosphorylated. What we can see is the catalytic salt bridge formed in our crystal structure, which means the bond that locks the activation loop into an “in” and inactive conformation (E310-422) is broken but the final assembly of the activation Loop is still not complete as the R-spine is broken and the DFG is in an out configuration

7. What does pTyr216 do? When they look at the Y419F mutant in Figure 3A, there is a significant amount of overall pTyr but this does not seem to be accounted for by pY509. Is this due to Tyr216?

Response 7: This is an interesting observation, as both c-terminal Tyr530 and Tyr 216 (SH2) are located in close proximity (13 Å) in the inactive c-Src crystal structure (PDB ID: 2SRC, see image below). We show that Y216 is a bona fide auto-phosphorylation site and that it appears to be up-regulated in the case of the Y530F mutants, possibly as a compensatory feedback (see Fig.S3 and S4). MS data confirms that the Y530F is hyperphosphorylated compared with c-Src wild-type (see Fig. 3, 4 and Fig. S4 A-F)

Figure. Cartoon representation of c-Src (PDB ID: 2SRC) showing the spatial location of c-terminal Y530 and Y216 (SH2) and their distance.

8. The interactions of the C-Tail with the Glycine-rich Loop (G-Loop) are intriguing. Although they point out that a palindromic sequence flanks Ty530, they do not explain why this is necessary. They also do not mutate the single sites or each PQ to see if this diminishes phosphorylation of the C-Tail. Would this not validate the importance of the palindromic sequence on both sides of Y530?

Response 8: We agree these are unexpected and very interesting findings. What we postulate based on our crystal structure and the biochemical data presented on Fig. 7 is that there are contacts between residues of this palindrome sequence at the c-terminus of the substrate molecule and the G-loop of the active molecule that are required for the correct positioning of the c-terminal segment in order to undergo intermolecular phosphorylation. If we perturb the palindrome sequence e.g. 531X mutant (PQYstop) the protein lacks the capacity to auto-phosphorylate on the c-terminal Tyr530 and have a slower phospho-tyrosine activity in auto-phosphorylation experiments (Fig. 7B). Interestingly this deleterious effect is not caused by a direct effect on the net catalytic activity, as both wild-type and the 531X constructs have similar activities in phosphorylating an exogenous substrate surrogate and peptide (see Fig. 7C and E). It is not caused either by a perturbation of the consensus phospho-motif sequence as the enzymatic activity towards a wild-type c-terminal peptide containing Y530 (STEPQYQPGEN) and a peptide derived from the 531X construct (STEPQY) is exactly the same (see this new data on Fig. S9). We have made three more mutants targeting the palindromic and c-terminal segment residues. First, we generated a 529X (PAstop) mutant lacking Tyr 530. This mutant appears functionally normal in terms of phospho-tyrosine activity both auto- and phosphorylation towards a peptide substrate, and behaves overall as the 528X construct (see new Fig. S8). Based on MD data simulations (see also answer to next point 9), the c-terminal residue E521 makes important contacts with R463 that also contributes for the proper positioning of the c-terminal segment prior intermolecular phosphorylation. A c-Src E521A mutant display slower auto-phosphorylation kinetics specifically on Tyr530 auto-phosphorylation (see new Fig. S8). Furthermore, we also generated a PAYPA mutant. This is a critical mutant lacking Q529 and 531 side-chains while keeping intact Y530. This mutant specifically lacks the capacity to auto-phosphorylate on c-terminal Y530 (total loss) despite being functional in terms of auto-phosphorylation and catalytic activity (see new Fig. S10). These results indicate once more that the perturbation of the contacts between the c-terminal palindromic motif of the substrate molecule and the G-loop of the active kinase are required for the auto-phosphorylation in trans of the c-terminal Tyr 530.

9. MD Simulations. Although this is well beyond the scope of this paper, it is likely that MD simulations might capture some of the interactions shown here. The authors should show the temperature factors for the two molecules in the crystal structures. The region around the nucleotide is probably very stable whereas the C-tail of the adjacent molecule docked into the active site of the kinase is likely to be more flexible. A segment of the A-Loop is totally disordered and the temperature factors would likely show that the AL in general is not very stable.

Response 9: We agree with the reviewer that MD simulations can capture some of the interactions describe in our study. In fact, we have made long range (1.2 μ s) MD simulations based on our crystal structure to capture how the c-terminal segment of the substrate molecule can accommodate on the active site of the active kinase molecule to undergo intermolecular phosphorylation (see new data Fig. 6E) in several replicates (see also Fig. S7). In particular we measured the distances between the γ - phosphate group of the ATP in the active site of the active kinase and the side-chain of Tyr 530 on the substrate molecule and found stable interactions during the 1.2 μ s simulation run. These data confirm the crystal structure and

biochemical results presented in our study supporting further our interpretation and conclusion.

10. Inhibitor. Why did the authors use Ponatinib, which is a type II inhibitor? This is probably why they have trapped this intermediate conformation. Did they try other inhibitors? Type II inhibitors tend to stabilize an open conformation where the N and C-lobes are apart. Did they try a type I inhibitor which may have trapped a more fully close conformation where the activation loop is ordered? Actually, there is only one structure in the PDB showing Src that is phosphorylated on its activation loop, and this is not reported in any publication. This is curious and suggests that the pY416 may be hard to trap.

Response 10: We used Ponatinib because the crystal structure of c-Src in complex with this compound was not known by the time we set up the experiments. We have tried other type-I inhibitors and nucleotides with full-length constructs of WT and Y/F mutants of both the activation and c-terminal segment. Unfortunately, in the case of the full-length constructs the crystals we obtained diffracted poorly. We wanted initially to capture the conformational landscape of the active site, but it must be so dynamic and versatile that it is very difficult to trap. We agree with the reviewer by looking at the ratio between c-Src crystal structures solved so far with a phosphorylated and un-phosphorylated activation loop Tyr.

11. Activation Loop. The authors say that the activation loop, which in this structure must be phosphorylated on Y416, is disordered. They should show this very carefully and indicate exactly which regions are disordered as well as the temperature factors. The Activation Loop begins with the DFG aspartic acid, and this Asp seems to be ordered. Is this correct? Immediately after the p-site (Y419) is the P+1 loop, which is thought to be important for recognizing the P+1 residue in the substrate although this may be different for tyrosine and serine kinases. This region is important as R422 is anchored to E310 in inactive Src, which causes the A-Loop to be in an inhibitory “In” conformation instead of in an “Out” conformation, which is associated with the active state. In this structure the Activation Loop and at least part of the subsequent Activation Segment is in a dynamic intermediate state. Which residues exactly are disordered? In the open and active conformation, the HRD arginine typically interacts with the p-Site residue on the Activation Loop. Since the pY416 is not visible, what about the HRD arginine? If the HRD Arg is not anchored to the phosphate on the Activation Loop then the AL is clearly in some kind of transition state.

Response 11: As the reviewer correctly indicates, the activation loop in our crystal structure is disordered. The activation-loop missing aa sequence goes from residues 410, just after the DFG-motif (aa 407-9) which is ordered and out, to residue 426, so R422 is not captured in the structure and we cannot provide evidence about the activation loop being in an out configuration (active state). However, as we mentioned on **response 6** we do not have evidence the activation loop in the crystal structure is phosphorylated

Reviewer #2:

Remarks to the Author:

In this manuscript, the authors bring forward evidence that c-Src-Y530 is a c-Src auto-phosphorylation site and c-Src-Y419 phosphorylation is required for c-Src-Y530 intermolecular phosphorylation. The authors also show that Tyr419 residing in the activation-loop undergoes fast kinetics and a cis-to-trans phosphorylation-switch in controlling Tyr530 auto-phosphorylation. Interestingly, Tyr419 also controls enzyme specificity and no-catalytic function as a substrate of c-Src. Impressively, c-Src c-terminal residues Y531 to 536 are required for c-Src Y530 and global auto-phosphorylation, and a mutant deleted of these residues seems to show dominant negative effect to the WT c-Src in auto-phosphorylation. These findings provide some new insights on the activation of c-Src from the view of structure. However, there still remain some concerns to be clarified:

Major concerns:

1. Except experiments relating to drosophila, the most of this work was done with purified protein in vitro. The c-Src auto-phosphorylation paradigm was, of course, supposed to occur in a scenario without upstream signaling regulation. I want to know how such auto-phosphorylation paradigm is regulated intracellularly by upstream signals to meet specific biological functions, in other words, the author should find a biological or pathological context under which such c-Src auto-phosphorylation paradigm play a role.

Response 1: In our manuscript we provide evidence of a self-autonomous auto-phosphorylation mechanisms that controls the functional and conformational landscape of c-Src, and how cancer associated c-terminal deleted variants perturb this mechanism. Further, we found that a double phosphorylated c-Src on both the activation loop and c-terminal segment is still an active protein in vitro (Fig. S11A). The fact that dual phosphorylated Src members are found in cancer cells (Hardwick and Sefton, 1997; Nika et al., 2010; Sun et al., 1998) and that is samples from TNBC patients elevated levels of both Y419 and Y530 phosphorylation are found (Nelson et al., 2020) argues that the crosstalk between the two sites could be physiologically important. These previous finding together with our data provides a plausible scenario for an alternative paradigm for the mechanism of c-Src activation driven by an auto-phosphorylation mechanism.

Previous studies demonstrate that CSK mediated phosphorylation of c-terminal Tyr 530 inactivates c-Src only when the protein is not previously auto-phosphorylated (Sun et al., 1998). These previous findings highlighted that auto-phosphorylation may compete with CSK and other c-terminal targeting kinases. Our work dissects the auto-phosphorylation mechanisms in an isolated system with no CSK input. As already indicated, the fact that high levels of dual phosphorylated c-Src is found in cancer cells patient samples of some types of aggressive cancers e.g. TNBC (Nelson et al., 2020) indicates this can be physiologically relevant.

As we state in the discussion: Our work provides answers to two important questions for a better understanding of the role of c-Src in human cancers. On one hand, hyper-activation of c-Src in human cancers might result from the perturbation of auto-phosphorylation and allosteric control. Large-scale genomic sequencing projects indicate that gene amplification and activating mutations in c-Src do not play a

significant role in human tumor biology (Bailey et al., 2018; Curtis et al., 2012). In fact, paradoxical high levels of phosphorylated c-Src in both activation and c-terminal segments are found in aggressive cancer types such as TNBC (Nelson et al., 2020) and NSCLC (unpublished). We have data supporting these findings where a previously phosphorylated c-Src protein (90-120 min) is active and able to phosphorylate an intact substrate surrogate with faster kinetics than the non-phosphorylated protein (Fig. S11A). On the other hand, c-Src may function independent of CSK, which may not always play an anti-oncogenic role through the negative regulation of c-Src in carcinogenesis. In this line, elevated expression of CSK in human cancer cell lines appears to correspond to elevated c-Src protein-tyrosine kinase activity (Watanabe et al., 1995) e.g. CSK is widely expressed in HT-29 and SW620 cells that contain high-levels of active c-Src (Li et al., 1996). Furthermore, genetically engineered mouse models showed that in the absence of CSK, c-Src phosphorylation at Tyr 527 in vivo was reduced to about 20-50% of the level in wild-type cells. These results additionally supported the notion that auto-phosphorylation and/or other kinase could be driving c-terminal tyrosine phosphorylation (Imamoto and Soriano, 1993; MacAuley et al., 1993)

In addition, we find three critical scientific evidences that supports the notion that c-Src intrinsic activity independent of direct upstream inputs can play an important role in cancer cells:

i) c-Src can phosphorylate and activate directly RTKs such as EGFR and RET. For example, it is well established that EGFR is phosphorylated directly on Tyr 845 by c-Src and that this phosphorylation event is important for the biology of cancer cells, in particular DNA synthesis, and malignant cell proliferation (for a comprehensive review see, (Sato, 2013). In the same line, it has been shown that inactive RET mutations that prevent cellular proliferation could be rescued by c-Src and v-Src promoted intermolecular phosphorylation (Kato et al., 2002). We have extensive data showing that c-Src can directly phosphorylate RET on the activation loop both in vitro and in cell-based assays (work in progress)

ii) Implications in the mechanism of resistance to anti-cancer therapies. c-Src have been shown to play an important role in the response to anti-EGFR therapies, and co-treatment with c-Src inhibitors re-sensitizes resistant cancer cells (for a comprehensive review see (Belli et al., 2020).

iii) In the same line, the paradoxical activation of c-Src and its downstream phosphorylation cascade induced by Src-targeted and RTK-targeted kinase inhibitors have huge implications also in the mechanism of resistance to many TKIs and anticancer-therapies (Higuchi et al., 2021). In the paper by Higushi and co-workers, they highlight a self-autonomous mechanism induced by the binding of an ATP-competitor inhibitor that relieves c-Src auto-inhibition and favors an open state compatible with substrate binding (e.g. FAK) and phosphorylation. Whether this model applies also to auto-phosphorylation needs to be further explored.

2. Given Src Tyr419 auto-phosphorylation and Tyr530 phosphorylation has already been well known to regulate c-Src enzymatic activity reversely, the authors should provide evidence that both Tyr419 phosphorylation and Tyr530 phosphorylation occurs simultaneously at the comparable level in mammalian cell or tissue to confirm their in vitro result that Tyr419 and Tyr530 are sequentially phosphorylated. For example, immunohistochemical staining of successive tissue sections with

antibodies against phosphorylated Src-Y419 and Src-Y530, individually, or immunoprecipitation of c-Src followed by WB with antibodies against phosphorylated Src-Y419 and Src-Y530, individually.

Response 2: We have tested a set of different cell lines: TPC1, MZ-CRC-1, KELLY, LAN1, TT, SK-N-AS, MCF7, MCF7CTED, BT459, MDA-MB-231), some of them also treated with LOXO-292 (1 mM, 90 min) in WBs analyses using total c-Src and Tyr419 and Tyr530 phospho-specific antibodies and found that some cell lines display simultaneous high levels of Tyr530 and Tyr 419 phosphorylation such KELLY, LAN1, TT and TPC1-LOXO-292-treated cells (see Fig. S11B). These observations were further supported by HEK293T cells ectopically expressing increasing amounts of c-Src display a dose-dependent increase in the phosphorylation levels of both c-terminal Tyr530 and activation loop Tyr419 (see also response 5)

3. According to the authors' results, c-Src-Y419 auto-phosphorylation is required for c-Src-Y530 phosphorylation. This should be confirmed by replacement of endogenous c-Src with c-Src-Y419F and further detection of c-Src-Y530 phosphorylation signal in mammalian cells.

Response 3: Our findings are based on in vitro experiments using recombinant protein and a highly controlled experimental system. What we conclude is that Y419 is required for Y530 auto-phosphorylation. In a cellular system, we cannot assure that another kinase could be targeting c-terminal tyrosine in the absence of activation loop Y419. To monitor auto-phosphorylation in a cell-based context is extremely difficult, as it is not possible to control the input of other kinases that can interact and phosphorylate c-Src in trans nor the fact that ATP is already present in the system. From our previous experience with other protein kinases e.g. RET in previous published work (see Plaza-Menacho et al Mol Cell 2014, and Cell Reports 2016); the same question was also raised by reviewers. We found that in order to track temporal kinetics of the auto-phosphorylation reaction precisely and accurately, the only possible way was in vitro using highly pure, monodisperse and un-phosphorylated protein in a fully controlled test tube reaction in the presence of exogenous MgCl₂ and ATP in a time-course experiment.

4. The authors show that Src42A-Y400F almost completely rescued the drosophila phenotype caused by overexpression of wildtype Src. This is predictable, because such phenomenon has been previously observed by other groups (Pedraza et al., 2004; Putz, 2019; Forster and Luschnig, 2012; Nelson et al., 2012) and it is well known that Y419F mutation severely disrupts Src enzymatic activity. However, it is unknown whether phosphorylation of Y400 (corresponding to Y419F) is required for the phosphorylation of drosophila tyrosine corresponding to Y530F, so the phosphorylation levels of Y400 and the drosophila tyrosine corresponding to Y530F should be detected in drosophila with overexpression of wildtype Src42A, Src42A-Y400F or Y to F mutant of drosophila tyrosine corresponding to Y530F, together with the observation of corresponding phenotypes.

Response 4: As far as we are aware, in none of the scientific works cited by the referee the Src42A activation loop Y400F mutant allele was used nor evaluated. They used in all of them mainly a constitutively active (CA) allele with a c-terminal Y/F mutation (Y511F), a kinase dead allele (KM) with a mutation in the catalytic lysine (K276M), and a loss of function F80 allele, but not an activation loop mutant allele

(see below). Furthermore, we do not present any phenotypic rescue related to Src overexpression (extra-long tubes) with the Y400F mutant allele. In fact, we rescue the loss of function of Src (F80 allele) by the Y400F mutant (see Fig. 5D and E).

In any case the second part of the comment is an interesting question raised by the reviewer. We have tried to monitor dSrc42A auto-phosphorylation using commercially available c-Src phospho-specific antibodies (phospho-Src Tyr419 (D49G4) CST #6943 and phospho-Src Tyr 530 (ThermoFisher 44- 662G and CST#2105), but they appears not to recognize activation-loop nor c-terminal segment tyrosines in dSrc. We thought initially, this was due to their consensus sequence differing from their human/chicken/mouse homolog sequence from which the antibody was raised against. However, we just found that the phospho-Src Tyr 419 Thermo # 44-660G can recognize dSrc phosphorylated on the activation loop. In any case, as we indicated in the manuscript, we are currently working on the structural and molecular characterization of a full length dSrc42A construct, and the dissection of the molecular mechanisms of auto-activation will be submitted and published as a follow up story due to the time frame given for the resubmission.

A detailed list of the constructs used in the works cited by the referee:

Pedraza et al., 2004: Wild-type src64, src42 and csk cDNA were cloned into the pUAST vector for P-element transformation (Brand and Perrimon, 1993). The Src64YF and Src42YF constitutively active mutants were generated by PCR, confirmed by sequencing and cloned into the pUAST vector. UAS-p21 and GMRGal4 were generous gifts from I Hariharan and M Freeman respectively (Pedraza et al., 2004)

Nelson et al., 2012: The specific alleles used for images are as follows: Src4226-1, Src64KO, nrv223B, dDaamEx68, vermKG07819, and Atp1R2. The UAS-Src42WT and UAS-Src64 lines are described in 22. The UAS-Src42KM transgene contains a K276M mutation that abolishes catalytic activity as described in (Shindo et al., 2008) (Nelson et al., 2012)

Forster and Luschnig, 2012: Src42A26-1, UAS-Src42A.CA (Y511F), UAS-Src42A.KM (K295M) and aa variants UAS-Src42A, UAS-Src42A, UAS-Src42A (Forster and Luschnig, 2012)

Putz et al., 2019: To investigate whether elevated, constitutively-active or kinase-deficient SRC disrupts the N-Cad AJ between R3 and R4, UAS-transgenic lines for Src42, constitutively active Src42CA, kinase deficient Src42KD and Src64 were used (Putz, 2019)

5. According to in vitro data, the authors concluded that cancer associated c-terminal deleted variants inhibit allosterically wildtype c-Src activity by a dominant negative effect. This proposal should be verified by overexpression of these c-terminal deleted variants in drosophila or mammalian cells to further observe the relating phenotypes.

Response 5: We agree with the referee this is an important point. For this purpose, we have conducted experiments in HEK293T cells in which we ectopically expressed increasing amounts of c-Src full-length WT and a c-terminal deleted variant 531X to evaluate the phosphorylation levels of c-Src (see figure below panel A). In addition,

we performed combinatorial experiments in which a fixed amount of c-Src WT plasmid was co-expressed with increasing amounts of the 531X to evaluate the dominant negative effect by the mutant allele (see figure below panel B).

First, we overexpress in HEK293T cells increasing amounts of c-Src WT and we observed a dose dependent increase in the levels of Tyr 419, Tyr 530 and total phospho-tyrosine (see panel A, below). This contrasted with the lower levels of total and Tyr419 phosphorylation levels showed by cells expressing the same amount of the c-Src 531X mutant. These data indicate a potential detrimental effect by the 531X mutant. In the case of Tyr 530 phosphorylation, as expected from our in vitro experiments with recombinant protein, there was no detectable Tyr 530 phosphorylation signal by the c-terminal palindromic deleted variant (see panel A, below). As a control, we used antibodies that detected total c-Src protein, one of them was directed against a C-terminal epitope (present only in the WT c-Src). The second antibody directed to an N-terminal epitope resulted in the identification of both WT and the c-terminal deleted 531X variant (see panel A, below)

Next, we co-expressed c-Src WT and 531X plasmids in HEK293T cells to capture the dominant negative effect of the c-terminal deleted variant on the wild-type allele. In particular, we co-expressed a fixed amount of c-Src WT (0.5 μ g) with increasing amounts of c-Src 531X (0, 0.5, 1, 2 and 4 μ g) and we could not observe any transactivation effect by the WT construct. In fact, we found a detrimental impact not only on the levels of Tyr530 phosphorylation (es we expected) but also on the phosphorylation levels of the activation loop Tyr419 when compared with the 531X construct alone. These data suggest a potential dominant negative effect of the 531X allele over the wild-type (see panel B, below). However, we acknowledge that to monitor auto-phosphorylation and its allosteric regulation in a cell-based system is very complicated and makes us to take cautiously the message about this point in the manuscript. We have opted to remove the conclusive statement in the highlights (see page 3), and see modified text also on the introduction on page 6.

Figure. HEK293T cells were transfected with increasing amounts of the indicated plasmids using Fugene following manufacturer's protocol. Whole cell lysates were subjected to WBs using the indicated antibodies.

6. It is very interesting that c-Src Y419 phosphorylation plays roles in its specificity toward substrate. What is the mechanism? Is Src-Y530 phosphorylation involved in such mechanism. This should be at least discussed.

Response 6:

We agree with the reviewer this is an interesting and unexpected finding. The data we present in this manuscript related to the function of the activation loop is dual. On one hand we show that Y419 is required for the substrate molecule to be presented into the active site of an active kinase molecule for intermolecular phosphorylation (non-catalytic role, see Fig. 4C). On the other hand, we show that a non-phosphorylatable activation loop Y419F mutant is catalytically competent but is able to phosphorylate preferentially a set of substrates e.g. FAK and Src versus others e.g. RET (see Fig. 3B and C-F). Interestingly, when we look at the consensus sequence of an optimal c-Src substrate, it is very similar to the aa sequence of the activation loop of RET GLSDVYEEDEYVKRSQ with a EED motif at position -4, -3 and -2 from Y905 (see Fig. 4G). How can the activation loop control directly the selection of the substrate molecule? We speculate that the dynamic nature of the activation loop will allow to reach the P+1 substrate binding site and sterically impede or modulate substrate access into the active site. Alternatively, the activation loop may prime protein-protein contacts with the substrate molecule for adequate positioning into the substrate binding site prior catalysis.

Minor concerns:

1. There are many spelling mistakes in this manuscript. For example, 1) page 1, address section: "Tel.: +34 +34"; 2) a full stop is needed for the end of Summary; 3) page 8, line 15: "20.000 rpms"; 4) page 11, line 4, "2450U/ml" should be "2450 U/mL"; 5) page 19, title of paragraph 2, "-inter-molecular", and so on.
2. Some scale bars are missed (Fig. 5A and 5B).

Response:

We have corrected the spelling mistakes throughout the text, we thank the reviewer for checking this.

Reviewer #3

Comments:

In this article, authors Cuesta et al., show how auto-phosphorylation of c-Src undergoes conformational changes to bring about two different roles.

Response:

It is not clear to me what the referee means by the two different roles

Two major stumbles of this manuscript is

1. Line numbers are added to indicate the comments

2. The flow of the manuscript is not uniform. It is a mixture of too many writing styles. It is extremely distracting and hard to understand. So authors should make the style of writing in the manuscript to be same and uniform.

Response:

1. My apologies for the inconvenience, I am not sure if there is a format pre-requisite mandatory by Nat Comm. If this would be the case, we will send the revised version with line numbers.

2. Thanks for the detailed editorial assessment of our manuscript. The manuscript has been written by the corresponding author (myself) and the Drosophila section by my colleague Dr. Marta Llimargas. We appreciate this is an elaborated and multidisciplinary work putting together different pieces of information and data from different fields, such molecular biology, biochemistry, structural biology and Drosophila. We acknowledge that for a Drosophila person the manuscript may be difficult to read at some parts due to the hard-core structure-function analyses. We have tried to unify the writing style and simplify the message at some points of the revised manuscript as much as possible.

Minor Comments

In Summary:

The lines "In line with this, we visualize by X-ray crystallography a snapshot of Tyr 530

ready entry prior catalysis" needs to be restructured.

Similarly the lines "We show that c-terminal residue ... the active kinase" has to be broken into two lines.

Response:

We have restructured and break into simpler sentences the indicated text in the summary. Thanks for the suggestion.

In Introduction

pg.4 – line 7 – the word "codified" should be replaced as "codes for"

pg.5- line 20 – the word "corrupted" does not sound scientific and needs to be replaced.

Response:

We have replaced the word "codified" by "coded for" and removed the word "corrupted"

In Materials and Methods

Plasmids

word “codifying” that is occurring in several places needs to be replaced.
pg.7- line 1- word “codifying” needs to be changed to “containing”;
pg.7- line 3 and 11- word “codifying” needs to be changed to “coding”
pg.7- line 8- word “codifying” needs to be changed to “codes for”
pg. 7-lines 7, 8,10 in the word “drosophila” D has to be changed to capital letter
pg. 8 line 15– “20.000” rpm should be 20,000 rpm
pg.10 line 14- “ factor o” must be “factor of”
pg.12 line 8 – “ ion target values were 3e6 for MS (maximum IT of 25 ms) and 1e5 “ – does IT mean ion target, if so IT should be written in brackets as ion target (IT) values 6 in 3e6 should be made as superscript; 5 in 1e5 should be made as superscript
pg.12 line 18- E. coli should be italicized
pg.12 line 21-please expand FDR

Response:

We have done all the corrections indicated by the referee in the M&M section. We appreciate his/her editing effort.

Results

pg.19 line 9- “catalytically” should be changed as “catalytic”
pg.21 line 9-“catalytically impaired” should be changed as “catalytic activity impaired”
pg.22 line 2 Drosophila melanogaster should be italicized not just Drosophila
pg.25 line 12- “ we use in time course auto-phosphorylation assays “must be “we performed a time course of auto-phosphorylation assay using a”

Response: We have done all the corrections indicated by the referee in the results section. We appreciate his/her editing effort.

Discussion

pg.27 line 4- “According with this” should be either made as “Accordingly” or replace with some other word
pg.27 line 20 - words “with: i) results demonstrating that should be rephrased as “with result demonstrates that: (i)”
pg.27 line 25 - statement “play a compartmentalization rather that a functional or catalytic role” should be changed as play a compartmentalization role rather than a functional or catalytic one.
pg.28 line 6 – Line “Auto-phosphorylation...c-terminal Tyr-530” needs rephrasing!
pg.28 line 9- it is mentioned as “See graphic abstract”! Where is it??
pg.28 line 14- it is mentioned as “Step-2”. Which one/Where is Step-1?
pg.28 line 12-Line “The first step...Try 419” is too big and complex to understand ! It has to be made into smaller statements and understandable.

Pg.29 line 5 – It begins with Third, where is second?? Only First of all is mentioned in pg.27 line 11. But second point is missing!

Response: We have done the corrections indicated by the referee in the discussion section. We appreciate his/her editing effort and patience.

Major Comments

The concept and functional role of c-Src has already been shown using several techniques (although authors have missed to refer to those work here)...

Response: I am not sure what concept and functional role of Src is referring to the reviewer on this point. Probably, the reviewer refers to the current paradigm where CSK-phosphorylated c-terminal Tyr 530 engages its own SH2-domain to adopt an inactive

closed-conformation. Our work here provides an alternative model for the mechanism of c-Src regulation by a self-autonomous (intrinsic) mechanisms driven by auto-phosphorylation. We provide evidence of a self-autonomous auto-phosphorylation mechanism (independent of CSK) that controls the functional and conformational landscape of c-Src; and how cancer associated c-terminal deleted variants perturb this mechanism. Our work also reveals a bi-directional crosstalk between the activation and c-terminal segments that controls the allosteric interplay between substrate and enzyme acting kinases during autophosphorylation. First, we demonstrate that c-terminal Tyr 530 is a de facto autophosphorylation site that displays delayed kinetics compared with the much faster kinetics of activation loop autophosphorylation. In particular we dissected the dynamic changes of the intra- and inter-molecular components for the complete mechanism of c-Src autophosphorylation and further dissected the catalytic versus the non-catalytic properties during the autophosphorylation process by biochemistry and structural analyses. We provide further evidence that an activation loop mutant of Src is an active protein when compared with the wild-type. Further, we found that the detrimental effect of a human Src Y419F was not caused by the perturbation of the intact catalytic activity but rather by the perturbation of the non-catalytic properties of Src as a substrate. We validate these findings (i.e. an activation loop mutant of Src is an active protein) in *Drosophila*. We also found that a double phosphorylated c-Src on both the activation loop and c-terminal segment is still an active protein in vitro. The fact that dual phosphorylated Src members are found in cancer cells (Hardwick and Sefton, 1997; Nika et al., 2010; Sun et al., 1998) and that in samples from TNBC patients elevated levels of both Y419 and Y530 phosphorylation are found (Nelson et al., 2020) argues that the crosstalk between the two sites could be physiologically important. These previous finding together with our data provides a plausible scenario for an alternative paradigm for the mechanism of c-Src activation driven by an auto-phosphorylation mechanism, independent of CSK-regulation.

We have now mentioned in the introduction the role of CSK on c-Src Tyr530 phosphorylation including and citing Levinson's Csk-c-Src crystal structure. On new Fig. 2A, we have included a graphical depiction on the role of CSK in the current model for c-Src regulation (see introduction page 5 and new Fig. 1A). Furthermore, we cite previous studies where they identify Tyr 530 as an autophosphorylation site, although in these studies no kinetics nor temporal/dynamic changes of this phosphorylation site were evaluated (Boczek et al., 2019)

...in this study using crystal structures the authors have captured only the events of changes in configuration from closed to open structures, their status of phosphorylation associated on Tyr 416 and Tyr 530 residues. They have shown the importance of the allosteric phospho-switch and their possible implications. This might shed some light on investigation of cancerous cells in future.

Response: In our study, we apply an integrated and multidisciplinary approach, we use biochemistry, biophysics, tandem mass spectrometry LC-MC/MC, X-ray crystallography, MD simulations and an in vivo *Drosophila* model, in order to dissect the structural, molecular and dynamic determinants that control the complete c-Src autophosphorylation process. Our work also reveals a bi-directional crosstalk between the activation and c-terminal segments that controls the allosteric interplay between substrate and enzyme acting kinases via the c-terminal palindromic phospho-motif during autophosphorylation. The fact that dually-phosphorylated Src members are found in cancer cells (Hardwick and Sefton, 1997; Nika et al., 2010; Sun et al., 1998) and that in samples from TNBC patients elevated levels of both Y419 and Y530 phosphorylation are found (Nelson et al., 2020) indicate altogether that the crosstalk between the two sites should be physiologically relevant.

pg.22-Section 6:

The work on Drosophila is not done in the right way to address the role of c-Src mutant in A-loop.

Response: Our Drosophila model is not intended to prove or demonstrate the role of Drosophila Src in cancer; it is intended to confirm and validate in vivo our working hypothesis and results obtained from in vitro experiments showing that a human c-Src Y419F mutant is still an active catalytic protein, when compared with Src WT. In our case, we wanted to assess if Src42A Y400A is an active protein and exerts a transforming phenotype in different tissues in accordance to its catalytic activity (Fig. 5A and B). We have clarified this point when we introduce the Drosophila work (see page 21, results section 6)

For this purpose, we have used an experimental system widely employed in Drosophila research like the Gal4/UAS system using the GMR promoter for ectopic expression in the eye (Pedraza et al., 2004; Putz, 2019; Read et al., 2004; Vidal et al., 2007); the SalEPV promoter for targeted wing expression (Cruz et al., 2009; Molnar et al., 2022; Parsons and Parsons, 1997); and the breathless (btl) promoter for ectopic overexpression in the developmental tracheal system (Forster and Luschnig, 2012; Kanaoka et al., 2023; Nelson et al., 2012). Furthermore, we have performed rescue experiments in a genetic background of loss of function of Src (SrcF80) and show in trachea that the activation loop mutant of Src is able to rescue the phenotype. This is a genetic evidence that Src42A Y400F is active, which is the main rationale and question being answered.

Src products (be Src64B which is not used in this study or Src42A used in this study) in Drosophila are implicated more in cytoskeletal regulation in addition to its role as oncogenes. The authors have performed the overexpression studies using the mutant construct in non-specific tissues which may not reflect the normal or appropriate physiological conditions. The observed phenotypic effects may be due to the role in regulation of cytoskeletal activities and nothing to do with its oncogenic role.

Response: We have over-expressed Src42A WT and A-loop mutant Y400F constructs under three different tissue specific promoters (eye, wing and trachea). We are not evaluating the role of Drosophila Src in cancer or in cancer specific tissues. We are only testing its functionality, whether it is catalytic functional by means of assessing a phenotype in different tissues. We are not trying to identify the targets nor the pathways by which the phenotypes and oncogenic effect are produced (Fig. 5A, B and C)

Instead authors must have used Drosophila model of cancer with any combination of activated mutations (Example Ras Protein or Notch components! Refer Flybase for work on the Drosophila models) to show the functional activity of the mutant construct. Such studies are already shown using Src64B as well as for Src42A. Authors should refer to Flybase for all the work using flies to design experiments.

Overexpression studies using UAS/GAL4 system is completely misleading. Authors must have generated clones in activated Ras85D and/or activated Src42A to understand the functional role. They can also use RAS-CSK system to understand the SRC activity. Otherwise this simple overexpression and rescue experiments may not reflect the appropriate function. Because titration levels of CSK activity on CSK targets cannot be assessed or ruled out by changes in phenotypes. They must be due to non-specific activity.

Response: Again, this is not the question we are trying to answer here. Our *Drosophila* model is not intended to prove or demonstrate the role of *Drosophila* Src in cancer; it is intended to confirm and validate *in vivo* the results obtained from *in vitro* experiments. The rescue experiment addresses this point (Fig. 5C and D). Src function is required in tracheas for proper morphogenesis (Fig. 5 C). In the absence of active Src (SrcF80) there is a clear loss of function phenotype. This phenotype is rescued by expressing SrcY400 in tracheas. This indicates that it is functional under physiological conditions; for example, a catalytically inactive SrcKM, cannot rescue this phenotype, (Nelson et al., 2012).

Following the referee indications, we have referred now in Material and Methods to Flybase for work on the *Drosophila* models

Since this entire work shown in Section 6 needs more work to be performed to prove the functional role of Src42A mutant. This section 6 may be either removed or at least moved to supplementary data.

This work is not yet ready for publication in Nature publication in this stage.

Reviewer #4

Remarks to the Author:

The manuscript of Cuesta et al. addresses an important problem and delivers a significant amount of meticulous work and important results. However, I can't recommend it for publication in Nature Communications in its current form without significant revision. The main problem is that the current context of the problem is not presented properly and the results are not interpreted in a satisfactory way. The following are the particular points I would like to address.

p. 5. "In the current paradigm, phosphorylation at Tyr 530 by c-terminal Src kinase (CSK) is inhibitory, while phosphorylation on Tyr 419 in the activation loop is activating, although neither of these phosphorylation sites by themselves exerts full positive or negative regulatory control"

Here and further in the text, the authors describe the activation of Src, the role of dimerization, and SH2, and SH3 domains, however, the main point of the manuscript is the phosphorylation of Y419 and Y530. They refer the reader to two Roskoski reviews to understand what are the current problems of the current paradigm and mention that "the precise role of Y419 autophosphorylation... not yet fully understood". In my opinion, the double phosphorylation of Src has to be explained in detail as this is the main problem. The role of CSK has to be clearly outlined and put in the context of the Src activation (see for example Fig.8 in Sun et al. 1998 PMID: 9794236). The distinction between cis and trans autophosphorylation should be made in the Introduction to put the results in the general context. The structural aspect of the CSK phosphorylation of Y530 should be also introduced as the authors present their structure of the suggested autophosphorylation complex of Src. They mention Levinson's structure of the CSK/Src complex in the results but it has to be presented in the Introduction as the general picture of the problem looks rather vague and doesn't provide a comprehensive description of the current state.

Response:

We agree with the reviewer comments and suggestions and we have now included in the revised version of the manuscript more details and information about the highlighted points as listed below:

i) We have made more emphasis on the double phosphorylated c-Src paradigm by providing new data on the phosphorylation (see revised introduction page 5, and discussion on pages 33 and new supplemental figure S11B)

ii) We have clearly outlined in the introduction the role of CSK on c-Src Tyr530 phosphorylation including and mentioning Levinson's Csk-c-Src crystal structure. Further, on new Fig. 2A, we have included a graphical depiction on the role of CSK in the current model for c-Src regulation (see introduction page 5 and new Fig. 1A)

iii) We have made a distinction between cis and trans autophosphorylation at the introduction, and we have introduced as well as the distinction on catalytic versus non-catalytic functions in protein kinases (see introduction page 5) to put the results in the general context.

p.17 "c-terminal Tyr 530 and SH2 Tyr 216 show slower rates of autophosphorylation reaching maximum levels at later time points (30-60 min)."

How these times are related to the CSK rates? Can authors comment on the biological relevance of slow (1-hour) rates of autophosphorylation?

Response p.17: In the manuscript by Advani et al. (Advani et al., 2017) the authors look at the phosphorylation of c-Src Tyr530 in vitro by recombinant Csk and Chk, and found that Csk was a better catalyst of the reaction compared with Chk. To reach this conclusion, in these experiments the authors did not performed time-course phosphorylation assays but instead they measured the activity at a final time point of 20 min. Although, these data apparently indicate higher preference of Csk for c-terminal Tyr 530 phosphorylation, without observing the saturation kinetics of the reaction it is difficult to compare the rates of phosphorylation by Csk and those related to c-Src auto-phosphorylation when the assays have been conducted under different conditions.

The slower timing in the process of c-terminal Tyr 530 phosphorylation compared with the activation loop correspond to a sequential and coordinated series of phosphorylation and conformational events, where fast activation loop auto-phosphorylation in cis switches into a trans-phosphorylation component that precedes at later stages the c-terminal intermolecular phosphorylation. In our case, despite we refer to the time required for the auto-phosphorylation reaction to reach saturation to 30-60 min, we are able to detect already significant phosphorylation levels of Tyr 530 from min 15 or earlier by means of WBs using phospho-specific antibodies and tandem LC/MS-MS. We speculate that the slower kinetics of c-terminal auto-phosphorylation can account for a different functional facet where at early time points activation loop phosphorylation is required for the protein to adopt a catalytically competent open conformation and select for the substrate to be phosphorylated i.e. substrate specificity. At later stages, a fully auto-phosphorylated c-Src molecule on both the activation loop and c-terminal segment may adopt a close conformation but presenting the activation loop in an "out" configuration (non-catalytic function) to favor protein-protein interaction with substrates and or other kinases that regulate c-Src function in trans for complex assembly prior phosphorylation (see Fig.S11A)

p.22. Labels on Fig.S6D should be larger. There are panels F, G, and H in Figure S6 but they are not discussed in the text.

Response p.12: The labels are now larger from size 17 to size 20, and the complete set of panels are discussed in the text (see supplemental information, Fig. S6). Please note that in order to focus our work and to avoid redundancy we have removed some structural information related to Fig. S6.

p.24. "This disposition is analogous to the one observed in the crystal structure of the Csk-c-Src complex where the interface was formed by the c-terminal α 1 helix with

the preceding α H- α I loop of c-Src and the α D helix of Csk at the entrance of the active site (Levinson et al., 2008)".

As I mentioned earlier the CSK/Src complex should be mentioned in the Introduction as the reader needs to know that the structural context of Y530 phosphorylation is already known. The word "analogous" is very elusive and doesn't provide an important comparison between the two complexes. My understanding is that the CSK/Src complex is different as the C-terminal part of Src and the palindromic motif are not in contact with the G-loop of CSK. Levinson et al. argue that it is still possible that Y530 can be positioned at the active site of the enzyme. The authors do not discuss the difference and later in the Discussion say that their "crystal structure captures the arrangement of the substrate and active kinases during inter-molecular autophosphorylation" (p. 27). This is not necessarily true. The structure is compatible with the arrangement.

Response p.24: We agree with the reviewer comments and suggestions. In the revised version, the crystal structure of the CSK and c-Src complex is now mentioned in the introduction (see previous response on remarks to the author, and page 5 revised introduction). The reviewer is correct and we should have specified the differences between the crystal structure of the CSK-c-Src complex (Levinson et al. 2008) and our crystal structure. We have now included in the results section the following modified text (see page 25).

"This disposition is similar to some extent to the one observed in the crystal structure of the Csk/c-Src complex where the interface was formed by the c-terminal α I helix with the preceding α H- α I loop of c-Src and the α D helix of Csk at the entrance of the active site (Levinson et al., 2008)". The main different however between the two crystal structures is that in the CSK/c-Src complex the c-terminal segment of the c-Src substrate molecule is not in contact with the G-loop of CSK. In any case, both structures are compatible with the c-terminal Y530 to be positioned for ready entry into the active site of the enzyme".

In the discussion we have modified the text according to the reviewer indications as follow (see page 27)

"Our crystal structure is compatible with the arrangement of the substrate and active kinases during inter-molecular autophosphorylation"

p.25. Throughout Fig. 7 proteins are labeled as WT, 528X, and 531X. It would be helpful for the reader to have the same labeling at the A panel for the C-terminal sequences.

Response p.25: We have now made the nomenclature consistent among the different constructs for the c-terminal sequences on Figure 7

"Unexpectedly, the c-Src 531X mutant, despite having a phosphorylatable c-terminal Tyr 530, lacked also the capacity to auto-phosphorylate on this residue, meaning that residues 531 to 536 are required for c-terminal auto-phosphorylation"

It is not really unexpected as usually; a peptide requires P+1 residue to be docked properly at the active site. In general, the text is full of subjective terms such as "unexpectedly" or "strikingly" (7 times). If the authors consider a particular result surprising they have to explain why or just avoid unnecessary emotional adjectives. What is surprising for one person may be not surprising for another.

Response: We have done enzymatic assays using peptides derived from an intact c-terminal Tyr 530 (STEPQYQPGEN) sequence and also the 531X sequence lacking the right part of the palindrome (STEPQY). We found that both constructs WT and 531X efficiently phosphorylate the 531X sequence-derived STEPQY peptide with no significant differences in their K_M nor V_{max} values. These data conclusively demonstrate that the lack of auto-phosphorylation on Tyr 530 seen by the 531X mutant is not driven by a perturbation of the consensus phospho-motif sequence surrounding Tyr 530 nor differences in their intact catalytic activities. These data have been included in the revised manuscript supplemental Fig. S9.

We get the point of the reviewer, we have tried to avoid the use of unnecessary emotional adjectives in the revised manuscript, thanks for the advice. We have tried to avoid the overuse of e.g. strikingly see page 20 and 22.

p. 26 "A quick inspection of the results reveals that both c-terminal variants display comparable catalytic activities to the wild-type as indicated by total phosphorylation levels of the substrate (Fig. 7E)."

There are obvious differences between the activities. The authors should specify what they consider to be "comparable" and why this level of similarity is enough for their conclusions.

Response p.26: In Fig. 7E we measured and referred to the capacity of active c-Src 3D constructs (WT and c-terminal variants) to phosphorylate a catalytically impaired Src KD K298M mutant as a substrate surrogate. In this assay we aim to measure the intact enzymatic capacity toward the substrate. We found no differences between the 3 constructs. It is true that in this experiment we can still capture the capacity of c-Src 3D to auto-phosphorylate (see upper 50 kDa band) in the presence of the substrate. In this case we can see indeed differences in the levels of auto-phosphorylation (comparable and consistent with data from Fig. 7B), but in this particular experiment what we are measuring is the activity towards the substrate (KD K298M, see lower 32 kDa band in the gel), and this activity does not change among the different constructs tested.

p.27. "we identify a c-terminal palindromic phospho-motif containing Tyr 530 on the substrate molecule engaging the P-loop of the active kinase"

The fact that there is a palindromic sequence at the tail is indeed intriguing. Apparently, the authors consider it to be so important that they use it in the title of the manuscript. However, the importance of the sequences being palindromic is not discussed. What is the author's view on this observation? How conserved is the palindrome? Can it be just a random fact?

Response p.27: We agree with the reviewer that the findings of the palindromic phospho-motif are intriguing and interesting. This motif is highly conserved in all the vertebrate c-Src orthologs from human, chicken, rat to xenopus and zebrafish; However, it is not conserved in invertebrates (*Drosophila*, *C. elegans*, *Hydra*, and bacteria *E. fluvialitis*, and *M. ovata*), see Segawa et al 2007. We do not believe this is a random fact due to the functional, structural and genetic evidence provided that altogether indicate this is an important functional motif. In addition to the 531X (PQYstop) mutant we have generated a palindrome PAYAP mutant. This is a critical mutant lacking Q529 and Q531 side-chains while keeping intact Y530. This mutant lacks the capacity to auto-phosphorylate specifically on c-terminal Y530 despite being functional in terms of auto-phosphorylation and catalytic activity towards peptide substrates (see new Fig. SX10). In total we have made three new mutants targeting the palindromic and c-terminal segment residues, namely a 529X and D521A. See also response 8 to reviewer 1. These results indicate once that the perturbation of the contacts between the c-terminal palindromic motif of the substrate molecule and the G-loop of the active kinase are required for the intermolecular auto-phosphorylation of c-terminal Tyr 530.

"... provides evidence of the existence of a complex allosteric node at the c-terminus of c-Src modulating the crosstalk between substrate- and enzyme-acting kinases"

This is indeed a very interesting finding that is similar to other examples of allosteric communication between the "bottom" of the C-lobe and kinase activity. For example, a set of Herman's papers on Tpk1 (PMIDs: 16751660,19364808). Presenting these results in a broader context would be beneficial for the reader.

Response: As suggested by the reviewer, in order to broaden the context of our findings we have included in the discussion information about other examples of asymmetric allosteric communication in protein kinases. In particular, we refer to the EGFR paradigm of asymmetric crosstalk between the C-lobe of the activating kinase and the N-lobe of the allosterically activated i.e. receiver kinase (Zhang et al., 2006). In this context the C-lobe of the 'activator' contacts the N-lobe of the 'receiver' at points of the α C-helix, the β 4/ β 5 loop and an N-terminal extension of the N-lobe (PDB ID: 2GS6). Formation of this asymmetric kinase dimer induces allosteric changes in the N-lobe extension of the receiver kinase leading to the conformational changes in its α C-helix and A-loop required to switch on the activated state (Zhang et al., 2006). This arrangement is analogous to the one observed for the CDK2/CyclinA complex (Jeffrey et al., 1995), see also PDB ID:1FIN. There are also examples of allosteric communication between the C-lobe of the active kinase and the substrate molecule mediated by the α H and α I helices interface of the kinase e.g. Tpk1 (Deminoff et al., 2006; Deminoff et al., 2009), see discussion page 34.

References

- Advani, G., Lim, Y.C., Catimel, B., Lio, D.S.S., Ng, N.L.Y., Chueh, A.C., Tran, M., Anasir, M.I., Verkade, H., Zhu, H.J., et al. (2017). Csk-homologous kinase (Chk) is an efficient inhibitor of Src-family kinases but a poor catalyst of phosphorylation of their C-terminal regulatory tyrosine. *Cell Commun Signal* 15, 29.
- Bailey, M.H., Tokheim, C., Porta-Pardo, E., Sengupta, S., Bertrand, D., Weerasinghe, A., Colaprico, A., Wendl, M.C., Kim, J., Reardon, B., et al. (2018). Comprehensive Characterization of Cancer Driver Genes and Mutations. *Cell* 174, 1034-1035.
- Belli, S., Esposito, D., Servetto, A., Pesapane, A., Formisano, L., and Bianco, R. (2020). c-Src and EGFR Inhibition in Molecular Cancer Therapy: What Else Can We Improve? *Cancers (Basel)* 12.
- Bjorge, J.D., Pang, A., and Fujita, D.J. (2000). Identification of protein-tyrosine phosphatase 1B as the major tyrosine phosphatase activity capable of dephosphorylating and activating c-Src in several human breast cancer cell lines. *J Biol Chem* 275, 41439-41446.
- Boczek, E.E., Luo, Q., Dehling, M., Ropke, M., Mader, S.L., Seidl, A., Kaila, V.R.I., and Buchner, J. (2019). Autophosphorylation activates c-Src kinase through global structural rearrangements. *J Biol Chem* 294, 13186-13197.
- Brand, A.H., and Perrimon, N. (1993). Targeted gene expression as a means of altering cell fates and generating dominant phenotypes. *Development* 118, 401-415.
- Cruz, C., Glavic, A., Casado, M., and de Celis, J.F. (2009). A gain-of-function screen identifying genes required for growth and pattern formation of the *Drosophila melanogaster* wing. *Genetics* 183, 1005-1026.
- Curtis, C., Shah, S.P., Chin, S.F., Turashvili, G., Rueda, O.M., Dunning, M.J., Speed, D., Lynch, A.G., Samarajiwa, S., Yuan, Y., et al. (2012). The genomic and transcriptomic architecture of 2,000 breast tumours reveals novel subgroups. *Nature* 486, 346-352.
- Deminoff, S.J., Howard, S.C., Hester, A., Warner, S., and Herman, P.K. (2006). Using substrate-binding variants of the cAMP-dependent protein kinase to identify novel targets and a kinase domain important for substrate interactions in *Saccharomyces cerevisiae*. *Genetics* 173, 1909-1917.
- Deminoff, S.J., Ramachandran, V., and Herman, P.K. (2009). Distal recognition sites in substrates are required for efficient phosphorylation by the cAMP-dependent protein kinase. *Genetics* 182, 529-539.
- Forster, D., and Luschnig, S. (2012). Src42A-dependent polarized cell shape changes mediate epithelial tube elongation in *Drosophila*. *Nat Cell Biol* 14, 526-534.
- Hardwick, J.S., and Sefton, B.M. (1997). The activated form of the Lck tyrosine protein kinase in cells exposed to hydrogen peroxide is phosphorylated at both Tyr-394 and Tyr-505. *J Biol Chem* 272, 25429-25432.
- Higuchi, M., Ishiyama, K., Maruoka, M., Kanamori, R., Takaori-Kondo, A., and Watanabe, N. (2021). Paradoxical activation of c-Src as a drug-resistant mechanism. *Cell Rep* 34, 108876.
- Imamoto, A., and Soriano, P. (1993). Disruption of the csk gene, encoding a negative regulator of Src family tyrosine kinases, leads to neural tube defects and embryonic lethality in mice. *Cell* 73, 1117-1124.
- Jeffrey, P.D., Russo, A.A., Polyak, K., Gibbs, E., Hurwitz, J., Massague, J., and Pavletich, N.P. (1995). Mechanism of CDK activation revealed by the structure of a cyclinA-CDK2 complex. *Nature* 376, 313-320.
- Kanaoka, Y., Onodera, K., Watanabe, K., Hayashi, Y., Usui, T., Uemura, T., and Hattori, Y. (2023). Inter-organ Wingless/Ror/Akt signaling regulates nutrient-dependent hyperarborization of somatosensory neurons. *Elife* 12.
- Kato, M., Takeda, K., Kawamoto, Y., Iwashita, T., Akhand, A.A., Senga, T., Yamamoto, M., Sobue, G., Hamaguchi, M., Takahashi, M., et al. (2002). Repair by Src kinase of function-impaired RET with multiple endocrine neoplasia type 2A mutation with substitutions of tyrosines in the COOH-terminal kinase domain for phenylalanine. *Cancer Res* 62, 2414-2422.

Levinson, N.M., Seeliger, M.A., Cole, P.A., and Kuriyan, J. (2008). Structural basis for the recognition of c-Src by its inactivator Csk. *Cell* 134, 124-134.

Li, S., Ke, S., and Budde, R.J. (1996). The C-terminal Src kinase (Csk) is widely expressed, active in HT-29 cells that contain activated Src, and its expression is downregulated in butyrate-treated SW620 cells. *Cell Biol Int* 20, 723-729.

MacAuley, A., Okada, M., Nada, S., Nakagawa, H., and Cooper, J.A. (1993). Phosphorylation of Src mutants at Tyr 527 in fibroblasts does not correlate with in vitro phosphorylation by CSK. *Oncogene* 8, 117-124.

Molnar, C., Reina, J., Herrero, A., Heinen, J.P., Mendiz, V., Bonnal, S., Irimia, M., Sanchez-Jimenez, M., Sanchez-Molina, S., Mora, J., et al. (2022). Human EWS-FLI protein recapitulates in *Drosophila* the neomorphic functions that induce Ewing sarcoma tumorigenesis. *PNAS Nexus* 1, pgac222.

Nelson, K.S., Khan, Z., Molnar, I., Mihaly, J., Kaschube, M., and Beitel, G.J. (2012). *Drosophila* Src regulates anisotropic apical surface growth to control epithelial tube size. *Nat Cell Biol* 14, 518-525.

Nelson, L.J., Wright, H.J., Dinh, N.B., Nguyen, K.D., Razorenova, O.V., and Heinemann, F.S. (2020). Src Kinase Is Biphosphorylated at Y416/Y527 and Activates the CUB-Domain Containing Protein 1/Protein Kinase C delta Pathway in a Subset of Triple-Negative Breast Cancers. *Am J Pathol* 190, 484-502.

Nika, K., Soldani, C., Salek, M., Paster, W., Gray, A., Etzensperger, R., Fugger, L., Polzella, P., Cerundolo, V., Dushek, O., et al. (2010). Constitutively active Lck kinase in T cells drives antigen receptor signal transduction. *Immunity* 32, 766-777.

Parsons, J.T., and Parsons, S.J. (1997). Src family protein tyrosine kinases: cooperating with growth factor and adhesion signaling pathways. *Curr Opin Cell Biol* 9, 187-192.

Pedraza, L.G., Stewart, R.A., Li, D.M., and Xu, T. (2004). *Drosophila* Src-family kinases function with Csk to regulate cell proliferation and apoptosis. *Oncogene* 23, 4754-4762.

Putz, S.M. (2019). Mbt/PAK4 together with SRC modulates N-Cadherin adherens junctions in the developing *Drosophila* eye. *Biol Open* 8.

Read, R.D., Bach, E.A., and Cagan, R.L. (2004). *Drosophila* C-terminal Src kinase negatively regulates organ growth and cell proliferation through inhibition of the Src, Jun N-terminal kinase, and STAT pathways. *Mol Cell Biol* 24, 6676-6689.

Sato, K. (2013). Cellular functions regulated by phosphorylation of EGFR on Tyr845. *Int J Mol Sci* 14, 10761-10790.

Shindo, M., Wada, H., Kaido, M., Tateno, M., Aigaki, T., Tsuda, L., and Hayashi, S. (2008). Dual function of Src in the maintenance of adherens junctions during tracheal epithelial morphogenesis. *Development* 135, 1355-1364.

Somani, A.K., Bignon, J.S., Mills, G.B., Siminovitch, K.A., and Branch, D.R. (1997). Src kinase activity is regulated by the SHP-1 protein-tyrosine phosphatase. *J Biol Chem* 272, 21113-21119.

Sun, G., Sharma, A.K., and Budde, R.J. (1998). Autophosphorylation of Src and Yes blocks their inactivation by Csk phosphorylation. *Oncogene* 17, 1587-1595.

Vidal, M., Warner, S., Read, R., and Cagan, R.L. (2007). Differing Src signaling levels have distinct outcomes in *Drosophila*. *Cancer Res* 67, 10278-10285.

Watanabe, N., Matsuda, S., Kuramochi, S., Tsuzuku, J., Yamamoto, T., and Endo, K. (1995). Expression of C-terminal src kinase in human colorectal cancer cell lines. *Jpn J Clin Oncol* 25, 5-9.

Zhang, X., Gureasko, J., Shen, K., Cole, P.A., and Kuriyan, J. (2006). An allosteric mechanism for activation of the kinase domain of epidermal growth factor receptor. *Cell* 125, 1137-1149.

REVIEWERS' COMMENTS

Reviewer #1 (Remarks to the Author):

Overall the manuscript is much improved. The authors have addressed most of the concerns that were raised by the reviewers. The paper now flows well and the major points are clearly articulated and emphasized. There are still some questions that need to be addressed, but overall I now support publication. This is careful work that addresses some important questions that are big challenges for the signaling community overall not just for Src. I consider this to be a seminal contribution although there remains the question of how relevant these steps are in cells vs. with the purified proteins. However, it is very clear that auto-phosphorylation of tyrosine kinases is far more complex than we initially anticipated.

Figures 1 and 2 are especially nice as they highlight the excellent yield that results from their modified protocol and also the kinetics of the three sites that undergo phosphorylation. The authors now also nicely discriminate between cis- autophosphorylation which happens fast and is concentration independent vs. trans-auto-phosphorylation, which is concentration dependent and represents a protein:protein phospho-transfer. The authors also look at peptide phosphorylation which represents classic Michaelis-Menten (MM) kinetics where the substrate is in large excess over the enzyme, while the cis- and trans- phosphorylations that take place with the protein do not represent MM kinetics. Somewhere the authors should emphasize this as I think many do not appreciate this distinction. It is so easy to use peptide substrates, but, as the authors show here, the results are different from when you use a protein substrate.

Points to address.

1. The numbering that the authors use for the P-sites is different from the published structures where Y419 is actually Y416 and Y530 is Y527 and Y216 is Y213. The authors should explain why there is this discrepancy or make their numbers consistent with the other numbering (ie in the 2src structure).

2. The authors should mention something more about Y216 as it seems to be robustly phosphorylated, and there does not seem to be much in the literature in this site. It is in the SH2 domain and if it is phosphorylated there is a nearby Arg in the SH2 domain that would likely interact. This might neutralize the P-Y230 site and "free up" the C-tail. Would this correspond to the more open construct that the authors discuss in figure 6? Something should be said about the importance of this site as all three seem to be phosphorylated in cancer cells. In the figures and in the text the authors need to state that this is a tri-phosphorylated protein, not a bi-phosphorylated protein. Even if they do not understand why this is so, it is the tri-phosphorylated protein that accumulates in cancer and it likely keeps the Y530 from interacting with the SH2 domain.

3. The authors should include in Figure 2 the cartoon showing the location of the three P-sites that the authors show in their response to the reviewers (p 5). This could be the last panel on that page where there is already a space.

4. The authors should show the entire Activation Segment sequence in Figure 3 where the AL tyrosines are aligned. Even though the number of residues in the AL differ for the various kinases, it would be nice to have the alignment. Also I think that the first residue in the Src sequence as shown is wrong. All of the sequences should start with the DFG motif to orient the reader and the authors could highlight which residues are conserved in all. They could extend the sequence all the way to the DxW motif at the beginning of the F Helix but at least they should extend it to the APE motif and have the FG motif and the APE motif included in the structure.

5. The authors should include the structure figure, which is now in the supplement as Figure 6, in the main text. It is an important part of their work, and they discuss extensively the details of the inhibitor binding and the dimer interface that is mediated by

the C-tail. If there is a limitation of figures, then Figure 1 on the expression could go in the supplement.

6. The authors should mention/emphasize that the C-tail and also the Activation Loop are all intrinsically disordered regions (IDRs) which allows for the flexibility that they see. They do emphasize that the N-terminal SH1 domain is intrinsically disordered.

7. The authors never see the fully extended AL even when it is phosphorylated. In the structure this is presumably because the type II inhibitor forces the kinase domain into an open conformation and separates the N and C-lobe similar to what was seen for LRRKs (AC Chem Biol. Weng, et al, 2022). Once glutamate in the C-helix is flipped to interact with the Lys in B3, the D corresponding to the DFG motif should also become part of this triad. It is not part of the triad which they do not mention, and, as the authors point out, the R-spine is broken because of the type II inhibitor. The r-sine and the triad are all hallmarks of an active kinase that can do trans-phosphorylation of a heterologous substrate. The final cartoon is a bit misleading because the type II inhibitor distorts the AL.

8. The simulations are interesting although only one of the replicates seems to correlate with the open and closed states. This suggests that the AL does not (at least over the time frame that they tested) spontaneously flip into its extended "active-like" conformation when you add ATP. Is the protein phosphorylated in the simulation? Clearly the inhibitory helix is broken but that is not very stable anyway.

Errors in spelling/grammar. There are multiple spelling and grammar mistakes. I have caught only a few here. It should be carefully edited.

Summary. asymmetric vs. assymmetric. Change globally.

Summary. ...prior to catalysis.

Intro. Line 4. ...that led to the discovery...

Intro, l. 11 understanding of proteins with tyrosine kinase activity.

Bottom, p 3 . The SH2 catalytic domains display...

P4, few studies to date...

P4. ...in the protein kinase field.

P 16, four lines from bottom. ...scanning fluorometry (DMS) (Figure 1E).

P 17. 5 lines from bottom: remove interestingly. These data highlight..

P 20 line 3 ...that switches over time...

P 20. ...the lack of the intermolecular...

...an activation loop mutant that cannot...

...showed constant catalytic activity over the range..

...used, whereas the ..

...significant...

...displayed increased catalytic efficiency compared to...

...a c-Src Y419F mutant did efficiently..

...activation loop; however, nd contrary to the ...

P21 ...however, to a lower level...

...controls, in addition to, ..

...developmental trachea, we ...

P23 ...causing the activation loop to...

P 24 ...is composed of the ...

...facing towards the aD helix...

...a close up of the...

...the main difference, however,

P25 ...furthermore, we probed the ...

... is localized 10 A below the...

... bottom again ...located 10 A below the ATP for at least 100 ns...

P 26 ...at codon 531 had a detrimental effect...

De facto should be consistently italicized throughout the paper.

Reviewer #2 (Remarks to the Author):

The authors have addressed all my concerns and I am satisfied by the revision.

Reviewer #3 (Remarks to the Author):

Cuesta et al., Nat. Comm. NCOMMS-22-39284

Post-Revision

Comments/Remarks to the Authors:

The Authors should have listed the experiments deleted, modified and added to the revised version. They have not made significant changes to the work, instead they have remixed the experiments and provided the data.

The flow of presentation is not uniform and not clearly stated, making it hard for the reviewers to understand. For example, in the abstract the lines "Perturbation of the phosphor-motif....allosterically inhibiting the active kinase" is so complex to understand.

There are several places at which modifications are required. Since line numbers are not included again in the revised manuscript, it is so hard to mention all the grammatical mistakes and sentence framing issues. Example pg.4 in the third line from bottom "Protein kinase 'filed' that are", check for the meaning.

Authors have written the manuscript throughout in a handwavy manner. For Example, pg.4 last two lines, "SFKs expression....of the diseases", authors have mentioned "some hemotologic malignancies". What do they mean by some? Even the Abstract is not comprehensive and compact.

Experiments are not well-planned to understand the in vivo role of Src proteins (both wildtype and mutant). Since there are previous reports in Drosophila showing different outcomes with varying levels of Src signaling (Vidal et al., 2007), so will be the case with mutant construct used in this study. In addition, there are also reports showing that Src42A (using GMR GAL4) is not a potent construct in activation of Src signaling (Poon et al., 2018). Instead authors

Also Src-oncogene at 42A (Src42A) encodes a non-Receptor Tyrosine Kinase (nRTKs) and is shown to

have a dual role of both inhibiting and activating tyrosine phosphorylation. Therefore simple overexpression will not convey the appropriate physiological function. In addition, Src42A can target any protein with SH2 domain. So phenotypes observed and rescue by simple overexpression can be due to non-specific activity.

Also just using three different tissue-specific GAL4 drivers does not justify the purpose of authors to validate the invitro work. What inference can be made by such unmonitored expression levels? Why Authors have not even monitored the protein expression levels with the phenotype observed. Authors should have generated overexpression clones/MARCM clones in eyes or wingdiscs or used Split Fluorescent protein system (Kamiyama et al., 2021) in a cell-specific manner to understand the physiological function.

The study is all about two tyrosines and their regulation . But only one mutant has been made? The authors have not proved whether they have activation role or inhibiting role. Do they show dominant negative role?

During the revision time, authors could have extended the work to make further analysis. Without clarifying the appropriate role, just the preliminary result should not inserted in this work.

The Drosophila work need not be part of this work because they are extremely negligible relevant data is generated by them to understand the physiological role. Infact these experiments are raising more questions than clarifying the question addressed.

Therefore this Drosophila work should be saved for extended work of this present study, rather than including them here.

Reviewer #4 (Remarks to the Author):

I would like to thank the authors for their comprehensive answers. My concerns were properly addressed and I can recommend the manuscript for publication.

Reviewer 1:

Overall the manuscript is much improved. The authors have addressed most of the concerns that were raised by the reviewers. The paper now flows well and the major points are clearly articulated and emphasized. There are still some questions that need to be addressed, but overall, I now support publication. This is careful work that addresses some important questions that are big challenges for the signaling community overall not just for Src. I consider this to be a seminal contribution although there remains the question of how relevant these steps are in cells vs. with the purified proteins. However, it is very clear that auto-phosphorylation of tyrosine kinases is far more complex than we initially anticipated.

Figures 1 and 2 are especially nice as they highlight the excellent yield that results from their modified protocol and also the kinetics of the three sites that undergo phosphorylation. The authors now also nicely discriminate between cis-autophosphorylation which happens fast and is concentration independent vs. trans-auto-phosphorylation, which is concentration dependent and represents a protein:protein phospho-transfer. The authors also look at peptide phosphorylation which represents classic Michaelis-Menten (MM) kinetics where the substrate is in large excess over the enzyme, while the cis- and trans-phosphorylations that take place with the protein do not represent MM kinetics. Somewhere the authors should emphasize this as I think many do not appreciate this distinction. It is so easy to use peptide substrates, but, as the authors show here, the results are different from when you use a protein substrate.

Response:

We are happy with the positive evaluation of the revised work. We appreciate the very careful and knowledgeable evaluation of our work by reviewer 1. Following the indications of the reviewer, we have emphasized the difference between phosphorylation of exogenous peptides following Michaelis-Menten (M-M) kinetics and the cis- and trans-autophosphorylation of a protein kinase that do not represent M-M kinetics (see results section page 10). We have included the following text:

"Note that peptide phosphorylation follows classic Michaelis-Menten kinetics where the substrate is in large excess over the enzyme, while the cis- and trans-phosphorylation that take place with the intact protein do not represent Michaelis-Menten kinetics, and are usually subjected to allosteric control".

Points to address.

1. The numbering that the authors use for the P-sites is different from the published structures where Y419 is actually Y416 and Y530 is Y527 and Y216 is Y213. The authors should explain why there is this discrepancy or make their numbers consistent with the other numbering (i.e. in the 2src structure).

Response:

As we indicate in material and methods we have used constructs with the human sequence of c-Src (UniProtKB accession: P1293), and consequently we have kept the human numbering. It is true that historically people have used and kept the chicken sequence numbering even though they have worked and studied the human sequence (e.g. PDB: 1FMK, Xu et al, 1997). So human Tyr 216 corresponds to chicken Tyr 213, and in the same way Tyr419 (human) is equivalent to Tyr 416 (chicken) and the same applies to Tyr 530 and Tyr 527. We have checked throughout the text so we are consistent with our own data.

2. The authors should mention something more about Y216 as it seems to be robustly phosphorylated, and there does not seem to be much in the literature in this site. It is in the SH2 domain and if it is phosphorylated there is a nearby Arg in the SH2 domain that would likely interact. This might neutralize the P-Y230 site and "free up" the C-tail.

Would this correspond to the more open construct that the authors discuss in figure 6? Something should be said about the importance of this site as all three seem to be phosphorylated in cancer cells. In the figures and in the text the authors need to state that this is a tri-phosphorylated protein, not a bi-phosphorylated protein. Even if they do not understand why this is so, it is the tri-phosphorylated protein that accumulates in cancer and it likely keeps the Y530 from interacting with the SH2 domain.

Response:

The referee is totally right, we should have said more and speculate in fact that this residue when phosphorylated can be implicated in the liberation of the c-tail from the SH2 domain. We have commented on this in the results section page 7.

3. The authors should include in Figure 2 the cartoon showing the location of the three P- sites that the authors show in their response to the reviewers (p 5). This could be the last panel on that page where there is already a space.

Response:

Following the reviewer's indication we have included a cartoon on figure 2 depicting the third phospho-site summarizing the data from panel 2B and C, and introducing for the first time the self-autonomous regulation of c-Src by autophosphorylation (see Fig. 2e, and page 7). This implies that in our final model (Fig. 9) a fully phosphorylated c-Src molecule could adopt also an extended conformation. Please note that our MS data identified many other autophosphorylation sites (see supplementary figure 3) that would also contribute to the final conformational and phosphorylation state of the protein. We have specifically mentioned this modifying accordingly the figure 9 and cite on page 7:

"Autophosphorylated Tyr 216 on the SH2 domain can coordinate with the side chain of R208 (see PDB ID: 2SRC) and contribute to the liberation of the c-terminal Tyr 530, in line with a model where a tri-phosphorylated c-Src molecule would adopt an extended conformation (see Fig. 2e)".

4. The authors should show the entire Activation Segment sequence in Figure 3 where the AL tyrosines are aligned. Even though the number of residues in the AL differ for the various kinases, it would be nice to have the alignment. Also I think that the first residue in the Src sequence as shown is wrong. All of the sequences should start with the DFG motif to orient the reader and the authors could highlight which residues are conserved in all. They could extend the sequence all the way to the DxW motif at the beginning of the F Helix but at least they should extend it to the APE motif and have the FG motif and the APE motif included in the structure.

Response:

Following the reviewer's indication we have included now a proper sequence alignment from the entire activation segment from the DFG- to the APE-motifs see new Fig. 3g.

5. The authors should include the structure figure, which is now in the supplement as Figure 6, in the main text. It is an important part of their work, and they discuss extensively the details of the inhibitor binding and the dimer interface that is mediated by the C-tail. If there is a limitation of figures, then Figure 1 on the expression could go in the supplement.

Response:

We agree with the reviewer and have now moved Fig. S6 to the main text and is now Fig. 6. Table S1, is now part of the main text and appears as table 1.

6. The authors should mention/emphasize that the C-tail and also the Activation Loop are all intrinsically disordered regions (IDRs) which allows for the flexibility that they see. They do emphasize that the N-terminal SH1 domain is intrinsically disordered.

Response:

We agree with the reviewer and mention now at the results section (page 3) that the activation segment of c-Src is an intrinsically disordered region. We mention in the text:

"The SH1 catalytic domain display intrinsic tyrosine kinase activity, which is further regulated by highly dynamic activation and c-terminal terminal regulatory segments, which are also intrinsically disorder regions".

7. The authors never see the fully extended AL even when it is phosphorylated. In the structure this is presumably because the type II inhibitor forces the kinase domain into an open conformation and separates the N and C-lobe similar to what was seen for LRRKs (AC Chem Biol. Weng, et al, 2022). Once glutamate in the C-helix is flipped to interact with the Lys in B3, the D corresponding to the DFG motif should also become part of this triad. It is not part of the triad which they do not mention, and, as the authors point out, the R-spine is broken because of the type II inhibitor. The r-sine and the triad are all hallmarks of an active kinase that can do trans-phosphorylation of a heterologous substrate. The final cartoon is a bit misleading because the type II inhibitor distorts the AL.

Response:

We agree with the reviewer that once the glutamate in the α C helix engages the catalytic lysine on β 3, then the glutamate of the DFG-motif become part of the triad, but in our structure, it is not the case (we mention this now on the result section page 24). This is a signature of an active kinase as the reviewer points out. In our crystal structure the R-spine is broken due to the binding of the type-II inhibitor and consequently the triad cannot be form, but the α C- β 3 catalytic salt bridge is formed and not broken by the inhibitor. We mention on page 13-14:

"Note that the α C- β 3 catalytic salt bridge is still formed and is not broken by the inhibitor, however the glutamate of the DFG-motif that is part of the catalytic triad is not engaged due to the breaking of the R-spine by the compound".

8. The simulations are interesting although only one of the replicates seems to correlate with the open and closed states. This suggests that the AL does not (at least over the time frame that they tested) spontaneously flip into its extended "active-like" conformation when you add ATP. Is the protein phosphorylated in the simulation? Clearly the inhibitory helix is broken but that is not very stable anyway.

Response: Our molecular dynamics simulations start from the active conformation with ATP already bound to both monomers. Both monomers are also phosphorylated at Y419 in the simulations. We would like to clarify that our simulations do not capture the closed to open transition of the activation loop as the simulation already starts off in the open (active) conformation. Additional details are provided in the methods under "Molecular dynamics simulations".

To recapitulate our results, we wanted to highlight three replicates which capture Y530 from the substrate Src moving within 10 Å of the ATP molecule which is bound to the active site of the enzyme Src. Shown in the supplementary data, these were replicate 4 from 350 - 1200 ns, replicate 5 from 350 - 600 ns, and replicate 8 from 850 - 1200 ns.

Errors in spelling/grammar. There are multiple spelling and grammar mistakes. I have caught only a few here. It should be carefully edited.

Summary. Summary. Intro. Line 4. Intro, l. 11 Bottom, p 3

asymmetric vs. assymetric. Change globally. ...prior to catalysis.

...that led to the discovery...

understanding of proteins with tyrosine kinase activity. . The SH2 catalytic domains display...

P4,

P4.

P 16,

P 17.

P 20 line 3 ...that switches over time...

P 20.

P21

P23 P 24

P25

...the lack of the intermolecular...

...an activation loop mutant that cannot...

...showed constant catalytic activity over the range.. ...used, whereas the ..

...significant...

...displayed increased catalytic efficiency compared to... ...a c-Src Y419F mutant did efficiently..

...activation loop; however, nd contrary to thehowever, to a lower level...

...controls, in addition to,..

...developmental trachea, we ...

...causing the activation loop to...

...is composed of the ...

...facing towards the aD helix...

...a close up of the...

...the main difference, however,

...furthermore, we probed the ...

few studies to date...

...in the protein kinase field.

four lines from bottom. ...scanning fluorometry (DMS) (Figure 1E). 5 lines from bottom: remove interestingly. These data highlight..

... is localized 10 Å below the...

... bottom again ...located 10 Å below the ATP for at least 100 ns... P 26 ...at codon 531 had a detrimental effect..

De facto should be consistently italicized throughout the paper.

Response:

We have corrected all the indicated spelling and grammatical errors

Please note that on page 15 and 16 we have corrected the text for clarity as follows:

"where Y530 is stably localized below a distance of 10 Å from the γ -phosphate group of the ATP molecule"

Reviewer #3 (Remarks to the Author):

Cuesta et al., Nat. Comm. NCOMMS-22-39284

Post-Revision

Comments/Remarks to the Authors:

The Authors should have listed the experiments deleted, modified and added to the revised version. They have not made significant changes to the work, instead they have remixed the experiments and provided the data.

The flow of presentation is not uniform and not clearly stated, making it hard for the reviewers to understand. For example, in the abstract the lines "Perturbation of the phospho-motif, allosterically inhibiting the active kinase" is so complex to understand.

There are several places at which modifications are required. Since line numbers are not included again in the revised manuscript, it is so hard to mention all the grammatical mistakes and sentence framing issues. Example pg.4 in the third line from bottom "Protein kinase ~filed" that are, check for the meaning. Authors have written the manuscript throughout in a handwavy manner. For Example, pg.4 last two lines, "SFks expression of the diseases", authors have mentioned "some hemotologic malignancies". What do they mean by some? Even the Abstract is not comprehensive and compact.

Experiments are not well-planned to understand the in vivo role of Src proteins (both wildtype and mutant). Since there are previous reports in Drosophila showing different outcomes with varying levels of Src signaling (Vidal et al., 2007), so will be the case with mutant construct used in this study. In addition, there are also reports showing that Src42A (using GMR GAL4) is not a potent construct in activation of Src signaling (Poon et al., 2018). Instead authors

Also Src-oncogene at 42A (Src42A) encodes a non-Receptor Tyrosine Kinase (nRTKs) and is shown to have a dual role of both inhibiting and activating tyrosine phosphorylation. Therefore simple overexpression will not convey the appropriate physiological function. In addition, Src42A can target any protein with SH2 domain. So phenotypes observed and rescue by simple overexpression can be due to non-specific activity.

Also just using three different tissue-specific GAL4 drivers does not justify the purpose of authors to validate the invitro work. What inference can be made by such unmonitored expression levels? Why Authors have not even monitored the protein expression levels with the phenotype observed. Authors should have generated overexpression clones/MARCM clones in eyes or wingdiscs or used Split Fluorescent protein system (Kamiyama et al., 2021) in a cell-specific manner to understand the physiological function.

The study is all about two tyrosines and their regulation. But only one mutant has been made? The authors have not proved whether they have activation role or inhibiting role. Do they show dominant negative role?

During the revision time, authors could have extended the work to make further analysis. Without clarifying the appropriate role, just the preliminary result should not be inserted in this work.

The Drosophila work need not be part of this work because they are extremely negligible relevant data is generated by them to understand the physiological role. Infact these experiments are raising more questions than clarifying the question

addressed.

Therefore this Drosophila work should be saved for extended work of this present study, rather than including them here.

Response: As we already mentioned in our previous response, the Drosophila work was intended to test in an *in vivo* system the functionality of an activation loop mutant Src42A Y400F. We did not pretend to carry out a comprehensive analysis of all the putative activities and physiological role that the mutant protein may have in *in vivo* conditions. We agree with the reviewer that Src42A may have many different targets and this analysis would represent a new research project itself that is beyond the scope of this paper. In this study we find that an activation loop mutant protein is not a dead-protein and is functional in Drosophila because it can produce phenotypes (in contrast to what was expected, and in line with our own results) in conditions of overexpression. In addition, the rescue experiments indicate that the protein can act as a wild type protein restoring the defects of lack of Src activity, in contrast to a kinase-dead form that cannot rescue the defects (Nelson et al. 2012). We have included data with a loss of function Src^{KM} mutant in trachea, that shows contrary to the Y400F mutant, developmental defects (see Fig 5d). Altogether our results strongly suggest that Src42A Y400F is a functional protein that behaves similar to the wild type, in agreement with the strong biochemical and structural data of this manuscript.

In light of the comments of the reviewer 3, we have slightly modified the text and figure accordingly to make this message clearer (see results section, and new Fig. 5d). We have stated in the results (page 12-13) and in the discussion (page 20) sections that expression levels were not directly assessed and that there is a possibility that the phenotype is due to worse accumulation of the mutant protein.